# Characteristics and causes of natural and human-induced landslides in a tropical mountainous region: the Rift flank west of Lake Kivu (DR Congo)

Jean-Claude Maki Mateso[1,2], Charles L. Bielders[2], Elise Monsieurs[3,4,5], Arthur Depicker[6], Benoît Smets[3,7], Théophile Tambala[1], Luc Bagalwa Mateso[1] and Olivier Dewitte[3]

[1]Centre de Recherche en Sciences Naturelles, Department of Geophysics, Lwiro, DR Congo

[2]Université catholique de Louvain, Earth and Life Institute – Environmental Sciences, Louvain-La-Neuve, Belgium

[3]Royal Museum for Central Africa, Department of Earth Sciences, Tervuren, Belgium

[4]University of Liège, Department of Geography, Liège, Belgium

[5]F.R.S.-FNRS, Brussels, Belgium

[6]KU Leuven, Department of Earth and Environmental Sciences, Belgium

[7]Vrije Universiteit Brussel, Department of Geography, Brussels, Belgium

*Correspondence to*: Jean-Claude Maki Mateso (jean-claude.maki@uclouvain.be; makigeo2013@gmail.com)

## Abstract

Tropical mountainous regions are often identified as landslide hotspots with growing population pressure. Anthropogenic factors are assumed to play a role in the occurrence of landslides in these densely populated regions, yet the relative importance of these human-induced factors remains poorly documented. In this work, we aim to explore the impact of forest cover dynamics, roads and mining activities on the characteristics and causes of landslides in the Rift flank west of Lake Kivu in the DR Congo. To do so, we compile a comprehensive multi-temporal inventory of 2730 landslides. The landslides are of different types and are grouped into five categories that are adapted to study the impact of human activities on slope stability: old (pre 1950's) and recent (post 1950's) deep-seated landslides, shallow landslides, landslide associated mining and landslides associated with road construction. We analyze the landslides according to this classification protocol via frequency-area statistics, frequency ratio distribution and logistic regression susceptibility assessment. We find that natural factors contributing to the cause of recent and old deep-seated landslides were either different or changed over time. Under similar topographic conditions, shallow landslides are more frequent, but of smaller size, in areas where deforestation has occurred since the 1950's. We attribute this size reduction to the decrease of regolith cohesion due to forest loss, which allows for a smaller minimum critical area for landsliding. In areas that were already deforested in 1950's, shallow landslides are less frequent, larger, and occur on less steep slopes. This suggests a combined role between regolith availability and soil management practices that influence erosion and water infiltration. Mining activities increase the odds of landsliding. Landslides associated with mining and roads are larger than shallow landslides but smaller than the recent deep-seated instabilities, and they are controlled by environmental factors that are not present under natural conditions. Our analysis demonstrates the role of human activities on the occurrence of landslides in the Lake Kivu region. Overall, it highlights the need to consider this context when studying hillslope instability characteristics and distribution patterns in regions under anthropogenic

pressure. Our work also highlights the importance of using landslide classification criteria adapted to the context of the Anthropocene.

## 1 Introduction

Tropical mountainous regions are often identified as landslide hotspots with particularly vulnerable populations (Vanacker et al., 2003; Broeckx et al., 2018; Froude and Petley, 2018; Emberson et al., 2020). Nevertheless, the current knowledge on landslide processes in these regions remains commonly limited as it is mostly derived from susceptibility models made at continental or global levels (Stanley and Kirschbaum, 2017; Broeckx et al., 2018). Because they are not based on detailed local inventories, such models do not allow to properly consider region-

specific characteristics and causes of landslides (Depicker et al., 2020).

The growing demographic pressure and widespread land use and land cover (LULC) changes are expected to increase the frequency and impacts of landslides in tropical mountainous regions, especially in rural environments (Vanacker et al., 2003; Sidle et al., 2006; DeFries et al., 2010; Mugagga et al., 2012; Guns and Vanacker, 2014; Froude and Petley, 2018; Depicker et al., 2021a; Muñoz-Torrero Manchado et al., 2021). Deforestation and the

associated loss of tree roots usually lower the slope stability by decreasing the effective regolith cohesion and altering surface and subsurface water drainage patterns; whose effects are particularly pronounced on the occurrence of shallow landslides (Sidle and Bogaard, 2016). Mining, quarrying and road construction alters the environment and commonly increases landslide occurrence (e.g. Sidle et al., 2006; Brenning et al., 2015; Arca et al., 2018; McAdoo et al., 2018; Vuillez et al., 2018; Muñoz-Torrero Manchado et al., 2021;Tanyaş et al., 2022).

However, the impact of these anthropogenic factors on landslide processes (e.g. types, size, dynamics) depends on their timing and their legacy effect. It also depends on other environmental conditions such as slope angle and lithology (Depicker et al., 2021b). Developing further our understanding of landslides and their natural- and human-induced causes is therefore needed, especially in regions such as the tropics where the dearth of data is commonplace (Dewitte et al., 2022).

To achieve this, a detailed multi-temporal regional landslide inventory spanning several decades is essential (Guzzetti et al., 2012). New methodologies have been proposed in the past years to automatically map landslides with the use of, for example, Earth Observation data and machine learning techniques (e.g. Prakash et al., 2021). 2021). However, such automatic approaches only perform well with recent landslides with a clear spectral signature. Furthermore, they are not always well adapted to an accurate understanding of the processes (Jones et

al., 2021), especially when the landscapes are complex and highly influenced by human activities (Jacobs et al., 2018). The need for a visual identification of landslides is even more important when the movements that are studied are older and have occurred at an unknown period, much before the availability of satellite images (Van Den Eeckhaut et al., 2005; Pánek et al., 2021).

Historical aerial photographs offer the best opportunity at the regional level to work across several decades, both

to compile a landslide inventory but also to reconstruct LULC changes (Glade, 2003; Guns and Vanacker, 2014; Shu et al., 2019). It is complementary to very high spatial resolution satellite images such as those available on © Google Earth (Fisher et al., 2012), which are widely used in the identification of landslides in many environments (e.g. Broeckx et al., 2018; Pánek et al., 2021). Fieldwork is also essential in order to validate observations made

from the different image sources, to discriminate between deep-seated and shallow processes, or to confirm depth estimates (Dewitte et al., 2021). Field surveys also help to understand the role of human activities on slope dynamics (Dewitte et al., 2021). Overall, sufficiently long and precise multi-decadal records of landslide activity, types and LULC are rare (e.g. Glade, 2003; Guns and Vanacker, 2014; Shu et al., 2019).

The aim of this work is to explore the role played by natural and human factors on the occurrence of landslides in a rural tropical mountainous region under high anthropogenic pressure. More specifically, we are interested in the Rift flank west of Lake Kivu, a region in the DR Congo where recent studies have shown that landslides are frequent and that recent deforestation has impacted the occurrence of shallow landslides (Maki Mateso and Dewitte, 2014). We aim to: (1) further develop the existing landslide dataset and compile a comprehensive multi-temporal regional landslide inventory spanning several decades; (2) describe the general characteristics of the landslides, and (3) analyze their causes according to different controlling factors, with special attention to multi-decadal forest cover dynamics. Historical aerial photographs and field surveys are key sources of information in this study.

## 2 Environmental settings and current knowledge of the landslide processes

The study is conducted in the Rift flank west of Lake Kivu in the DR Congo (Fig. 1a). It is one of the most seismic regions of the African continent, crossed by active faults and composed of six main rock types of varying age (Fig. 1b) (Delvaux et al., 2017; Laghmouch et al., 2018). A significant portion of the study area is made of lithologies from the Archaen, the Mesoproterozoic and the Neoproterozoic, with various degrees of chemical weathering and fracturing (Kampunzu et al., 1998). Lastly formed rocks are the old Neogene basalts, highly weathered, that were deposited between 11-4 Ma. The presence of mineral resources (gold and 3T minerals - tin, tantalum and tungsten) favours the proliferation of, often illegal, artisanal and small-scale mining and quarrying (Van Acker, 2005; Geenen, 2012; Bashwira et al., 2014). Industrial mining is not present in the region and there is no new road construction associated with it (Bashwira et al., 2014).

The region has a tropical savannah/monsoon climate tempered by its elevation (Peel et al., 2007). The natural vegetation is mainly montane forest, still preserved in the Kahuzi-Biega National Park (Imani et al., 2017).Between the 17th and 18th centuries, the first intense human disturbance occurred as result of deforestation (Nzabandora and Roche, 2015). The roads built during the late 19[th] and first half of the 20[th] centuries 20[th] played a key role on further expanding the intense human disturbance (Aleman et al., 2018). There has been significant deforestation and forest loss in recent decades as well (Basnet and Vodacek, 2015; Depicker et al., 2021a,b). Selective logging is done for energy needs, house construction, furniture production and dugout canoes. Clearcutting, mostly small-scale, is associated with agriculture, mining and quarrying activities and road construction (Musumba Teso et al., 2019; Drake et al., 2019). After deforestation, the land is often permanently converted to agricultural land (cropland, grassland) or tree plantations (Depicker et al., 2021a). In some places, however, natural regeneration of the forest takes place (Masumbuko et al., 2012).

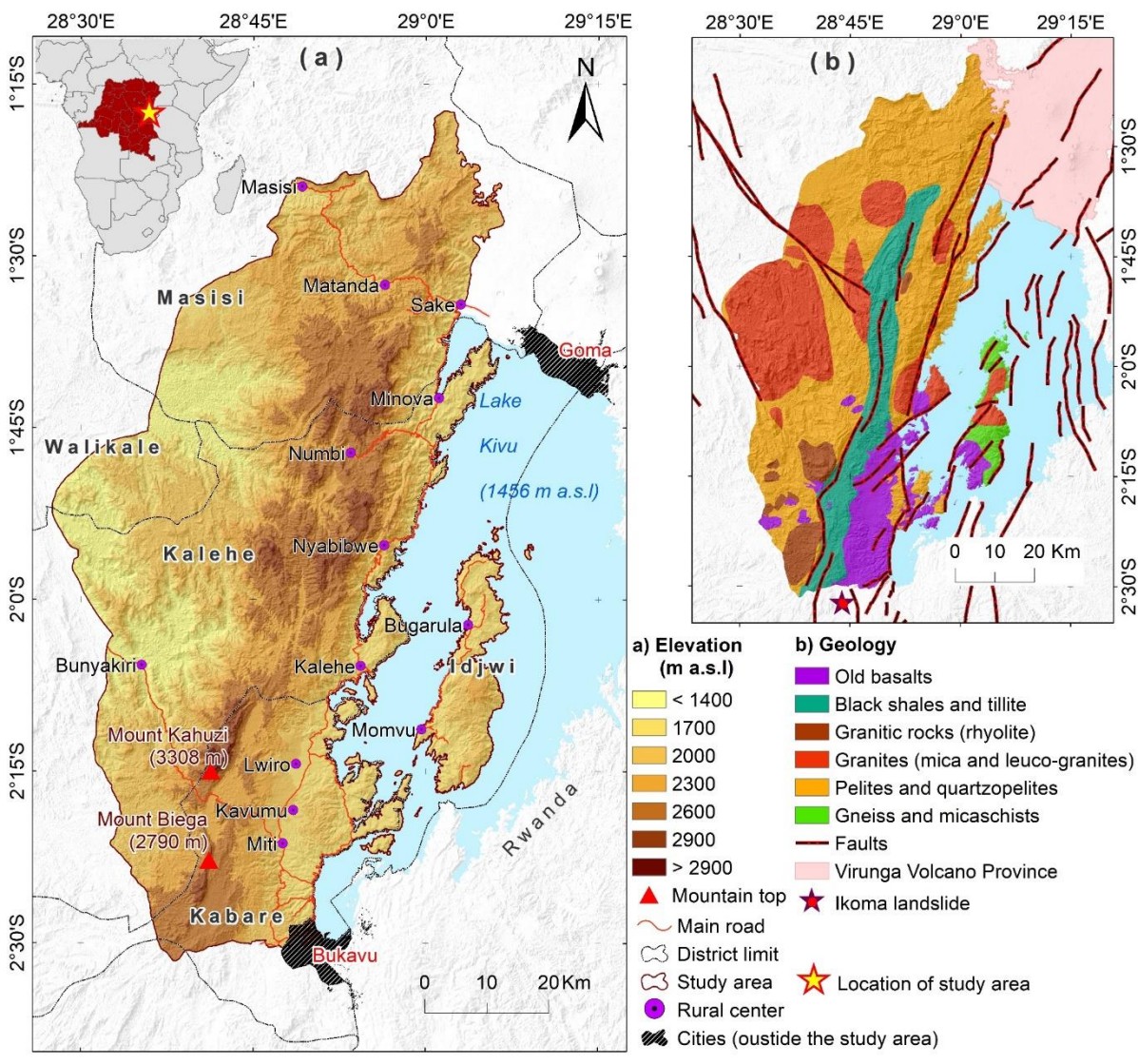

Figure 1: (a) Relief and (b) geology of the study area. The study area covers the districts of Kabare, Kalehe, Walikale, Masisi and Idjwi. Topography is derived from SRTM 1 arc second (https://lpdaac.usgs.gov/products/srtmgl1v003/). Lithology and fault maps are from Laghmouch et al. (2018).

The study area (~ 5,700 km²) is one of the most densely populated regions of the DR Congo with more than 200
inhabitants/km² living mainly from agriculture, mining and quarrying activities (Linard et al., 2012; Michellier et al., 2016; Trefon, 2016). This region plays a key role in the supply of food and charcoal to the smaller rural centers and to the cities of Goma and Bukavu. Over the last decades, the population in both cities increased from a few tens of thousands to more than one million inhabitants (Michellier et al., 2016). The population growth in the study area was partly caused by the influx of Rwandan refugees in 1994-1995, as well as the growing artisanal mining
industry that offers job opportunities (Bashwira et al., 2014; Van Acker, 2005; Butsic et al., 2015;). The road network is relatively limited (Fig. 1a). Most roads are dirt roads and are poorly maintained, and there are no built-up walls (concrete, gabions) to stabilize the cut slopes.

A first preliminary inventory of a few hundred landslides showed that the landslide processes are diverse and that their impacts on rural development can be high (Maki Mateso and Dewitte, 2014). The inventory over the North

Tanganyika-Kivu Rift region (hereafter called NTK Rift) of which our study area is a subregion was further expanded by Depicker et al. (2020) through the use of © Google Earth imagery. This inventory consisted of shallow and deep-seated landslides and did not distinguish for landslides susceptibility type. Depicker et al. (2020) showed that, in addition to slope angle, land cover is a key landslide predictor in the NTK Rift region. A more detailed investigation of the annual evolution of the forest cover over the last 20 years showed that deforestation increases the erosion due to the occurrence of new shallow landslide erosion 2-8 times during a period of approximately 15 years before it eventually falls back to a level similar to forest conditions (Depicker et al., 2021b). A catalogue of > 150 accurately dated landslide events, i.e. landslides that can be clearly associated with a common well-defined triggering rainfall event over the same area, was compiled for the NTK Rift for the last two decades. It allowed demonstrating the role of rainfall seasonality on the occurrence of new landslides (Monsieurs et al., 2018b; Dewitte et al., 2021). Some landslide events consist of clusters of several hundreds of shallow slope failures. The spatial extent of such clustered events can be larger than 10 km². A few events like these occur during each wet season (Depicker et al., 2020; Dewitte et al., 2021). They are commonly associated with particularly intense convective rainfall (Monsieurs et al., 2018b). None of the dated landslide events were triggered by earthquakes (Dewitte et al., 2021). This does not discard the role of earthquakes in triggering landslides in the region, but instead reminds us that the return period of earthquakes with a magnitude large enough to trigger slope instabilities can be much longer than a few decades (Delvaux et al., 2017). Their potential impact, rather localized compared to that of climatic drivers, can be inexistent during a narrow time window of observation (Dewitte et al., 2021; Depicker et al., 2021b).

Landslides can also occur due to rock weathering and regolith formation (Dille et al., 2019). In other words, the long-term evolution of these preconditioning drivers alone can explain that a slope can also fail without any apparent trigger. This implies that the many landslides that occur in isolation of other events must be interpreted with care in terms of origin. For these features, it is not clear from a visual analysis of the satellite images whether they can be directly linked to a direct trigger. In addition, many landslides occur in isolation along roads (Dewitte et al., 2021). Some of the larger, historical, landslides (i.e. landslides that do not appear active in our oldest source of information) are probably more than 10,000 years ago (Dewitte et al., 2021).

### 3 Material and methods

### 3.1 Landslide inventory

The landslide inventory is a significant update of the inventory compiled by Depicker et al. (2020) who used only © Google Earth imagery for mapping the features whatever their type, age, and rainfall, seismic or non-triggered origin as explained in Section 2. Since the focus of Depicker et al. (2020) was to study landslides over a much larger region than the one of the present research, their inventory was not only built on a limited search-time on our study area but, also, without any field survey. Moreover, in our research, we differentiated between types and timing of landsliding as these are key elements to consider to differentiate between natural and human-induced landslides (Hungr et al., 2014; Slide and Bogaard, 2016). In order to propose a landslide classification system adapted to our research objective, we strongly relied on three image products:

- a careful and detailed 3D (elevation exaggeration of 1) visual interpretation of © Google Earth images from 2005 to 2019, which provides a complete coverage of the region at very high spatial resolution (~0.5 m), often multi-temporal;

- the interpretation of two hillshade images derived from a TanDEM-X digital elevation model (DEM) provided at 5 m resolution and covering a large part of the region (see Albino et al., (2015) and Dewitte et al., (2021) for technical explanation on the production of the DEM). Despite some artefacts present in the DEM (Albino et al., 2015), this resolution allows to visually identify geomorphological features relevant for characterizing landslide processes (Dewitte et al., 2021). The hillshade images were produced with a sun elevation angle of 30° and sun azimuth angle of 315° and 45;

- the stereoscopic analysis of historical panchromatic photographs acquired during the 1955-1958 period at the scale ~1/50,000 (i.e. about 1 m spatial resolution on the ground); the photographs are conserved at the Royal Museum for Central Africa (RMCA, Belgium).

The historical aerial photographs allowed to differentiate between old deep-seated landslides (i.e. landslides with
175 an unknown time of origin and already present on the photographs) and recent deep-seated landslides that have occurred during the last 60 years (i.e. after the acquisition of the photographs). The aerial photographs were not used for mapping shallow landslides since this inventory would be biased. Indeed, the spatial resolution of the photographs is twice lower than that of the images in © Google Earth. Furthermore, the photographs provide a single temporal cover, whereas the multi-temporal © Google Earth images are composites over 13 years, i.e. the
180 age difference between the oldest and youngest image (e.g. Minova, Kalehe, Matanda in Fig. 1).

The estimation of the depth of a landslide is important when the role of LULC is to be considered; shallow landslides being much more sensitive to changes in vegetation characteristics than deep-seated landslides (Sidle and Bogaard, 2016). In the literature, a landslide is usually defined as shallow when the depth of its surface of rupture ranges between 2 to 5 m (Keefer, 1984; Bennett et al., 2016; Sidle and Bogaard, 2016). Here, landslides
with a depth < 5 m were considered as shallow. This criteria is based on the numerous field observations in the region that show that regolith can easily develop over a depth of several meters and that trees often show deep rooting systems. Following the approach of Depicker et al. (2020) and Dewitte et al., (2021), the distinction between deep-seated and shallow landslides was made by visually estimating the relative landslide depth from © Google Earth and the 5 m resolution TanDEM-X hillshade images. Extensive in situ-field observations of several
hundreds of recent landslides where then carried out to validate the assessment. The landslides occurring in mining and quarrying sites were all classified as mining landslides. A specific attention was also given to the landslides occurring along roads. Mining and road landslides are assumed to be related to important anthropogenic changes in the topography. Once they have occurred, field observations show that these landslides are commonly reworked and often further excavated. Therefore, for these two types of landslides, their depth was not assessed.

Six field surveys were conducted over the period 2016 to 2019 to validate the landslide inventory and get extra information on the landslide timing and their causes and triggers. Additional landslides identified only in the field were not considered in the analyses as they would bias the regional landslide distribution. The work was carried out by selecting representative areas with various types of landslides and areas with less or no landslides. These areas, that cover a total of ~20% of the region, were selected based on different landscape characteristics (lithology,

slope, LULC), while taking into account accessibility and safety issues that prevent to access many places (Jaillon, 2020). We also used information from media and grey literature (student theses, field reports from local research, and academic institutions and the civil protection).

The frequency of landslide surface area distributions were analyzed to check the completeness of the inventory and also enable comparison with other inventories in different environments. If the area frequency density can be properly fitted to an inverse gamma Γ distribution, it is considered representative of the study area (Malamud et al., 2004). A bad fit could suggest that the inventory is biased and/or incomplete. Indeed, the use of several data sources in the inventory could bias the distribution of landslides, especially bearing in mind the limitations related to the interpretation of satellite images (Guzzetti et al., 2012). We performed this analysis separately for five categories of the inventory considered together or in isolation: all landslides, old and recent deep-seated landslides, shallow landslides, mining landslides and road landslides (see Section 3.1). The analysis of the frequency area distributions for the different shallow landslide populations defined according to the LULC and its dynamics was also used to infer about differences in environmental characteristics and slope failure mechanisms (Malamud et al., 2004; Van Den Eeckhaut et al., 2007; Guns and Vanacker, 2014; Tanyaş et al., 2018). Box-plots complemented the shallow landslide area analysis.

The extent of the study area is relatively small when considering regional climatic characteristics and the time window of the shallow landslide inventory built from Google Earth imagery is limited to a few years. Therefore, the location and spatial properties (areal extent, number of occurrences) of a rainfall-triggered landslide event forming a cluster of slope failures depends strongly on the stochastic nature (location, extent and magnitude) of the triggering rainfall event and less on local terrain conditions. The consideration of all landslides of such a cluster could bias the analysis by giving an excessive weight to the local terrain conditions (Depicker et al., 2020). Thus, for the shallow landslide susceptibility analysis (see Section 3.2), we retained a maximum of 30 landslides per cluster, randomly sampled in order to strengthen the statistical analysis and avoid overfitting. The choice of this selection is also guided by the concern to have at least the minimum of data required for training and validating the susceptibility models (Depicker et al., 2020). For the inverse Γ analysis, those landslides selected per cluster and other isolate landslides are called distributions *minus event.*

**3.2 Multi-decadal forest dynamics**

In the study area, the agricultural land use is complex (multiple cropping, multi-layer farming) and highly dynamic due to crop rotations and associations, shifting cultivation, and the bimodal annual rainfall pattern (Heri-Kazi and Bielders, 2021b). A detailed regional land use mapping serving as input variable in our susceptibility for shallow landslides and their distribution analysis (see Section 2.3) is therefore not feasible (e.g. Jacobs et al., 2018) which is an approach that differs from what can commonly be done in non-tropical environments (e.g. Chen et al., 2019; Shu et al., 2019). However, the dynamics of the forest can be better constrained. Here, to complement the year to year analysis conducted by Depicker et al. (2021b; see Section 2.1) that focused on the impact of deforestation on shallow landslides over the last 20 years, we reconstructed the forest dynamics over the last ~60 years (Fig. 2). We used the 1 m resolution orthomosaic generated from the RMCA's aerial photographs of the years 1955-1958 (Depicker et al.,2021a); these photographs being the only existing pre-satellite era source of information. The forest areas were delineated visually. The 2016 forest cover was extracted from the continental ESA CCI land

cover model which is available at a 20 m resolution (ESA, 2016). This satellite-based product has an accuracy of roughly 86 % in the region and has demonstrated its relevance in another study on landslides (Depicker et al., 2021b). Note also that between 2016 and 2019, i.e. the date that corresponds to the most recent images in Google Earth used for the inventory, very little forest cover changes were observed.

Knowing that the natural vegetation of the study area is forest (Section 2), in 1955-58, 42 % of the territory was already deforested (Fig. 2a). From 1955-58 to 2016, the loss of forest continued, the forest cover decreasing from 58 % to 24 % of the study area. The area affected by the forest loss over the last 60 years is larger than the remaining permanent forest (Fig. 2b). The comparison of forest areas between 1955-58 and 2016 allows to consider four classes for the forest dynamics:

- permanent forest corresponds to forest areas that are present at both dates.
- the forest loss class corresponds to forests present in 1955-58 that have disappeared in 2016. Since it is impossible to identify for each portion of the landscape the exact cause of forest loss, this class contains a mix of various forest management practices and other causes of forest cut/removal.
- the forest gain class represents the new forest that has appeared since 1955-58. Similarly, the causes associated with the occurrence of new forest are not exactly known; afforestation and natural forest regeneration being certainly processes at play.
- permanent anthropogenic environment (e.g. cropland, grassland, built-up land) means that the landscape was not forested in both dates and it is assumed that it remained so during that period.

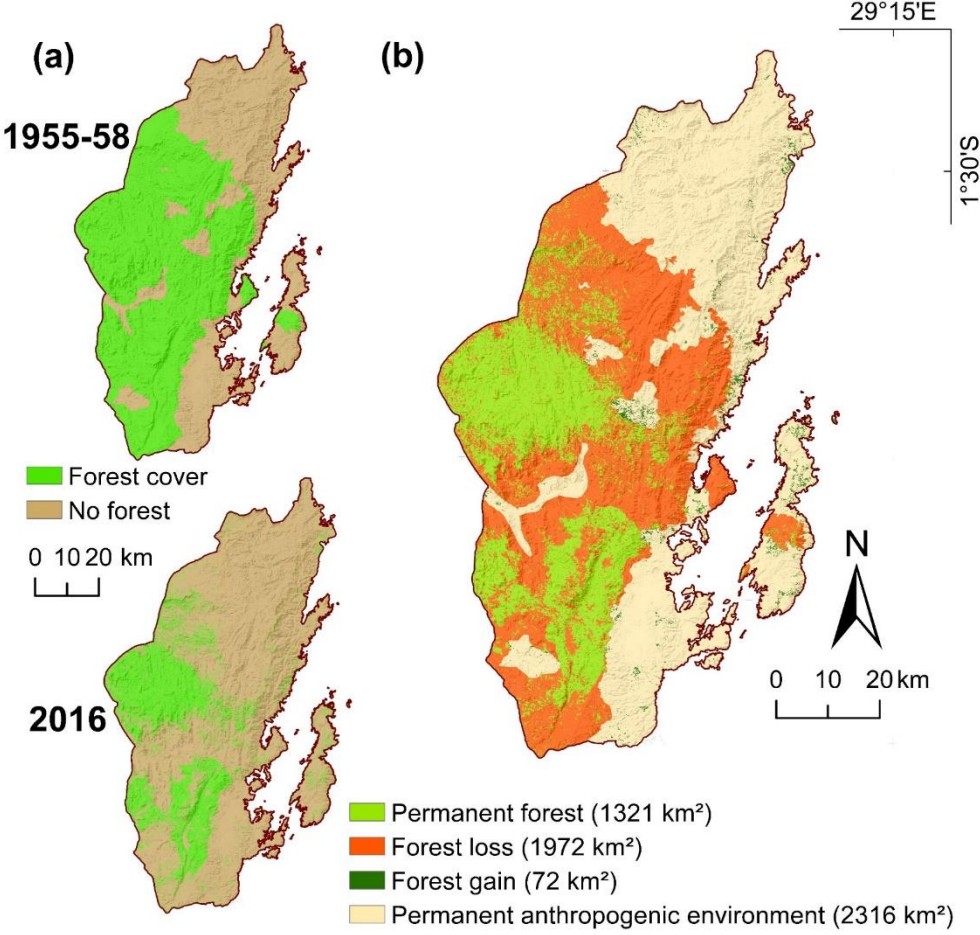

Figure 2: Forest cover dynamics over the last 60 years. (a) Forest cover in 1955-58 and 2016; (b) Areas of forest cover change between 1955-58 and 2016. Details for the images used in this figure are in Table 1.

**3.3 Landslide susceptibility and distribution analysis**

Landslide susceptibility approaches are commonly used to determine the factors that control the occurrence of landslides. There are numerous approaches which are more or less complex in terms of modelling implementation, data needs, and result interpretability (Reichenbach et al., 2018). Since our study does not aim to develop a new methodology nor to show the ability to use complex methods; we relied on a logistic regression approach (Hosmer and Lemeshow, 2000) to determine the predictor variables related to the occurrence of the different types of landslides. Logistic regression is a straightforward and relatively low-data demanding method that has been widely used (Reichenbach et al., 2018) and that allows a rather easy interpretation of the results (e.g. Jacobs et al., 2018; Depicker et al., 2020).

Frequency ratio (Lee and Pradhan, 2007) models were used as a more simple but complementary approach to better understand the role of each variable in the contribution of the landslide occurrence in terms of process characterization. For example, when slope angle is highlighted by a logistic regression model as a significant variable, we still remain unaware of the types of slopes that actually influence the occurrence of landslides.

The analysis was carried out according to the five categories of landslides defined in Section 3.1. The analysis was done at the scale of one point (pixel) per landslide to avoid spatial autocorrelation (e.g. Jacobs et al., 2018; Kubwimana et al., 2021). The point is manually positioned in the central region of the visually delineated landslide's source area to represent as close to reality as possible the conditions that caused its occurrence. In doing so we also avoid the selection of the highest point of the landslide that rarely corresponds to its initiation point (Dille et al., 2019). As stressed by Tanyaş et al., (2018), landslides growth with time. Therefore, considering one pixel per landslide instead of its whole source area allows to avoid a temporal-induced bias. The digital elevation model used for the analysis (see Table 1) is posterior to the occurrence of the old deep-seated landslides. Therefore, for deep-seated landslides, a point outside the source area where topography does not appear to have been disturbed by the instability is visually determined for the calculation of the slope associated with the landslide origin. Calculating the slope values at the level of the landslide source for this type of landslide would give values that are the consequences of landslides rather than the causes of their origin.

### 3.3.1 Predictor variables and landslide causes

The purpose of this research is to examine the predictor variables (See Figure S1 for the predictor variables not displayed in the main manuscript) that contribute to the susceptibility of the different landslide categories. As such we mainly investigate the causes of the landslides. Nevertheless, the predictors highlighted by the susceptibility analysis may also help to discuss triggering conditions since the tectonic, landscape and climate of a region are commonly interlinked (Whipple, 2009; Whittaker, 2012).

We used eight predictors that can be considered as natural factors that cause landslide occurrence (Table 1): elevation, slope angle, planar curvature, profile curvature, topographic wetness index (TWI), slope aspect, lithology, and distance to faults. Although these predictors are commonly used (Reichenbach et al., 2018), it is worth specifying that, here, elevation is used as proxy for local climatic conditions, namely orographic rainfall and the probability of convective rainfall/thunderstorms, as the resolution of regional-climate derived products is too low (at least 2.8 km) to accurately capture at the scale of our study area the effect of elevation on rainfall (Monsieurs et al., 2018a; Van de Walle et al., 2020). Distance to fault is used to determine the possible contribution of seismic activity in the occurrence of deep-seated landslides not only as a triggering factor (e.g. Keefer, 1984), but also as a measure of rock weathering (e.g. Vanmaercke et al., 2017). Using the fault pattern is the most appropriate option to tackle the seismic zonation context since the most detailed seismic hazard assessment for this part of the continent is at a spatial resolution of 2.2 km; i.e. at a resolution that is too coarse for our study (Delvaux et al., 2017).

Table 1. Landslide predictor variables used for the susceptibility and frequency ratio analyses and the ancillary data from which they are derived.

| Variable | Type | Source |
|---|---|---|
| - Elevation (m) | Continuous | |
| - Slope angle (°) | Continuous | |
| - Profile curvature (m$^{-1}$) | Continuous | |
| - Plan curvature (m$^{-1}$) | Continuous | Nasa Shuttle Radar Topography Mission (SRTM) Version 3.0 Global 1 arc second Data (Temporal Extent 2000-02-11 to 2000-02-21) |
| - Topographic wetness index | Continuous | |
| - Slope aspect (°) | Categorical | |
| • north | Dummy | |
| • northeast | Dummy | https://lpdaac.usgs.gov/products/srtm gl1v003/ |
| • east | Dummy | |
| • southeast | Dummy | |
| • south | Dummy | |
| • southwest | Dummy | |
| • west | Reference* | |
| • northwest | Dummy | |
| - Lithology | Categorical | |
| • Old basalts | Dummy | |
| • Black shales and tillite | Dummy | |
| • Granites (mica and leuco-granites) | Dummy | Geological map of the Kivu at scale 1/500,000 (Laghmouch et al., 2018) |
| • Granitic rocks (rhyolite) | Reference* | |
| • Pelites and quartzopelites | Dummy | |
| • Gneiss and micaschists | Dummy | |
| - Distance to faults (m) | Continuous | |
| Distance to roads (m) | Continuous | https://www.openstreetmap.org/history#map=9/-2.0475/28.5535 |
| - Forest dynamics between 1955-58 and 2016 | Categorical | Forest cover in 2016: (ESA, 2016: http://2016africalandcover20m.esrin.esa.int/viewer.php) |
| • Permanent forest | Reference* | |
| • Forest loss | Dummy | Forest cover in 1955-58: Historical aerial photographs and derived orthomosaics from RMCA (see Section 2.1) |
| • Forest gain | Dummy | |
| • Permanent anthropogenic environment | Dummy | |

* Each dummy variable is compared with the reference group.

Besides the natural factors, we identified two anthropogenic predictors (Table 1): forest dynamics and distance to roads. For the forest dynamics, we considered the four classes identified in Fig. 2. The main roads were retrieved from OpenStreetMap. Good knowledge of the study area and the analysis of the very high-resolution Google Earth images allowed us to verify the high accuracy of the road network proposed by OpenStreetMap. Using the

historical photographs, we observe that the main roads date back to the colonial times and that no major changes in the network have occurred over the last 60 years. The few recent landslides that are observed in the field along

these roads confirm the assumption that the direct impact of the main roads on the occurrence of recent landslides is currently limited. These landslides are clearly linked to the road cut topography, i.e. topographic conditions that cannot be constrained at the resolution of the SRTM elevation data (1" or roughly 30 m). They are often of very limited size, i.e. at a size that is too small to be features that can be identified in © Google Earth in a consistent manner. For our study, the distance to roads is taken as a proxy for human settlement, trail density, and intensity and diversity of agricultural practices. Since motorized transportation means are very limited in the region, the population growth, the expansion of villages and the agricultural activities are highly associated with the main road networks.

Prior to analysis, the non topographically-derived predictor variables were resampled at the resolution of the SRTM DEM data; a resolution that is commonly used in many susceptibility analyses (Reichenbach et al., 2018). Furthermore, in a region of Uganda located in a relative proximity, Jacobs et al. (2018) evidenced that the 1 arc second SRTM DEM clearly outperforms higher resolution products derived, in that specific case, from TanDEM-X. The association between the dependent variable and each predictor variable was tested using the Pearson $\chi^2$ test at a 95 % level of confidence (e.g. Van Den Eeckhaut et al., 2006). The predictors were tested for multicollinearity, variables with variance inflation factor (VIF) > 2 being excluded from the analysis (e.g. Van Den Eeckhaut et al., 2006). The flat areas (slope angle < 1°) that are spread across the region were not excluded from the analysis since their total extent is limited and their impact on the inflation of susceptibility model performance would be minor (Brenning, 2012 ).

For the analysis of deep-seated landslides, the predictor variables associated with anthropogenic activities were excluded. For the shallow landslides, the 'distance to faults' variable was also excluded. As explained earlier, the shallow landslide inventory represents a narrow time window of observation. As such, the spatial distribution of the shallow landslides could be biased by the stochastic pattern of the recent heavy rainfall events and anthropogenic disturbances rather than being the reflect of the longer-term impact of weathering conditions associated with seismicity (Depicker et al., 2021b).

### 3.3.2 Logistic regression

Logistic regression is used to describe the relationship between a binary dependent variable (the presence or absence of landslides) and one or more independent predictor variables (Hosmer and Lemeshow, 2000). Hence, the logistic regression does not only require landslide data, but also non-landslide data. We sampled this non-landslide data by generating a number of random points that is equal to the number of landslides in the inventory in order to avoid prevalence (Hosmer and Lemeshow, 2000). Non-landslide points were randomly generated outside a 40 m buffer zone around landslide areas. The basic equation for logistic regression is:

$$\log\left(\frac{P}{1-P}\right) = \alpha + \sum_{i=1}^{n} \beta_i X_i \tag{1}$$

where *P is* the likelihood of landslide occurrence and takes values between 0 and 1, α is the intercept of the model, $X_i$ represents *i*-th of *n* predictors, and $\beta_i$ the accompanying coefficient that has to be fitted to the data.

Calculations were performed in an RStudio environment version 1.4.1717 with LAND-SE software (Rossi and Reichenbach, 2016). In order to be considered in the final logistic regression equation, continuous variable coefficients needed to be significant at the 95 % level of confidence (e.g. Jacobs et al., 2018). For categorical variables, as soon as one dummy variable was significant, all other dummy variables were included in the model (e.g. Depicker et al., 2020). The quality of the models was judged by (i) the prediction rate (e.g. Depicker et al., 2020), (ii) a visual plausibility inspection of the susceptibility maps after reclassifying each map into four classes of increasing susceptibility that cover 40 %, 30 %, 20 %, and 10 % of the study area, and (iii) considering the area under the curve (AUC) of the receiver-operating-characteristics curve (ROC). The AUC values vary between 0 and 1 and can be interpreted as the model's capacity of differentiating between landslide and non-landslide locations. An AUC = 0.5 shows that the model performance is equivalent to random classification, while an AUC = 1 indicates a perfect classification (Hosmer and Lemeshow, 2000). Training and validation datasets were taken in the proportions of 70 % and 30 %, respectively (Broeckx et al., 2018; Fang et al., 2020).

We assessed the importance of each individual predictor for the logistic regression in two ways. First, we calculated the AUC for landslide susceptibility models that only relied on the considered predictor, to assess the extent to which this predictor can be used to differentiate between landslide and non-landslide locations. Although this is quite a straightforward approach that does not consider the possible interplay among predictor variables, this allow to have a first quantitative insight on the importance of each variable to the susceptibility models (Depicker et al., 2020). A second way to determine the impact of the predictors was the analysis of the odds ratio (OR). The OR of a predictor expresses how a change of a predictor value translates into an increase/decrease in the odds of landsliding, whereby the odds of landsliding is calculated as $\frac{P}{1-P}$ (see Eq. (1)). The $OR_i$ of predictor $i$ is calculated as:

$$OR_i = e^{\beta_i \delta_i},\qquad(2)$$

whereby $\beta_i$ is the coefficient of predictor $i$, and $\delta_i$ is the increase in predictor $i$. For continuous variables an arbitrary but realistic value for $\delta_i$ is chosen. For the dummy variables, $\delta_i$ equals 1. For the categorical variables, the OR for each dummy reflects an increase or decrease relative to the reference variable (Kleinbaum and Klein, 2010).

### 3.3.3 Frequency ratio

The frequency ratio model considers each landslide predictor variable individually and classifies its values into a set of bins to indicate for each bin of the predictor variable the probability of occurrence of a landslide (Lee and Pradhan, 2007; Lee et al., 2007; Kirschbaum et al., 2012). The frequency ratio is calculated as:

$$\mathrm{Fr}_{cb} = \frac{a_{cb}/a_T}{A_{cb}/A_T},\qquad(3)$$

where $\mathrm{Fr}_{cb}$ is the frequency ratio value for a bin $b = (1,2,\dots,n)$ of a predictor variable $c = (1,2,\dots,m)$, $a_{cb}$ is the cumulative landslide area within bin $b$ of predictor $c$, $a_T$ is the cumulative landslide area in the entire study area, $A_{cb}$ is the area attributed to bin $b$ of predictor $c$, and $A_T$ is the total extent of the study area.

## 4 Results

### 4.1 Landslide inventory

Overall, we mapped 2730 landslides (Fig. 3a; Table 2). The landslides are diverse in terms of size, age and type (Fig. 4). The inventoried landslides cover ~3 % of the study area. The largest landslide is an old and deep-seated complex movement (426 ha), while the smallest detected landslide is a shallow debris avalanche (16 m$^2$). The landslides are grouped into five categories (Fig. 3a; Table 2):

- old deep-seated landslides represent 45,5 % of the inventoried landslides and cover 93 % of the total
landslide affected area. Most of these landslides are of the rock slide type. Rock avalanches, although much less frequent, are also present. Rockfalls can be associated with the presence of the main scarps of these old landslides. However, they have not been considered in the inventory and the subsequent analysis;

- shallow landslides represent 40.4 % of inventoried landslides, but represent only 2.7 % of the total
affected area. Most of these landslides are of the debris avalanche type. These landslides are all recent and clearly associated with rainfall. The landslides clustered events all fall in this category;

- recent deep-seated landslides represent a small percentage of landslides (5.8 %) but cover an area (2.9 %) similar to shallow landslides. Most of the landslides are of the slide type. Their trigger, when identified, is associated with rainfall;

- mining landslides (that also include quarrying landslides) represent 5.6 % of the inventoried landslides and cover 1.2 % of the total landslide affected area;

- road landslides: the inventory shows that 115 landslides are located within 50 meters of roads. 60 of these landslides are shallow, 13 recent and deep-seated, 35 old and deep-seated, and 7 are mining landslides. Only the shallow and recent deep-seated landslides were classified as road landslides; i.e. a total of 73
landslides The old deep-seated landslides located close to roads were retained in the old deep-seated landslide category because their timing is likely to precede road construction. The mining landslides were also retained in their respective category.

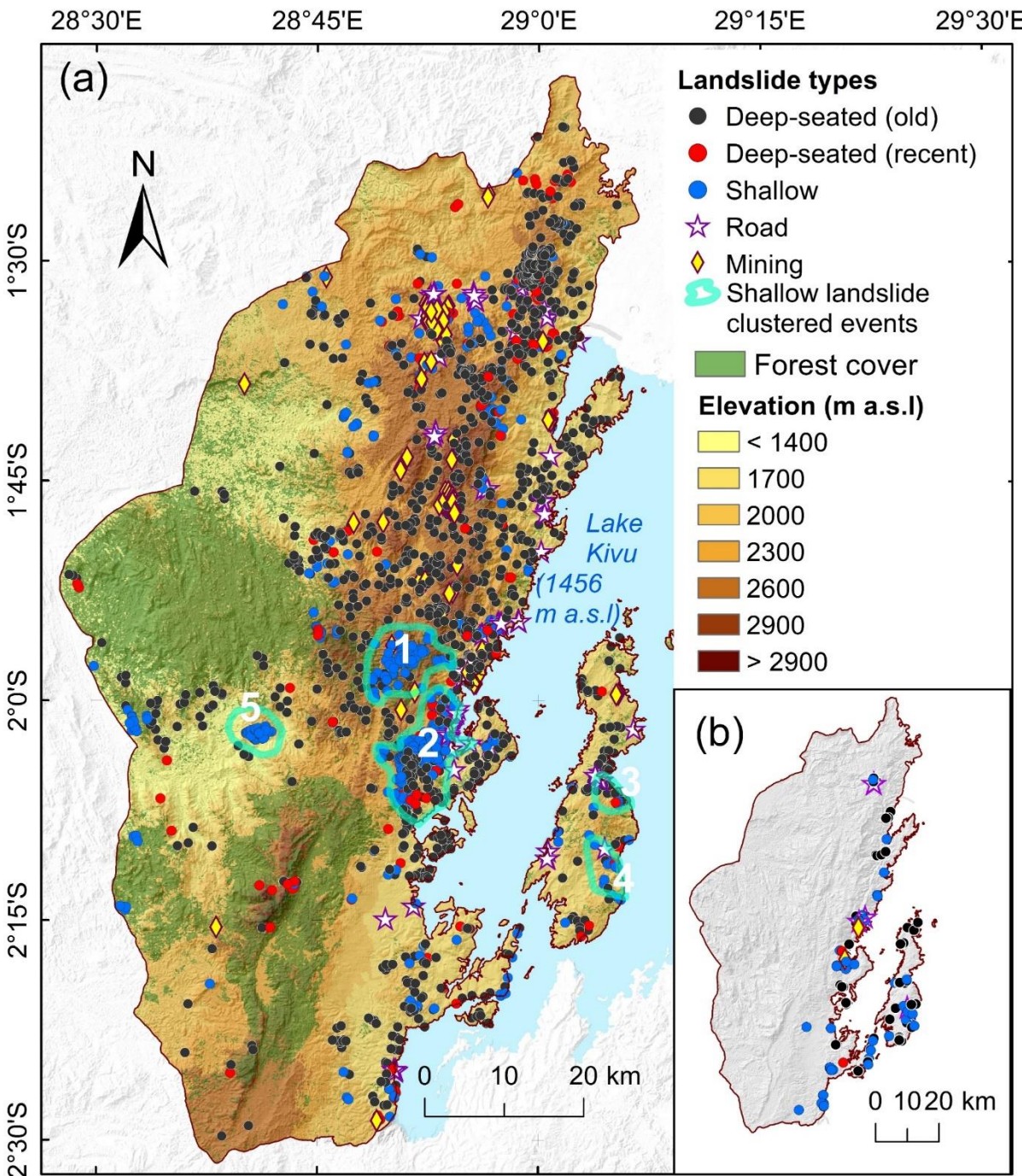

Figure 3: (a) Landslide inventory obtained from the image analysis and extent of the forest cover in 2016 (after ESA, 2016). Numbers represent clusters of shallow landslides that are associated with heavy rainfall events dated in ascending order from oldest to most recent. (b) Additional landslides identified only in the field.

We identified several shallow landslides clustered events. One of the events is related to the Kalehe rainstorm of October 25, 2014 (Fig. 3a: event 2; Fig. 4a) reported by Maki Mateso and Dewitte (2014). This rainfall triggered 634 shallow landslides, 346 of them being connected to talwegs and providing materials to 17 debris flows. Ten debris flows were particularly destructive and deadly when they reached villages on the shores of Lake Kivu (Maki Mateso and Dewitte, 2014). In this area, 14 shallow landslides present on © Google Earth images before this event

were reactivated. Field observations and interviews with local populations confirmed that the shallow landslides that are not associated with these clustered events are also rainfall-triggered.

Table 2: Typology, size properties, and identification methods of the inventoried landslides (LS). The percentages of landslides linked to the TanDEM-X hillshade images (% of LS in TanDEM-X) represent landslides that could not be very well identified in © Google Earth alone.

| Landslide type | Number of LS | % of LS | % of LS area | Max area (ha) | Min area (m²) | Average area (ha) | Standard deviation (ha) | % of LS in ©Google Earth | % of LS in TanDEM-X |
|---|---|---|---|---|---|---|---|---|---|
| Deep-seated (old) | 1243 | 45.5 | 93.0 | 426.4 | 604 | 12.6 | 26.8 | 94.9 | 5.1 |
| Deep-seated (recent) | 159 | 5.8 | 2.9 | 28.9 | 210 | 3.1 | 5.4 | 97.5 | 2.5 |
| Shallow | 1103 | 40.4 | 2.7 | 53.8 | 16 | 0.4 | 2.4 | 100 | 0 |
| Mining | 152 | 5.6 | 1.2 | 13.4 | 99 | 1.4 | 1.9 | 100 | 0 |
| Road | 73 | 2.7 | 0.1 | 2.0 | 149 | 0.3 | 0.3 | 100 | 0 |
| All landslides | 2730 | 100 | 100 | | | 6.2 | | 97.5 | 2.5 |

Table 3: Field-based validation of the landslides (LS) inventoried from the image analysis. True Positive (TP) = landslides that were mapped in the images and validated in the field. False Positive (FP) = landslides that were mapped in the images but not validated in the field. False Negative (FN) = landslides that were identified solely in the field. Precision = TP / (TP+TN)

| Landslide type | Number of LS mapped in the images and checked in the field | TP | FP | FN | Precision (%) | Total number of LS viewed in the field |
|---|---|---|---|---|---|---|
| Deep-seated (old) | 248 | 239 | 9 | 60 | 96 | 308 |
| Deep-seated (recent) | 47 | 44 | 3 | 4 | 94 | 51 |
| Shallow | 426 | 420 | 6 | 55 | 99 | 481 |
| Mining | 15 | 9 | 6 | 2 | 60 | 17 |
| Road | 50 | 45 | 5 | 5 | 90 | 55 |
| Total | 786 | 757 | 29 | 126 | 96 | 912 |

Landslide mapping was largely done using © Google Earth; the TanDEM-X hillshades being useful to confirm the identification of about one fifth of the old deep-seated landslides (Table 2). Fieldwork carried out to validate 786 landslides (25% of the inventory) showed that they were identified with a precision (TP/(TP + FP) of 96 % (Table 3). Old deep-seated landslides and shallow landslides were mapped with the highest precision. Mining landslides were mapped with a lower precision due to the difficulty of differentiating between landslide processes and anthropogenic soil disturbance in © Google Earth imagery. The field validation allowed to also map an extra 126 landslides (Fig. 3b) that could only be identified in the field (Table 3). For the old deep-seated landslides, this represents an extra 24% of observations (Table 3: see column FN). Nevertheless, landslides identified only in the field were not considered in the analysis to avoid biases due to overrepresentation.

Each debris flow is connected to up to hundreds of shallow landslides that act as source areas. A clear distinction was made between these sources and the debris flow path and deposition areas (Fig. 4a). Out of a total of the 184 debris flows identified from the images, 90 with a length-to-width ratio > 50 were excluded from the analysis since they show greater similarities to debris-rich floods than to the other landslides present in the region (Malamud et al., 2004). Nevertheless, the shallow landslides acting as source areas were kept in the analysis. Also, 22 very large, old, deep-seated landslides were excluded from the analysis because they have complex main scarps where it is difficult to determine the pixels that best represent the natural conditions of occurrence. Overall, from the 2730 landslides identified from the images, 2618 landslides were used for the subsequent analysis.

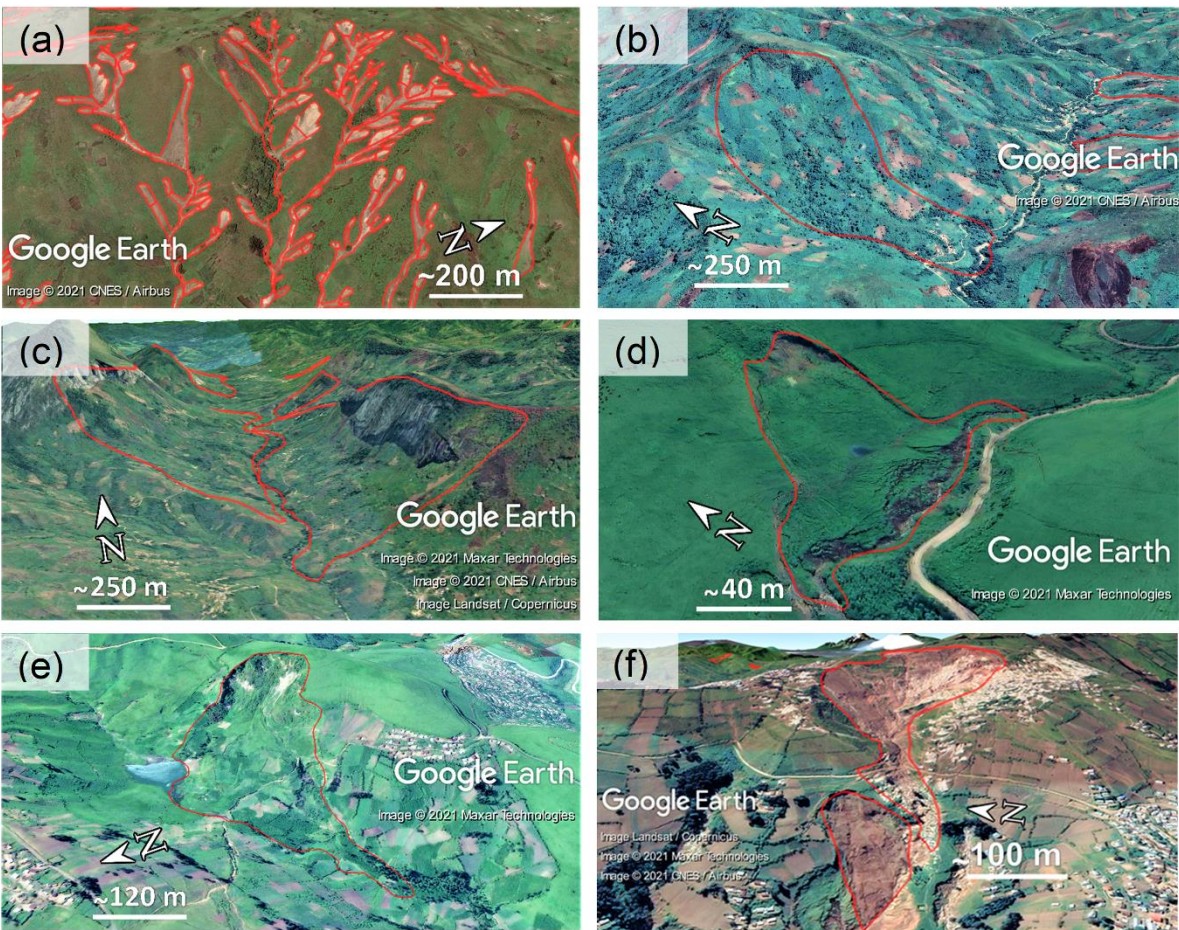

Figure 4: Examples of landslide types (according to Varnes' new classification – Hungr et al., 2014). (a) Cluster of recent debris avalanches and debris flows triggered during an intense rainfall event (25/10/2014) in the vicinity of Kalehe (-2.041°S, 28.874°E); the image illustrates a part of the landslides clustered event 2 shown on Fig. 3a). The source areas of these shallow landslides are identified. (b) Old earthflow (-2.053°S, 28.660°E). (c) Old rock slides/rock avalanches/ with path-dependent rockfalls (-2.007°S, 28.708°E). (d) Recent deep-seated rotational slide that occurred in 2002 (-1.530°S, 28.708°E). (e) Recent deep planar slide that occurred in 1994 and created a dammed lake (-1.521°S, 28.977°E). (f) Recent slides, flows and avalanches associated with mining activities that occurred from 2013 onwards (-1.563°S, 28.885°E).

Except for the recent deep-seated and mining landslides, the inverse gamma Γ distribution fits well the distributions for the other categories of the inventory, (Fig. 5a,c); which supports their use for further susceptibility analysis. The Wilcoxon rank comparison test confirms significant statistical differences (p-value < 0.05) among the area distributions (Fig. 5b).

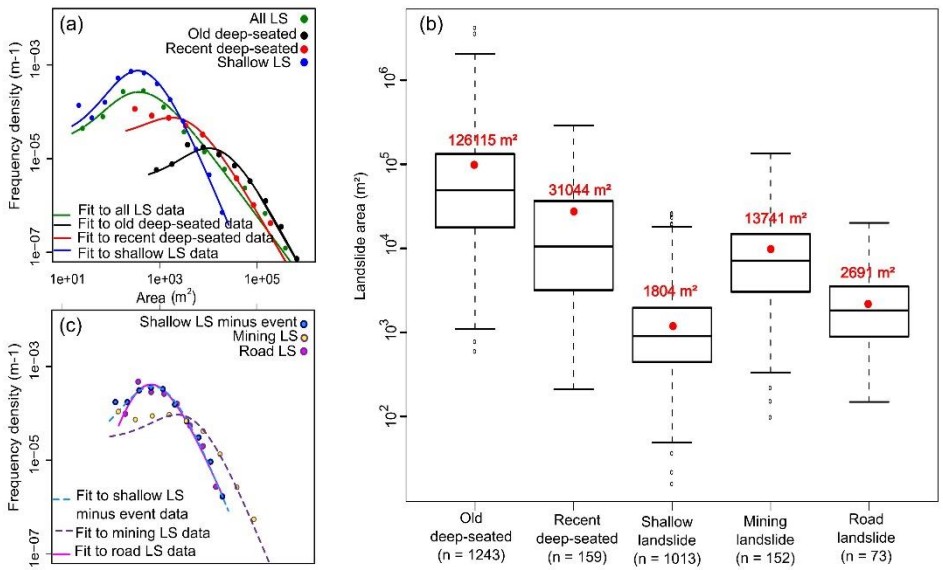

Figure 5: Landslide (LS) area characteristics. (a, c) Landslide frequency-area distributions for each landslide category. (b) Boxplots showing the distribution of landslide area for each landslide category. Boxplots show the lower and upper quartiles and median. The whiskers of each box represent 1.5 times the interquartile range. The average area of the landslides (red dots) is provided for each boxplot and the outliers beyond whiskers are shown as dots. The number of landslides in each category is shown in brackets.

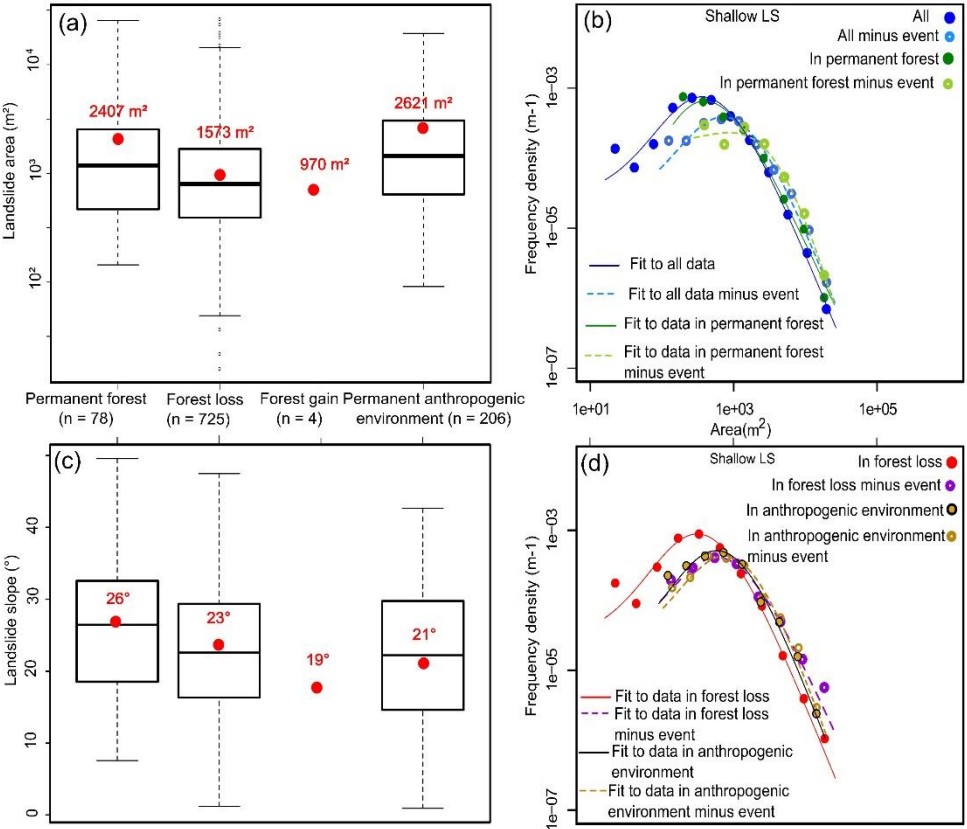

Figure 6: Shallow landslide characteristics and forest cover dynamics. (a, c) Boxplots showing the distribution of landslide area and landslide slope, respectively, for each land cover class. A detailed description of boxplots is provided in Figure 5. (b, d) Shallow landslide frequency-area distributions for each land cover class.

A majority (72 %) of the shallow landslides is found in areas of forest loss (Fig. 6). The landslides in the permanent anthropogenic environment have the largest mean area, followed by the landslides in permanent forest, and the landslides in areas of forest loss. In forest gain zones, landslides are on average the smallest. The Wilcoxon rank comparison test confirms significant statistical differences (p-value < 0.05) among the landslide area distributions. The same differences are also confirmed for the landslide slope distribution (Fig. 6c). In permanent forest areas, shallow landslides occur on steeper slopes compared to shallow landslides in anthropogenic environments (Fig. 6c). The analysis of the completeness of the inventory (Fig. 6b,d) shows that an acceptable distribution emerges for each category of shallow landslides except for the landslide inventory in *permanent forest minus event* (Fig. 6b).

**4.2 Landslide susceptibility and distribution analysis**

The Pearson $\chi^2$ tests confirm the association between the dependent variable and each predictor variable at a 95 % level of confidence. There was no multicollinearity between the predictors (VIF < 2) retained for this study.

Depicker et al. (2020) assessed the impacts of the size of the landslide training dataset to calibrate a landslide susceptibility model. They showed that the quality of a susceptibility assessment is questionable if the number of

landslides is too small. In view of the low number of recent deep-seated, mining, and road landslides in the present

study (Table 3), we did not calibrate susceptibility models from these three types of landslides. Instead, we tested

490 these inventories against the two susceptibility models computed from the shallow and/or old deep-seated landslide

datasets, from which we could derive prediction rates (Fig. 7).

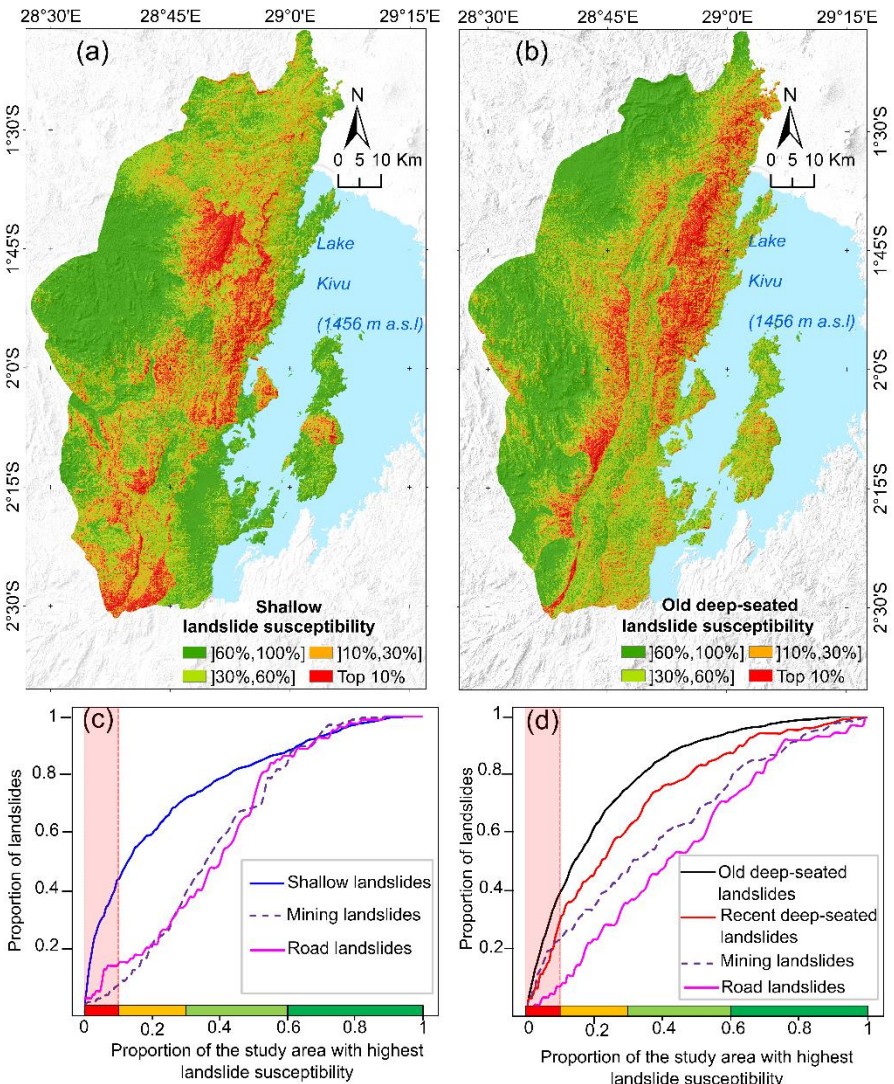

Figure 7: Landslide susceptibility models and prediction rates. (a) shallow landslides (AUC: 0.78); (b) old deep-

seated landslides (AUC: 0.82); (c) prediction rate curves for shallow, mining, and road landslides; (d) prediction

495 rate curves for old deep-seated, recent deep-seated, mining and road landslides. The red highlight (c, d) represents

the 10 % of the region with the highest landslide susceptibility values.

The univariate AUC values are all above 0.5 (Table, 4). All predictors considered for both categories of landslides

where thus considered in the multivariate logistic regression models. The two susceptibility models of shallow and

500 old deep-seated landslides show similar AUC and prediction rates (Fig. 7). At first sight, both models have spatial

similarities of high susceptibility on the eastern part of the region; while the entire western part is less susceptible

(Fig. 7a,b). However, when we go into detail, the spatial patterns of the susceptibility values of the two models are

quite different as it reflects the differences in the importance of the predictors included in the assessment (Table 4, Table 5).

Table 4: Relative importance of the predictors of the logistic regression models for shallow and old deep-seated landslides based on $AUC_i$ (ranked in descending order).

| Shallow landslides | | Old deep-seated landslides | |
|---|---|---|---|
| Predictor | $AUC_i$ | Predictor | $AUC_i$ |
| Forest loss | 63 | Profile curvature. | 65.7 |
| Elevation | 61.5 | Elevation | 65.3 |
| Slope angle | 60.1 | Distance to faults | 64.2 |
| Distance to roads | 59.7 | Slope angle | 64 |
| Pelites and quartzopelites | 58.9 | TWI | 63.8 |
| Permanent anthropogenic environment | 55.9 | Plan curvature. | 59.5 |
| TWI | 55.3 | Pelites and quartzopelites | 54.5 |
| Plan curvature. | 53.1 | South | 52.4 |
| East | 52.2 | North-east | 52 |
| South-east | 52.2 | North | 51.6 |
| Black shales and tillite | 51.8 | East | 51.1 |
| Old basalts | 51.8 | South-east | 50.7 |
| North | 51.8 | Granites (mica and leuco-granites) | 50.7 |
| Granites (mica and leuco-granites) | 50.8 | Old basalts | 50.5 |
| South-west | 50.7 | Gneiss and micaschists | 50.5 |
| Gneiss and micaschists | 50.6 | South-west | 50.3 |
| South | 50.6 | Black shales and tillite | 50.3 |
| Profile curvature | 50.4 | North-west | 50.2 |
| North-east | 50.4 | | |
| North-west | 50.1 | | |
| Forest gain | * | | |

* Only four landslides are present in this category.

Table 5. Results of the logistic regression models for shallow landslides and old deep-seated landslides.

| | | Shallow landslides | | Old deep-seated landslides | | Step |
|---|---|---|---|---|---|---|
| AUC | | 0.78 | | 0.82 | | |
| Predictor variable | | LR coef. | Odds ratio | LR coef. | Odds ratio | $\delta_i$ |
| (Intercept) | | -3.560 *** | | -1.661 *** | | |
| Elevation | | 0.001 *** | 1.857 | 0.002 *** | 2.535 | 500 |
| Slope aspect | Northwest | 0.842 * | 2.321 | -0.366 | 0.694 | 1 |

| | | Coef. | | OR | Coef. | | OR | |
|---|---|---|---|---|---|---|---|---|
| | West | Ref. | - | | Ref. | - | | |
| | Southwest | 0.674 | * | 1.962 | -0.232 | | 0.793 | 1 |
| | South | 0.630 | * | 1.878 | 0.032 | | 1.033 | 1 |
| | Southeast | 0.599 | | 1.820 | -0.345 | | 0.708 | 1 |
| | East | 0.513 | | 1.670 | -0.578 | ** | 0.561 | 1 |
| | Northeast | 0.622 | | 1.863 | -0.897 | *** | 0.408 | 1 |
| | North | 0.481 | | 1.618 | -0.831 | *** | 0.436 | 1 |
| Plan curvature | | -0.272 | * | 0.580 | 0.166 | *** | 1.394 | 2 |
| Profile curvature | | -0.190 | | 0.999 | -0.463 | *** | 0.998 | 0.005 |
| Slope angle | | 0.050 | *** | 1.649 | 0.033 | *** | 1.391 | 10 |
| Topographic wetness index | | 0.093 | | 1.000 | -0.281 | *** | 1.000 | 0.001 |
| Lithology | Old basalts | -0.753 | - | 0.471 | 0.201 | | 1.223 | 1 |
| | Black shales and tillite | -1.207 | *** | 0.299 | -1.358 | *** | 0.257 | 1 |
| | Granite coarse grain | -17.026 | - | 0.000 | -2.126 | *** | 0.119 | 1 |
| | Granitic rocks (rhyolite) | Ref. | | | Ref. | | | |
| | Pelites and quartzopelites | -1.274 | *** | 0.280 | 0.155 | | 1.168 | 1 |
| | Gneiss and micaschists | 0.506 | | 1.659 | -0.468 | | 0.626 | 1 |
| Distance to roads | | 0.000 | *** | 0.931 | no | | - | 500 |
| Distance to faults | | no | | - | 0.000 | *** | 0.914 | 500 |
| Forest cover dynamics | Permanent forest | Ref. | - | | no | | - | |
| | Forest loss | 0.922 | *** | 2.514 | | | | 1 |
| | Gain forest | no | | - | no | | - | |
| | Permanent anthropogenic environment | -0.159 | | 0.853 | no | | - | 1 |

No = variable not included in the logistic regression model

Ref. = reference category of the dummy variable

Coefficient included in the logistic regression model = *p-value < 0.05, ** p-value < 0.01, *** p-value < 0.001

Forest loss has a large influence on the occurrence of shallow landslides as deforestation increases the odds of

landsliding by a factor 2.5 (Table 4, Table 5). However, anthropogenic environments appear to be less landslide-prone than permanent forest. Elevation and slope angle are similarly important for the prediction of both types of landslides (Table 4) but have a slightly larger impact on the odds of deep-seated landsliding that on the odds of shallow landsliding (Table 5). Slope aspect has a greater impact on the occurrence of shallow landslides than for old deep-seated landslides. It appears that the plan curvature reduces the occurrence of shallow landslides while it

favours the occurrence of old deep-seated landslides. The effect of lithology is also different for shallow and deep-seated landslides. For shallow landslides, the gneiss and micaschists are most landslide-prone and the lowest susceptibility is associated with black shales, tillite and old basalts. For deep-seated landslides, black shales, tillite and old basalts favour landslides while gneiss and micaschists do not. 'Distance to roads' and 'distance to faults' have a significant but rather limited impact on shallow and old deep-seated landslides, respectively.

Mining and road landslides are poorly predicted using the shallow landslide model (Fig. 7c). The prediction of road and mining landslides using the deep-seated model is also poor, although less problematic for the mining landslides (Fig. 7d). Recent deep-seated landslides are reasonably well predicted using the old deep-seated landslide model, which validates to some extend the multi-temporal predicting performance of the assessment.

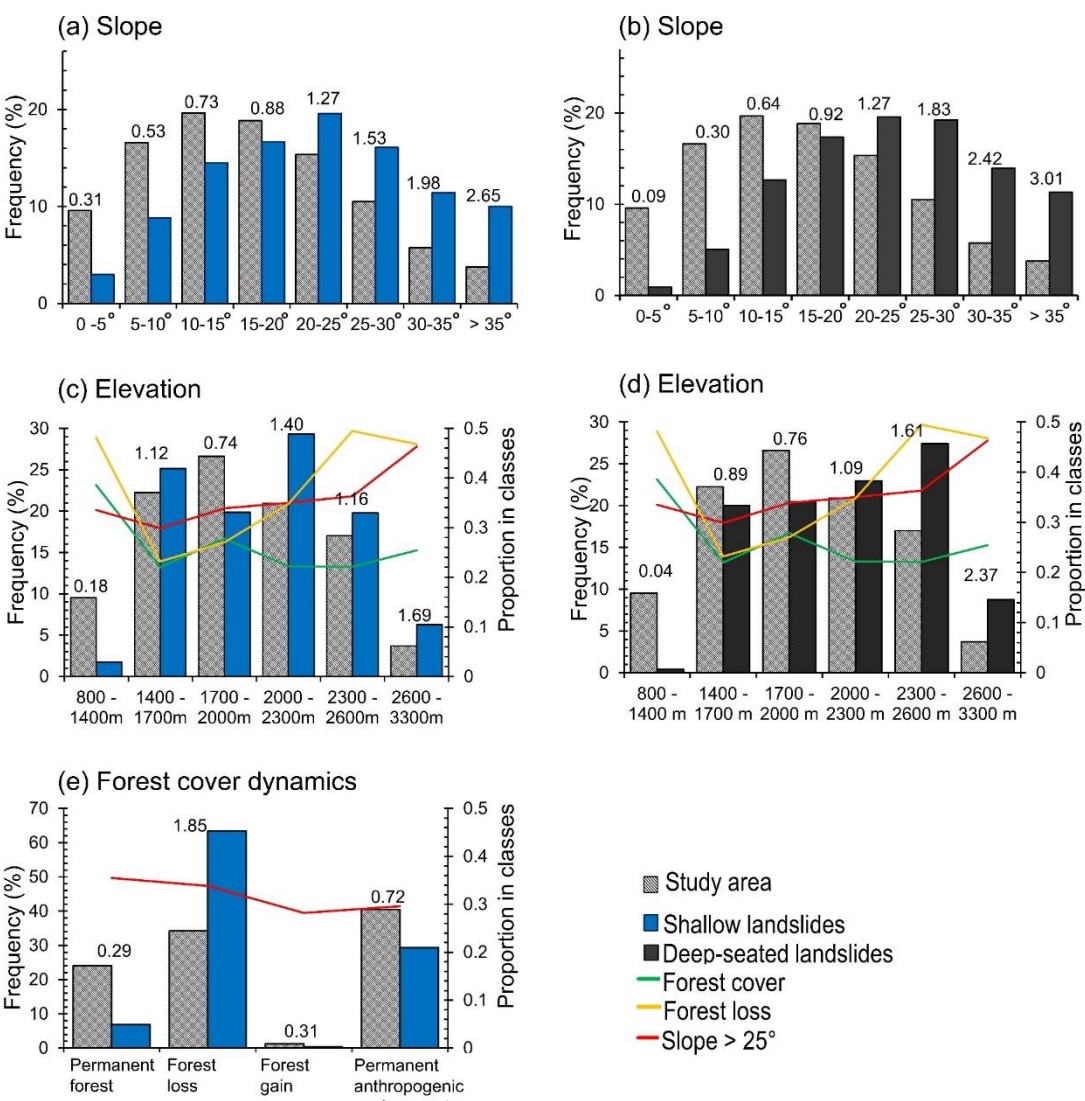


Figure 8: Frequency distribution for shallow and deep-seated landslides in function of key predictor variables. Figures c, d, and e allow a multivariate comparison of the predictors. The corresponding frequency ratio is shown for each class above the vertical bars. The green, orange and red curves indicate the proportion of forest cover, forest loss and slope > 25°, respectively, in the different classes of the predictor variables.


Slope angle is an important driver for shallow and old deep-seated landslides (Fig. 8a,b). Both types of landslides are favoured by slopes angles > 20-25°. We observe a trend in the landscape of increasing slopes and forest loss and decreasing forest cover with increasing elevation (Fig. 8c). The decrease in forest cover at high elevations is also associated with a natural change of the vegetation: bamboo vegetation is found at 2300-2600 m asl and

subalpine vegetation such as ferns occur at 2400-3300 m asl (Mokoso et al., 2013; Cirimwami et al., 2019). At higher elevations (> 2000 m), shallow landslides occur more frequently, and this can probably be explained by a cumulative effect of forest loss, steeper slopes and increased orographic rainfall associated with these elevations (Fig. 8c). The positive frequency ratio in the 1400-1700 m elevation class is related to the area of permanent anthropogenic environment. This zone is characterized by low forest cover and relatively low slopes (Fig. 8c).

Deep-seated landslides are also favoured by steeper slopes and higher elevations. Regarding the dynamics of forest cover (Fig. 8e), the occurrence of shallow landslides is favoured in the deforested areas.

## 5 Discussion

### 5.1 Landslide types and completeness of the inventory

Despite its high precision, and the fact that with more than 2700 mapped landslides we have identified more than three times as many features as in the inventory of Depicker et al. (2020), we are aware that the dataset is still incomplete. This is particularly the case for the shallow landslides because their inventory covers a maximum period of 13 years. Furthermore, their scars can quickly be altered by natural vegetation regrowth, land reclamation and erosion (Malamud et al., 2004; Van Den Eeckhaut et al., 2007; Kubwimana et al., 2021; Dewitte et al., 2022);

although, here, since we have used several image covers from Google Earth, this issue should be nuanced. In addition, small landslides frequently happen unnoticed at the resolution of the satellite images (Guzzetti et al., 2012). Finally, field validation showed that a significant proportion of old deep-seated landslides can be missed from image analysis (Table 3). This is because recognizing old deep-seated landslides may not be easy, particularly in forest areas (Malamud et al., 2004). While building the inventory, we remained conservative and mapped only

the features for which we had high confidence. As the protocol for landslide identification over the whole region was uniform and the number of identified landslides relatively important, we trust that the inventory is reliable and representative enough for the analysis.

The frequency area distributions of all landslides categories (Fig. 5a,c), with the exception of recent deep-seated and mining landslides, are similar to what has been observed in other parts of the world (e.g., Malamud et al.,

2004; Guns & Vanacker, 2014; Jacobs et al., 2017 ). For the recent deep-seated landslides, an overrepresentation is noticed at the level of the smallest landslides and the rollover is absent. Since the spectral signature of these landslides is pronounced, we cannot invoke here a problem of subjectivity in the mapping. Additionally, we can give a high trust in the completeness of the inventory as evidenced by field validation that showed that almost no landslides were missed (Table 3). Therefore, we posit that this divergence in size is related to a lower influence of

successive slope failure in the increase of landslide area through time; in other words, recent landslides did not have the time to growth (Tanyaş et al., 2018). This process of successive failures has been well documented for the Ikoma landslide, south of Bukavu (Figure 1b; Dille et al., 2019). The distribution of the mining landslides is irregular and different from what is typically observed, with a rollover that is flattened and a sudden increase in the frequency of the smallest slope failures. Similarly, to the inventory of the recent deep-seated landslides, the

completeness and the reliability of the mapped features cannot be much questioned. We suggest that this unusual area distribution is the result of the human-induced alteration of the environmental conditions (see Section 4.4).

To our knowledge, there are apparently no similar studies that have been carried out on artificial mining slopes. Further investigations on other cases would be needed to verify our hypothesis.

The presence of a rollover in the frequency-area distribution of the shallow landslides in the anthropogenic environment (Fig. 6b,d), compared to tropical mountains (Guns and Vanacker, 2014), is in opposition to what we could have expected considering the study by Van Den Eeckhaut et al. (2007). This study was also conducted in a densely populated rural environment and also relied on an inventory that is not associated with one single landslide triggering event. They did not find a positive power-law relation for the smaller landslides which is separated from the larger landslides by a rollover. This difference probably lies in the fact that our study area is much more landslide-prone. The research by Van Den Eeckhaut et al. (2007) was indeed carried out in a hilly region of Belgium where the temperate climate is much less favourable to the yearly occurrence of shallow landslides. Furthermore, the fact that our inventory covers a smaller time period than that of Van Den Eeckhaut et al. (2007), that our region is not altered by mechanized farming, and that human activities such as works associated with building and road construction and drainage systems are much less present, i.e. factors that are highlighted as causes of landslides in Belgium, are issues that can also be invoked to explain this divergence in the frequency area distribution of shallow landslides.

Under permanent forest, we do not observe a rollover point in the shallow landslide distribution, (Fig. 6b). We hypothesize that the smallest landslides may be hidden under the canopy and therefore less visible on satellite images. A second explanation is that the presence of trees and their roots increases slope stability and therefore the minimal critical area for landsliding (Milledge et al., 2014).

**5.2 Drivers of deep-seated landslides**

The old deep-seated landslide susceptibility model is the first model proposed for the region that focuses only on deep-seated processes. The model shows a good quantitative prediction performance, both in terms of AUC and prediction rate. The model shows that terrain morphology and seismic activity seem to play a dominant role in deep-seated landslide distribution in the study area. The frequency ratio analysis (Fig. 8b,d) further supports this as it highlights the association of landslides with steep slopes and higher elevations, i.e. in topographic contexts nearer to the ridge crests that are known to amplify seismic shaking (Meunier et al., 2008). The role of elevation as a driver of more humid conditions should, however, not be ignored as rainfall is also known to trigger deep-seated landslides (LaHusen et al., 2020). Also, the role of the long-term weathering of the landscape and the occurrence of non-triggered landslides should not be underestimated (Dille et al., 2019). Lithology is of lesser importance in our study area; which is in agreement with the findings of Depicker et al. (2021b) that show that at the regional scale, the various lithologies have similar rock strength properties. However, with a better defined lithological information, local specificities would certainly appear (Kubwimana et al., 2021). As we also show that the topography and the presence of faults play a role, it is another factor that can explain that the influence of lithology is somehow limited.

The lower prediction rate of the recent deep-seated landslides using the old deep-seated landslide model could be related to the fact that the observations are made on a period that is too short to apprehend the full panel of

environmental conditions that led to old deep-seated landslides. For example, no earthquake-induced recent deep-seated landslides were observed (Dewitte et al., 2021), whereas seismicity is an important component of the old deep-seated landslide model. In addition, the climatic and seismic conditions have evolved over the past tens of thousands of years (Felton et al., 2007; Wassmer et al., 2013; Ross et al., 2014; Smets et al., 2016). For example, the region experienced an abrupt shift from drier conditions to more humid conditions around 13,000 BP (Felton et al., 2007; Wassmer et al., 2013). In addition, about 10,000 BP, Lake Kivu water highstands were ~100 m above the current level, which could have triggered few large landslides (Ross et al., 2014; Dewitte et al., 2021). This change in the lake level was not only due to a shift in the climatic conditions but also to the formation of the Virunga Volcano Province that created a dam on the upstream part of the Rift basin that used to drain northwards (Figure 1b; Haberyan and Hecky, 1987). During that period of volcano formation, the regional geodynamics and the seismicity pattern were different (Smets et al., 2016). Hence a large part of the old deep-seated landslides may have been triggered under different conditions.

Old and recent deep-seated landslides differ also in terms of size (Fig. 4). There have not been any major events during the past 60 years that caused large landslides comparable to the largest old deep-seated landslides (of area $10^6\,\mathrm{m}^2$). We identify five possible factors to explain this difference. First, our window of observation is too narrow to apprehend the impact of forcing events of high-magnitude such as large earthquakes (Marc et al., 2019). Second, the past environmental conditions may have been more favourable to large slope failures. A third factor explaining the size difference between old and recent deep-seated processes is that larger landslides are less frequent but have a longer-lived morphology legacy; therefore, smaller old deep-seated landslides may no longer be visible. The fourth factor is that old landslides have a size that is the legacy of a history of phases of slope deformation, and not one single slope failure (Tanyaş et al., 2018) . Fifth, amalgamation must not be excluded (Marc and Hovius, 2015), especially for the eldest features. Overall, our current knowledge does not allow to give more credit to one factor in particular. The common sense is certainly to assume that the difference in landslide size is the reflection of a combination of factors.

**5.3 Drivers of shallow landslides**

Rainfall is the trigger of the shallow landslides that we have identified in this study, which is in agreement with the other studies in the region (Dewitte et al., 2021; Kubwimana et al., 2021). The spatial distribution of shallow landslides differs from the distribution of deep-seated landslides. This is mainly due to the anthropogenic factors such as deforestation that influence shallow processes (Table 4). The regional susceptibility model also indicates that deforestation is the most important factor in their occurrence (Table 5). Similarly, the analysis of frequency ratios shows that landslides disproportionately occur within areas that were deforested in the past 60 years, demonstrating the role of the forest in slope stabilization (Grima et al., 2020).

Shallow landslides in forest loss areas (Fig. 6a,b) have, on average, a smaller size compared to landslides in forest. This observation is in line with the findings of Depicker et al. (2021b) and is attributed to the decrease of regolith cohesion by reduced root cohesion and evapotranspiration due to forest loss (Glade, 2003; Masi et al., 2021), which allows for a smaller minimum critical area for landsliding (Vanacker et al., 2003; Milledge et al., 2014). In

short, human-induced land cover change is associated with an increase in the number of landslides and a shift of the frequency-area distribution towards smaller landslides (Guns and Vanacker, 2014).

In permanent anthropogenic environments (Fig. 6a,c), shallow landslides are less frequent, larger, and occur on less steep slopes as compared to shallow landslides in forest. Firstly, the steepest slopes in the anthropogenic

environments have been subject to increased landslide erosion the first few years after the original forest cover was removed (prior to 1955-1958) (Depicker et al., 2021b). As a result, we can assume that steep slopes in anthropogenic environments have less regolith available for landsliding compared to similar steep slopes in permanent forest areas. This process of regolith depletion is further exacerbated in cropland. Wilken et al. (2021) have measured in the region that erosion in cropland sites can reach up to about 40 cm in 55 years. Similarly, Heri-

Kazi and Bielders (2021a) measured mean erosion rates of the order of 11 mm/year on cropland. Regolith erosion has therefore the consequence of reducing the spatial extent of areas where landslides can occur. A second process that may explain the landslide pattern in the anthropogenic environments is that, in parallel to regolith erosion, one also has sedimentation and the formation of colluvium (Wilken et al., 2021); which results in local accumulation of material. The material forms a loose sedimentary deposit usually in places with lower slope angles. This could

be extra material available for the formation of landslides; the colluvium supply and a minimum depth of material being recognized as playing a key role in the occurrence of shallow landslides (Parker et al., 2016). Hence, we have less areas available for landslides, but a concentration of the susceptible places. A third explanation is probably related to soil management practices that influence erosion and water infiltration. In the region, usually on the less steep terrain, drainage ditches that favour water infiltration and hence an increase in pore-water pressure

are widely applied by farmers (Heri-Kazi and Bielders, 2021b).

### 5.4 Drivers of mining landslides and road landslides

The poor prediction rates of mining and road landslides when compared to the two shallow and deep-seated susceptibility models (Fig.7) shows that they respond to different environmental factors. Road construction and

mining activities are commonly associated with the presence of slope cuts and an increase of slope angle. These altered local topographic conditions cannot be constrained in the covariates derived from the SRTM or similar available products. In addition, the disturbances induced by roads and mining activities are not limited to the sole change of slope angle conditions. For example, this also implies changes in water runoff and infiltration, debuttressing, presence of fills and eventual overloading, excess stress from engine/digging, i.e., conditions that

can influence the size and frequency characteristics of landslides (Brenning et al., 2015; Arca et al., 2018; Froude and Petley, 2018; McAdoo et al., 2018; Vuillez et al., 2018; Tanyaş et al., 2022).

Road landslides are mostly shallow. While it is obvious that roads create favourable conditions for the initiation of landslides, as observed not only in the region (e.g. Kubwimana et al., 2021), but also worldwide (Froude and Petley, 2018; Sidle et al., 2006; Brenning et al., 2015; Arca et al., 2018; McAdoo et al., 2018; Vuillez et al., 2018;

Muñoz-Torrero Manchado et al., 2021; Tanyaş et al., 2022); an accurate spatio-temporal regional pattern of these human-induced slope failures cannot be assessed here. A substantial proportion of road landslides can only be observed in the field (Table 3). In addition, landslides along roads can easily disappear due to maintenance works.

Furthermore, many of the main roads were already present in the 1950's, their current impact therefore being altered.

Overall, mining conditions seem to lead to landslides whose smallest features are more frequent than what would occur under natural conditions as attested in the frequency area distribution (see Section 4.2). The area of mining landslides is significantly larger than that of road landslides and their regional distribution is slightly more in agreement with the characteristics of deep-seated landslides (Fig. 7d), which is logical as mining activities are related to the lithological characteristics of the landscape; i.e. a cause that typically has more influence on deeper

processes (Migoń, 2013; Dille et al., 2019).

Considering the recent development of the mining activities in the region (Butsic et al., 2015; Tyukavina et al., 2018; Musumba Teso et al., 2019), we can assume with confidence that the associated landslides represent slope instabilities that have occurred over a period of about 20 years whereas the recent deep-seated landslides represent slope failures that have occurred over the last 60 years. The distribution of the mining landslides is also restricted

spatially to some lithologies. With these specificities in mind and the fact that the number of inventoried mining and recent deep-seated landslides is relatively similar, respectively 152 and 159 (Table 2), this study confirms that mining activities increase the odds of landsliding. It has implication not only in terms of susceptibility assessment but also in assessing the population at risk, knowing that mined sites are populated. This is to be put in parallel with the findings of Depicker et al. (2021a) that show that the risk of shallow landslides has increased significantly

in the region during the last decades in the places where mining activities are found due, notably, to an increase in population.

**6 Conclusions**

The use of several sources of data allowed to build a very detailed and comprehensive landslide inventory in time and space for the region; a source of information unprecedented in such environments. This inventory enabled the

grouping of landslides into five categories: old and recent deep-seated landslides, shallow landslides, mining landslides and road landslides. Among deep-seated landslides, historical aerial photographs from the 1950's were an added value in the sense that they were used for differentiating between old (pre-1955) and recent (post-1955) slope processes. We deduce the differences in the driving factors and area distribution for old and recent deep-seated landslides, suggesting that factors of landslide occurrence are either different or change over time depending

on geodynamic and/or climatic conditions. The role of anthropogenic factors has been established in the occurrence of shallow landslides. Deforestation initially increases landsliding, but in the long term, when forest is permanently converted into agricultural land, landslide frequency appears to be lower compared to permanent forest lands. The impact of forest, forest cover changes and soil management practices depends on topographic conditions and regolith availability. The factors of occurrence of mining landslides significantly increase landsliding in areas that,

under natural conditions, would be less prone to slope failures. Our analysis shows that the importance of human activities must be considered when investigating landslide occurrence in regions under anthropogenic pressure. This is particularly needed when one sees that the changing spatio-temporal patterns of landslides associated with these activities tend to further exacerbate the risks that the population face.

On a more technical/methodological note, our study also demonstrates the importance of considering the timing and the depth of landslides as well as the differentiation between mining and road landslides. While several well-known landslide classification systems are used at the international level (Hungr et al., 2014; Sidle and Bogaard, 2016), these systems are not framed around the combination of the differentiation criteria that are used in this research. Our study does propose a unique effort at classifying landslide types in order to investigate them in the

context of the Anthropocene. We believe that our mapping effort and classification protocol is the most adapted (based on field observation and understanding of the landscape) in this case to address the problem of natural and human-induced landslides in the region. However, it certainly needs improvement to be used in a more universal way.

**Author contribution**

J-C.M.M C.B and O.D conceived the study. J-C.M.M processed and analysed the data and created the figures with inputs from C.B, O.D and A.D. J-C.M.M wrote the manuscript, with main inputs from C.B and O.D and contributions from A.D and B.S. J-C.M.M and O.D compiled the landslide inventory. J-C.M.M, O.D, E.M, T.T and L.B.M participated in the fieldwork data acquisition and interpretation. B.S. generated the orthomosaics from

aerial photographs. All the authors contributed to the final version of the paper. C.B. and O.D. obtained funding for this work.

**Declaration of Competing Interest**

The authors declare that they have no conflict of interest.

**Acknowledgements**

Jean-Claude Maki Mateso was supported by the Université catholique de Louvain (doctoral scholarship from the Administration of International Relations) and the Development Cooperation programme of the Royal Museum for Central Africa with support of the Directorate-General Development Cooperation and Humanitarian Aid of Belgium (RMCA-DGD). Arthur Depicker and Benoît Smets were supported by the PAStECA project (BELSPO

BRAIN-be Programme, Contract BR/165/A3/PASTECA; http://pasteca.africamuseum.be/). Elise Monsieurs was supported by a F.R.S. – FNRS PhD scholarship. The fieldwork was supported by the GeoRisCA (BELSPO SSD Programme, Contract SD/RI/02A; http://georisca.africamuseum.be/), RESIST (BELSPO STEREO-III Programme, Contract SR/00/305; http://resist.africamuseum.be/) and HARISSA (RMCA-DGD 2019-2024; https://georiska.africamuseum.be/en/projects/harissa) projects. We wish to thank B. Delvaux, J. Poesen and V.

Vanacker for their insightful discussions and recommendations regarding this research. A special thank goes to François Kervyn for his constant support to conduct research in the region.

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
