# Peer review of "Characteristics and causes of natural and human-induced landslides in a tropical mountainous region: the Rift flank west of Lake Kivu (DR Congo)"

_Natural Hazards and Earth System Sciences, 2021_

## Author Comment (AC1)

This study explored the impact of forest cover dynamics, roads and mining activities on the occurrence of landslides in the study area. The results showed that susceptibility patterns and area distributions are different between old and recent deep-seated landslides, and natural factors contributing to their occurrence were either different or changed over time, additionally, the forest dynamics and the presence of roads play a key role in their regional distribution pattern. I enjoyed reviewing your paper and believe it contributes to assess landslide susceptibility/risk for the local government. I have made comments in the hopes that they will be useful to improve the manuscript.

*The authors thank the reviewer for his/her assessment of the study's contribution to understanding landslide susceptibility/risk at the regional scale.*

General comments:

1. The abstract should be simplified, and it is the embodiment of the core of the article, so you can delete descriptions that are not very important. In addition, I suggest that research methods of article can be added in the abstract.

*We will simplify the abstract and, in the meantime, ponder the fact that adding methodological information is not contradictory to the request for simplification. We believe that the originality of our research lies more in the understanding of the slope processes rather than in the methods used for their investigation.*

2. In the introduction, you should be added some contents: (i) background information on the hazards of landslides, (ii) the methods of landside susceptibility, and you can analysis the advantages and disadvantages about different methods, (iii) influence factors of landslide should be listed and analyzed based on the previous achievements, especially in the study area or similar area, (iv) you can simplify some contents, such as lines 60 – 75.

*We agree with the comment and will include additional relevant information in this section. (i) With regard to the general background information on the hazards of landslides, we believe that the reviewer refers to the temporal aspects of the landslides. We will improve the content of the introduction that deals with this issue. Note that in that respect, lines 60-75 are relevant (iv). Since this paragraph is also key to support the compilation of the inventory, i.e. one of the key originalities of our research, we will avoid simplifying this section too much. (ii) Numerous susceptibility analysis methods have been used in the recent literature, with, as a common goal, the comparison of these methods. This has been extensively described by authors such as Reichenbach et al. (2018) who focus well on showing the advantages and limitations of each. Our study does not aim at evaluating the performance of one method or another but rather to determine the predictors related to the occurrence of the different types*

*of landslides studied. We therefore believe that expanding on this in the introduction is not relevant. Nevertheless, we will make sure that the fact that we will rely on a susceptibility assessment is understood before the presentation of the objectives. We will also include additional information on landslide susceptibility assessment to further support our methodological choices in the method section. (iii) The factors of landslide occurrence are described in the methodology and are those used in similar environments but notably in our study area. Nevertheless, the study opted for a selection of variables having a real supposed significance in the occurrence of a particular type of landslide. And it is one of the recommendations of the study to take into account these aspects in the selection of predictive variables supposed to contribute to their occurrence. We will make sure that that background information on this methodological approach is provided in a logical way. (iv) We will consider simplifying this section.*

3.  In the section 1.1, you can further analyze the relationship between LULC, population and landslides, because the article results showed that the forest dynamics and the presence of roads play a key role in their regional distribution pattern.

*The relevance of the remark is well taken in consideration. This aspect will be further developed in the section.*

4.  Authors have chosen 10 predictor variables use for the landslide susceptibility by applying different method, however, the triggering factor may be very difference for the shallow landslide and deep-seated landslide, and the assessment result will be changed, have you ever thought about that? If you considered, and you should be list evaluation factor for different landslide type.

*This study did not investigate the triggering factors in a direct way. For the shallow landslides, Dewitte al . (2021) demonstrate that all the observed hillslope instabilities during the last 2 decades are associated with rainfall. The research having been carried out in a data-scarce environment where timely access to information on the triggering of landslides is very difficult and where rain gauge information is also very scarce, an analysis of the triggering conditions could only be done over a region that is much larger than our study area using rainfall satellite products with a km-scale spatial resolution (Monsieurs et al., 2019a; 2019b). For a much larger area than our study area, Depicker et al. (2021b) also show the role of triggering rainfall in the spatial distribution of shallow landslides through the use, as rainfall data, of a regional climate model providing a resolution of 2.8 km. Considering what has already been done in other published research work and the context of data-scarcity of our study area, further analysis on the rainfall triggering conditions of the landslides would not have been possible at this stage. .*

*For the deep-seated landslides, a few of them are associated with rainfall events that are at the origin of landslide clusters. However, such information is only available for a limited number of recent landslides. For the very large majority of the deep-seated landslides, the triggering aspects can only be assumed, going from seismo-tectonic aspects to weathering and climatic controls (Dille et al., 2019; Dewitte et al., 2021). This information is already described in section 1.1.*

*We will make sure to make it clearer that the purpose of this research is to look at the susceptibility of the landslides, not at their triggering directly. Nevertheless, the difference of predictor variables for both shallow and deep-seated landslides highlighted through the susceptibility analysis allows to discuss triggering conditions (see section 4).*

5. Fig 7a and 7b presented the shallow landslide susceptibility and old deep-seated landslide susceptibility, author have analyzed the reason of differences, however, the results of fig 7a and 7b were also similar in a certain, you should be further explained.

*This is a relevant remark for which we will add the information about the aspects of similarities for both models.*

6. The distribution of different landslide was presented in the figure 8, meanwhile, authors should be further analyzed the reason.

*Some lines will be added for further explanation.*

7. In the section 4.3, authors have said rainfall is the trigger of the shallow landslides that we have identified in this study, and the reason explanation was lacked, however, this part have discussed that anthropogenic factors have an obviously effected on landslide, so you need further analyzed the relationship between shallow landslide and rainfall.

*In one of the above replies, we explain why the analysis of the rainfall as triggering factor is not possible in our research due to a lack of information (landslide dates, rainfall data) and the limited size of the study area. This is the reason why the triggering analysis could only be performed over a much large region than ours (Monsieurs et al., 1019a, 2019b). Nevertheless, we will highlight this issue better.*

Minor comments:

1. Lines 95-100 or 205: you can draw a figure about the change of LULC in the different years.

*This study considers the LULC on the long-term and, as said line 210, completes the study by Depicker et al. (2021b) who analysed the deforestation over the last 20 years and its impacts on landslides. We will explore if adding a figure on known LULC changes is relevant here since this is not the objective of our research.*

2. Line 110: you can draw a figure about population density or the change of population.

*There is very little information on the spatial distribution of the population characteristic. For example, the information from the spatially explicit Global Human Settlement Layer46, which is provided for four years: 1975, 1990, 2000 and 2015, is relevant for regional analysis. The data are available at: http://ghsl.jrc.ec.europa.eu/. The gridded data are the result of detecting the built-up land in satellite imagery and subsequently calculating the average population density per built-up pixel (at a 30 m resolution) by means of regional/national census data. However,*

*when looking a specific locations like our region, it shows some discrepancies. We will see to what extent adding a figure on the population is relevant. We may provide extra quantitative information depending on the quality of the available datasets.*

3. Lines 155- 160: add the website of different source data.

*We will add the references of these different data sources.*

4. Line 175: you can read the relevant references about landslide types, sush as Varnes, 1984; Cruden and Varnes, 1996; Hungr et al., 2014, and it may be better for your research.

*We are aware of these relevant references on the types of landslides according to the movement and the materials mobilized; in our research we make reference to Hungr et al. (2014).*

5. The section 2.2 may be put into section 1.1, you can check it.

*Since the reconstruction of the forest cover dynamics is a key element of this study; bringing new results, we believe that it should be included in a separate section of the materials and methods.*

6. Lines 300-3015: you can simplify.

*We will make these lines clearer.*

7. The format of Table 3 should be nice.

*We will arrange this table better.*

8. Lines 530-545: authors have discussed the difference between Van Den Eeckhaut's achievements and this study, and this is well. If you can add others' achievements that is in similar area or nearby the study area, and it may be better.

*We will consider the relevance of the remark with further documentation. However, to our knowledge, there is no other study of this kind in an area similar or nearby to ours.*

9. I suggest that the previous achievements (similar results or research) should be added, and they can abundant your research in the section 4.1-4.4.

*As best as we can, we have documented our discussions with relevant studies conducted in the tropical and mountainous context. And the previous work of our research teams has been documented in sections 4.1-4.4 as well. We will explore the very recent literature to see to what extend other achievements could help to support our discussion.*

References

Depicker, A., Jacobs, L., Mboga, N., Smets, B., Van Rompaey, A., Lennert, M., Wolff, E., Kervyn, F., Michellier, C., Dewitte, O. and Govers, G.: Historical dynamics of landslide risk from population and forest-cover changes in the Kivu Rift, Nat. Sustain., 4(11), 965–974, doi:10.1038/s41893-021-00757-9, 2021a.

Depicker, A., Govers, G., Jacobs, L., Campforts, B., Uwihirwe, J. and Dewitte, O.: Interactions between deforestation, landscape rejuvenation, and shallow landslides in the North Tanganyika–Kivu rift region, Africa, Earth Surf. Dyn., 9(3), 445–462, doi:10.5194/esurf-9-445-2021, 2021b.

Dewitte, O., Dille, A., Depicker, A., Kubwimana, D., Maki Mateso, J.-C., Mugaruka Bibentyo, T., Uwihirwe, J. and Monsieurs, E.: Constraining landslide timing in a data-scarce context: from recent to very old processes in the tropical environment of the North Tanganyika-Kivu Rift region, Landslides, 18(1), 161–177, doi:10.1007/s10346-020-01452-0, 2021.

Dille, A., Kervyn, F., Mugaruka Bibentyo, T., Delvaux, D., Ganza, G. B., Ilombe Mawe, G., Kalikone Buzera, C., Safari Nakito, E., Moeyersons, J., Monsieurs, E., Nzolang, C., Smets, B., Kervyn, M. and Dewitte, O.: Causes and triggers of deep-seated hillslope instability in the tropics – Insights from a 60-year record of Ikoma landslide (DR Congo), Geomorphology, 345, 106835, doi:10.1016/j.geomorph.2019.106835, 2019.

Hungr, O., Leroueil, S. and Picarelli, L.: The Varnes classification of landslide types, an update, Landslides, 11(2), 167–194, doi:10.1007/s10346-013-0436-y, 2014.

Monsieurs, Dewitte, Depicker and Demoulin: Towards a Transferable Antecedent Rainfall—Susceptibility Threshold Approach for Landsliding, Water, 11(11), 2202, doi:10.3390/w11112202, 2019a.

Monsieurs, E., Dewitte, O. and Demoulin, A.: A susceptibility-based rainfall threshold approach for landslide occurrence, Nat. Hazards Earth Syst. Sci., 19(4), 775–789, doi:10.5194/nhess-19-775-2019, 2019b.

---

## Author Comment (AC2)

**RC2**: 'Comment on nhess-2021-336', Anonymous Referee #2, 03 Dec 2021 reply

Reply to Referee #2

Dear Authors, I have read and carefully evaluatied your manuscript "Natural and human-induced landslides in a tropical mountainous region: the Rift flank west of Lake Kivu (DR Congo)". I am pleased to report that I found it a relevant, scientifically sound, and well drafted contribution to the journal. It surely deserves publication. However, I have some comments and I recommend to address them to further improve the paper.

Best regards.

*The authors sincerely thank the reviewer for his/her evaluation of our manuscript showing its relevance, scientific quality, and writing.*

---GENERAL COMMENTS---

My main concern is about the structure of the paper. Although it is excellently written, I found it too long and with many repetitions. These shortcomings can maybe be fixed with a reorganization of the paper structure. To be more precise, I found that some concepts are repeated at least twice. The first time in the material and methods section: there, they are outlined with a mid-level of detail, and many questions arise to the reader. Then, the results section repeats everything and add some more details answering most of the answers from the readers. This happens e.g. for landslide, forest, and parts of the analysis. Sometimes, things are repeated once more in the discussion. I think this structure does not help the reader and is not effective. You could try to either reorganize the structure (e.g. moving some preliminary results in the methods section) or shortening the information and comments in the material section to the minimum. In any case, please avoid repetitions and be concise and straight to the point.

*We understand that the structure of the document could be improved. We will do our best to make the reading lighter and avoid repetition. However, we will probably stick to the "method/result/discussion" structure. Note also that reviewer 3 specifically acknowledge the relevance of the structure.*

The state of the art review could be improved. Basically, the core of your study is a landslide susceptibility mapping (LSM) activity. Therefore, it would be advisable to include a paragraph about LSM. My advice is not to provide a detailed literature review, but you could focus on works that: (i) pertain to the same/nearby areas or areas with similar characteristics; use the same

susceptibility model; try deciphering the important role played by LULC dynamics or urbanization. In the literature, the last point is usually accounted for simply by using land cover maps and/or road network as input variables, but you may briefly acknowledge works that tried alternate approaches or specifically addressed this topic, such as:

Luti, T., Segoni, S., Catani, F., Munafò, M., & Casagli, N. (2020). Integration of remotely sensed soil sealing data in landslide susceptibility mapping. *Remote Sensing*, *12*(9), 1486.

Chen, L., Guo, Z., Yin, K., Shrestha, D. P., & Jin, S. (2019). The influence of land use and land cover change on landslide susceptibility: a case study in Zhushan Town, Xuan'en County (Hubei, China). *Natural hazards and earth system sciences*, *19*(10), 2207-2228.

Shu, H., Hürlimann, M., Molowny-Horas, R., González, M., Pinyol, J., Abancó, C., & Ma, J. (2019). Relation between land cover and landslide susceptibility in Val d'Aran, Pyrenees (Spain): Historical aspects, present situation and forward prediction. *Science of the total environment*, *693*, 133557.

Reichenbach, P., Mondini, A. C., & Rossi, M. (2014). The influence of land use change on landslide susceptibility zonation: the Briga catchment test site (Messina, Italy). *Environmental management*, *54*(6), 1372-1384.

*We agree, as also suggested by reviewer 1, that LSM information could be added in the introduction to better contextualize our research. However, we want to stress that LSM is only one part of our analysis. We appreciate the literature provided. We will see to what extend these studies are relevant to our study area and could be added if they bring an added value. Our research is in a tropical environment. As we pointed out in the manuscript, land uses are very changeable from one season to another and the use of polyculture in the same plot of land is commonplace. Therefore, a "traditional" land use analysis via susceptibility distribution is not possible. We will make sure that this aspect of our research is better highlighted.*

*As for urbanization, our study concerns a rural environment in Central Africa where major infrastructures are almost absent.*

To perform the LSM and to assess the variable importance you use logistic regression (LR) and frequency ratio (FR). These methods have a long tradition, but maybe they are a little outdated, as more effective and complex methods are continuously proposed (e.g. in the field of machine learning or deep learning). Don't you think this is a weakness of your work? I suggest defending the research strategy of using LR and FR on the introduction.

*We appreciate the relevance of your concern using LR and FR methodologies. In a regional study where our study area is included, Depicker et al (2020) used three susceptibility models (logistic regression, random forest, and support vector machine). These models give relatively similar results in terms of susceptibility assessment (both in terms of quantitative performance and geomorphological significance). The same conclusion about marginal differences between susceptibility models can be drawn from many other studies. Since the aim of our study was not to develop a new methodology or to show our ability to use complex methods; we relied on a LR*

*approach, i.e. a method that has been widely used in different regions of the world (Reichenbach et al., 2018) and that allows a rather easy interpretation of the results.*

*Regarding the FR, the goal of its use is to better understand the role of each variable in the contribution of the landslide occurrence in terms of process characterization. For example, when slope angle is highlighted by the LR as a significant variable, we still remain unaware of the types of slope that actually influence the occurrence of landslides.*

*We will add this information in the method section.*

To my understanding, the shape of the area-frequency curves is quite logical. It is normal to have a rollover: it can be interpreted that below that area, the inventory progressively becomes incomplete because smaller landslides are harder to identify (and map), for several reasons. So, I wouldn't spend so many energies to defend the presence of the rollover in your curves: it is a typical feature, useful to identify the size of the landslides that your model could probably miss.

*Thank you for the comment, we will take it into account to lighten the text.*

If I understood correctly, you assess the importance of a variable by running the susceptibility model with only that single variable. I am not very convinced about this approach. The possible interplay among variables is lost. Moreover, a single-variable susceptibility assessment seems of little use. At present, one of the reasons why more sophisticated LSM methods are used is that they also have internal modules that assess the variable importance.

*That is right; we have run the model for each predictor variable selected for shallow landslides and old deep-seated landslides. The goal was to evaluate the extent to which these predictors can be used to differentiate between landslide and no landslide locations. This step is to help us to better understand the multivariate LR models, although we agree that this is not to be considered without caveats. We will make sure to stress this better in the manuscript.*

---SPECIFIC REMARKS---

L27 which dynamics? Please, be more specific.

*Thanks for the comment, it is about the forest loss. We will correct this in the text.*

L35 which susceptibility models?

*As written, these are the susceptibility models calibrated for the shallow landslides and old deep-seated landslides. We will try make the text clearer.*

L56-59. It depends also how the human intervention was designed and executed. There is a big difference if you just cut a slope and build a house (or a road), or if the cut is accompanied by some additional works (drainages, concrete walls, …). This should also be highlighted elsewhere in the manuscript when you write about this issue.

*In this rural environment, concrete walls are almost non-existent. As for the drainage systems, when present, they are very basic (one or two usually-unmaintained ditches on (both) side(s) of the road. Note also that most of the roads are dirt roads, frequently impacted by rill and gully erosion due to a lack of maintenance. The roads are therefore contributing to an undesigned concentration and rerouting of the runoff. We will provide these details in the revised version.*

Section 1.1 Besides describing the lithology, a short overview of the geological setting could be a nice addendum to this section.

*We agree with the relevance of the brief description of the geological setting of the study area.*

Fig. 1 For the cities, I suggest using a color that better stands out from the colors used for elevation. E.g. black. And you could also add it in the legend. I initially confused cities outside the study area with parts of the study area.

*Thanks for the comment. We will adjust it.*

L155-160: From what dates are the images? (This is explained later, but at this point of the manuscript, it is a spontaneous question: see my first general comment).

*Thanks for the comment. We will indicate here the dates; also in line with your general comment.*

L174-178. Usually, a landslide is also considered shallow when the ratio depth/width or depth/length is small. I guess this is also your case?

*Our criterion was based on the estimated depth of the surface of rupture. This was assessed through the analysis of the main scarp shape. As we are looking at the conditions that prevailed at the source of the landslides, we did not investigate further their morphometry. Note that in general deep-seated landslides can also have a small depth/length ratio and that, on the other hand, shallow landslides can have a rather long runout (as is frequently observed in our study area).*

*.*

L225. This is not clear to me.

*The SRTM digital terrain model used for the topographic analysis is posterior to the occurrence of the old deep-seated landslides. Therefore, the digital terrain model is affected by the deformation due to the landslides. Calculating the slope values at the level of the main scarp for this type of landslide would give values that are the consequences of landslides rather than the causes of their origin. Hence, we placed a second point outside the landslide on the nearby slope that seems unaffected by landsliding. The morphology of this slope would better reflect the topographic similarity before the triggering of this landslide. We will make the text clearer.*

Table 1. the meaning of "reference" in the second column is explained only later. This is confusing.

*We will make it clearer.*

Table 1. The forest dynamics information is very interesting. In my opinion, it deserves also a figure. Unfortunately, the figure comes only after some pages. This is another example of specific issues comprehended in my first general comment.

*This figure was placed in the results section as it is the first map of the long-term forest cover dynamics for the region that has been produced hitherto. While Table 1 only serves to present the predictor variables to be used in the study.*

Figure 2. The forest cover color hides the information about elevation. Didn't you already display the elevation in Fig 1? Here, you could just use hillshade and forest cover.

*We tested several ways of presenting this figure, trying colors that contrast better with the elevation and the forest cover. In our opinion, both layers should appear on the figure to show a densification of landslides in the mountainous areas and sparseness in the forest area. Using hillshade alone will not allow readers to locate the topographic context where the different types of landslides are found. We try to present both types of information as best as possible without hiding the information in the underlying layer.*

377 "these sources"

*We will correct the mistake. Thank you for the comment.*

Table 4. It seems to me that the bedrock lithology has little influence in determining if a landslide will be shallow or deep seated. Maybe because the lithologies produce similar soils and the actual depth of soils (driven by morphology) is the real control?

*We indeed show that the lithology is of lesser importance in our study area. For the deep-seated landslides this is in agreement with the findings of Depicker et al. (2021b), which show that the various lithologies in the region have similar rock strength properties. As we also show that the topography and the presence of faults play a role, it is therefore no surprise that the role of lithology is somehow attenuated.*

*Note also that despite the fact that we use an unprecedented lithological information (in terms of accuracy and resolution), there is a lack of data on the regolith, its depths and the soil types. Field work confirms that soils and regolith can be very different in terms of type and depth, the latter one being highly variable spatially (sometimes over a very short distance of a few meters along a hillslope, one have regolith thickness that varies from > 10 m to nearly zero). The only way to provide an assessment of the regolith thickness at the regional scale is to assume that it varies with slope gradient. It is based on this assumption, that is further supported by the analysis of Depicker et al. (2021b) carried on shallow landslides in the region, that we explain the distribution of the shallow landslides in section 4.3.*

478. I like that recent landslides are reasonably well predicted by a model trained with the old ones. This is like a multitemporal validation. It could be worth mentioning it.

*Thank you for your appreciation! We will mention that.*

493-In an earlier part of the manuscript you mentioned that elevation can be considered a proxy for meteo-climatic characteristics. Why you discard this interpretation here?

*You are right. Thanks for the comment. We will add a few lines about this aspect.*

593 - Actually, the explanation may be that with this approach you artificially create incompleteness in your inventory. (this interpretation is in accordance with my general comment about frequency-area curves).

*In this section our interpretations take into account all 1013 shallow landslides inventoried (see Figure 6), hence there is no creation of incompleteness of the inventory. For some analyses of the frequency density curves excluding event-related landslides from the inventory, we only investigate the potential bias that landslide events could introduce.*

604 the influence of vegetation on slope stability is somehow a relevant part of the phenomena you are investigating, but this is never mentioned explicitly. Why didn't you openly prepare this issue in advance and you don't mention it explicitly? Forest loss means (I think) reduced root cohesion and reduced evapotranspiration. I would mention it clearly. You could also make reference to some works such as

Masi, E. B., Segoni, S., & Tofani, V. (2021). Root Reinforcement in Slope Stability Models: A Review. *Geosciences*, *11*(5), 212.

Schwarz, M., Preti, F., Giadrossich, F., Lehmann, P., & Or, D. (2010). Quantifying the role of vegetation in slope stability: A case study in Tuscany (Italy). *Ecological Engineering*, *36*(3), 285-291.

Arnone, E., Caracciolo, D., Noto, L. V., Preti, F., & Bras, R. L. (2016). Modeling the hydrological and mechanical effect of roots on shallow landslides. *Water Resources Research*, *52*(11), 8590-8612.

Glade, T. (2003). Landslide occurrence as a response to land use change: a review of evidence from New Zealand. *Catena*, *51*(3-4), 297-314.

*Thank you for the comment, we will take it into consideration.*

608-612. I think there is (also) another explanation: the slope value you are using is an averaged value, while the built environment may be characterized by a locally steeper value. As instance, in a slope cut you could have a small 90° slope, which may not be well captured by the DTM. Even outside artificial environment, a similar situation may be present.

*As mentioned in the methodology, we study the occurrence factors at the scale of a pixel, i.e. a point, taken manually, in the middle of the main scarp for shallow landslides. We have also discarded the shallow landslides found in direct proximity to the roads. Nevertheless, we can*

*indeed not ignore that the SRTM, because of its resolution, does not capture all the slope characteristics such as the cuts. We will mention this to better nuance our interpretation.*

670-675. The stylistic writing of this part is so different from the rest of the paper. Here the sentences are very short and telegraphic. I suggest to better link them.

*Thanks for the comment, we will work on this.*

References

Depicker, A., Jacobs, L., Delvaux, D., Havenith, H.-B., Maki Mateso, J.-C., Govers, G. and Dewitte, O.: The added value of a regional landslide susceptibility assessment: The western branch of the East African Rift, Geomorphology, 353, 106886, doi:10.1016/j.geomorph.2019.106886, 2020.

Depicker, A., Jacobs, L., Mboga, N., Smets, B., Van Rompaey, A., Lennert, M., Wolff, E., Kervyn, F., Michellier, C., Dewitte, O. and Govers, G.: Historical dynamics of landslide risk from population and forest-cover changes in the Kivu Rift, Nat. Sustain., 4(11), 965–974, doi:10.1038/s41893-021-00757-9, 2021a.

Depicker, A., Govers, G., Jacobs, L., Campforts, B., Uwihirwe, J. and Dewitte, O.: Interactions between deforestation, landscape rejuvenation, and shallow landslides in the North Tanganyika–Kivu rift region, Africa, Earth Surf. Dyn., 9(3), 445–462, doi:10.5194/esurf-9-445-2021, 2021b.

Reichenbach, P., Rossi, M., Malamud, B. D., Mihir, M. and Guzzetti, F.: A review of statistically-based landslide susceptibility models, Earth-Science Rev., 180(March), 60–91, doi:10.1016/j.earscirev.2018.03.001, 2018.

---

## Author Response (AR1)

Response to Referees comments:

**Natural and human-induced landslides in a tropical mountainous region: the Rift flank west of Lake Kivu (DR Congo)**

April 05, 2022

We would like to thank the editor and the three reviewers for their pertinent and constructive comments and also for their appreciation of the quality of the work. We have considered these comments which revealed some weaknesses in the presentation of our work. Through completing the sometimes-divergent suggested edits, the revised manuscript benefits from an improvement in the overall presentation and clarity.

Our answers to the reviewers' comments are presented as follows: the reviewers' comments are shown in **black**; the answers are text in *blue italics*. And the revised texts are in **green**. The lines of the final manuscript are shown in **purple**, while the lines of the manuscript with the tracked changes are in **orange**.

**Contents**

**I. Reply to Referee # 1**

This study explored the impact of forest cover dynamics, roads and mining activities on the occurrence of landslides in the study area. The results showed that susceptibility patterns and area distributions are different between old and recent deep-seated landslides, and natural factors contributing to their occurrence were either different or changed over time, additionally, the forest dynamics and the presence of roads play a key role in their regional distribution pattern. I enjoyed reviewing your paper and believe it contributes to assess landslide susceptibility/risk for the local government. I have made comments in the hopes that they will be useful to improve the manuscript.

*The authors thank the reviewer for his/her assessment of the study's contribution to understanding landslide susceptibility/risk at the regional scale.*

**1.1. General comments:**

1.1.1. The abstract should be simplified, and it is the embodiment of the core of the article, so you can delete descriptions that are not very important. In addition, I suggest that research methods of article can be added in the abstract.

*We have simplified the abstract while adding info on the methods*

**L20-22** / **20-22:** …To do so, we compile a comprehensive multi-temporal inventory of 2730 landslides of different types and analyze it via frequency-area statistics, frequency ratio distribution and logistic regression susceptibility assessment.

1.1.2. In the introduction, you should be added some contents: (i) background information on the hazards of landslides, (ii) the methods of landside susceptibility, and you can analysis the advantages and disadvantages about different methods, (iii) influence factors of landslide should be listed and analyzed based on the previous achievements, especially in the study area or similar area, (iv) you can simplify some contents, such as lines 60 – 75.

*We agree with the comment and included additional relevant information in this section. (i) With regard to the general background information on the hazards of landslides, we believe that the reviewer refers to the temporal aspects of the landslides. We improve the content of the introduction that deals with this issue, specifically with the issues related to human activities. Note that in that respect,* **lines 54-66/64-79**, *are relevant. (iv)Since this paragraph is also key to support the compilation of the inventory, i.e. one of the key originalities of our research, we avoided simplifying this section too much. (ii) Numerous susceptibility analysis methods have been used in the recent literature, with, as a common goal, the comparison of these methods. This has been extensively described by authors such as Reichenbach et al. (2018) who focus well on showing the advantages and limitations of each. Our study does not aim at evaluating the performance of one method or another but rather to determine the predictors related to the occurrence of the different types of landslides studied. We therefore believe that expanding on this in the introduction is not relevant; especially with the fact susceptibility is only one of the three methods that we use here (with frequency ratio and frequency-area statistics)*

*Nevertheless, we included additional information on landslide susceptibility assessment to further support our methodological choices in the method section (L223-236 / 246-260). (iii) The factors of landslide occurrence are described in the methodology and are those used in similar environments but notably in our study area. Nevertheless, the study opted for a selection of variables having a real supposed significance in the occurrence of a particular type of landslide. And it is one of the recommendations of the study to take into account these aspects in the selection of predictive variables supposed to contribute to their occurrence.*

L223-236 / 246-260: Landslide susceptibility approaches are numerous and more or less complex in terms of modelling implementation and result interpretability (Reichenbach et al., 2018). In a regional analysis where our study area is included, Depicker et al.( 2020) used three susceptibility models, namely logistic regression, random forests, and support vector machines. These models gave relatively similar results in terms of quantitative performance and geomorphological plausibility. The same conclusion about marginal differences between susceptibility models can be drawn from many other studies. Since our study does not aim to develop a new methodology nor to show the ability to use complex methods; we relied on a logistic regression approach (Hosmer and Lemeshow, 2000) to determine the predictor variables related to the occurrence of the different types of landslides. Logistic regression isa straightforward method that has been widely used (Reichenbach et al., 2018) and that allows a rather easy interpretation of the results (e.g. Jacobs et al., 2018; Depicker et al., 2020).

Frequency ratio (Lee and Pradhan, 2007) models were used as a complementary approach to better understand the role of each variable in the contribution of the landslide occurrence in terms of process characterization. For example, when slope angle is highlighted by a logistic regression model as a significant variable, we still remain unaware of the types of slopes that actually influence the occurrence of landslides.

1.1.3. In the section 1.1, you can further analyze the relationship between LULC, population and landslides, because the article results showed that the forest dynamics and the presence of roads play a key role in their regional distribution pattern.

*The relevance of the remark is well taken in consideration. As explained in "Minor comment 2" the data on population currently available do not allow a regional study at the scale of our study. In addition, we believe that such an analysis, although interesting, would be out of scope here. Nevertheless, we have added information on the roads.*

L108-110 / 122-124: … The road network is relatively limited. Most roads are dirt roads and are poorly maintained, and there are no built-up walls (concrete, gabions) to stabilize the cut slopes.

1.1.4. Authors have chosen 10 predictor variables use for the landslide susceptibility by applying different method, however, the triggering factor may be very difference for the shallow landslide and deep-seated landslide, and the assessment result will be changed, have you ever thought about that? If you considered, and you should be list evaluation factor for different landslide type.

*This study did not investigate the triggering factors in a direct way. For the shallow landslides, Dewitte al., (2021) demonstrate that all the observed hillslope instabilities during the last 2 decades are associated with rainfall. The research having been carried out in a data-scarce environment where timely access to information on the triggering of landslides is very difficult and where rain gauge information is also very scarce, an analysis of the triggering conditions could only be done over a region that is much larger than our study area using rainfall satellite products with a km-scale spatial resolution (Monsieurs et al., 2019a; 2019b). For a much larger area than our study area, Depicker et al. (2021b) also show the role of triggering rainfall in the spatial distribution of shallow landslides through the use, as rainfall data, of a regional climate model providing a resolution of 2.8 km. Considering what has already been done in other published research work and the context of data-scarcity of our study area, further analysis on the rainfall triggering conditions of the landslides would not have been possible at this stage. .*

*For the deep-seated landslides, a few of them are associated with rainfall events that are at the origin of landslide clusters. However, such information is only available for a limited number of recent landslides. For the very large majority of the deep-seated landslides, the triggering aspects can only be assumed, going from seismo-tectonic aspects to weathering and climatic controls (Dille et al., 2019; Dewitte et al., 2021). This information is already described in section 1.1.*

*We made clearer that the purpose of this research is to look at the susceptibility of the landslides, not at their triggering directly. Nevertheless, the difference of predictor variables for both shallow and deep-seated landslides highlighted through the susceptibility analysis allows to discuss triggering conditions (see section 4).*

**L247-249** / **274-276:** The purpose of this research is to examine the predictor variables that contribute to the susceptibility of the different landslide types; not to look directly for their triggering factors. Nevertheless, the different predictors, highlighted by the susceptibility analysis allow to discuss the triggering conditions.

1.1.5. Fig 7a and 7b presented the shallow landslide susceptibility and old deep-seated landslide susceptibility, author have analyzed the reason of differences, however, the results of fig 7a and 7b were also similar in a certain, you should be further explained.

*This is a relevant remark for which we added some information.*

**L447-448** / **501-502**: … (Fig. 7). At first sight, both models have spatial similarities of high susceptibility on the eastern part of the region; while the entire western part is weakly susceptible (Fig. 7a,b). However, when we go into detail, the …

1.1.6. The distribution of different landslide was presented in the figure 8, meanwhile, authors should be further analyzed the reason.

*Some lines are added for further explanation.*

**L484-485** / **549**: … Both types of landslides are favoured by slopes angles > 20-25°.

**L489-492 / 554-557**: … probably be explained by a cumulative effect of forest loss, steeper slopes and increased orographic rainfall associated to these elevations (Fig. 8c). The positive frequency ratio in the 1400-1700 m elevation class is related to the area of permanent anthropogenic environment. This zone is characterized by low forest cover and relatively low slopes (Fig. 8c).

    1.1.7. In the section 4.3, authors have said rainfall is the trigger of the shallow landslides that we have identified in this study, and the reason explanation was lacked, however, this part have discussed that anthropogenic factors have an obviously effected on landslide, so you need further analyzed the relationship between shallow landslide and rainfall.

*In one of the above replies, we explain why the analysis of the rainfall as triggering factor is not possible in our research due to a lack of information (landslide dates, rainfall data) and the limited size of the study area. This is the reason why the triggering analysis could only be performed over a much large region than ours (Monsieurs et al., 1019a, 2019b). Nevertheless, we highlight this issue better in section 2.3.1:*

**L247-249** / **274-276**: The purpose of this research is to examine the predictor variables that contribute to the susceptibility of the different landslide types; not to look directly for their triggering factors. Nevertheless, the different predictors, highlighted by the susceptibility analysis allow to discuss the triggering conditions.

**1.2. Minor comments:**

    1.2.1. Lines 95-100 or 205: you can draw a figure about the change of LULC in the different years.

*This study considers the LULC on the long-term and, as said line* **202** */* **224***, completes the study by Depicker et al. (2021b) who analysed the deforestation over the last 20 years and its impacts on landslides. Note that to also reply to the comment "Specific remarks: Table 1" raised by Reviewer #2, we have moved, from the results, the figure (formerly number "Figure 5" and now numbered "Figure 2"on the reconstructed LULC changes. We have also grouped the texts discussing this figure in Section 2.2.*

**L208-221 / 231-244 :**

In 1955-58, 42 % of the territory was already deforested (Fig. 2a). From 1955-58 to 2016, the loss of forest continued, the forest cover decreasing from 58 % to 24 % of the study area. The area affected by the forest loss over the last 60 years is larger than the remaining permanent forest (Fig. 2b). The comparison of forest areas between 1955-58 and 2016 allows to consider four classes for the forest dynamics. Permanent forest corresponds to forest areas that are present at both dates. The forest loss class corresponds to forests present in 1955-58 that have disappeared in 2016. Since it is impossible to identify for each portion of the landscape the exact cause of forest loss, this class contains a mix of various forest management practices and other causes of forest cut/removal. The forest gain class represents the new forest that has appeared since 1955-58. Similarly, the causes associated with the occurrence of new forest are not exactly known; afforestation and natural forest

regeneration being certainly drivers at play. Permanent anthropogenic environment (e.g. cropland, grassland, built-up lands) means that the landscape was not forested in both dates and it is assumed that it remained so during that period.

[Figure]

Figure 2: Forest cover dynamics over the last 60 years. (a) Forest cover in 1955-58 and 2016; (b) Areas of forest cover change between 1955-58 and 2016. Details for the images used in this figure are in Table 1.

1.2.2.  Line 110: you can draw a figure about population density or the change of population.

*There is very little information on the spatial distribution of the population characteristics. For example, the information from the spatially explicit Global Human Settlement Layer46, which is provided for four years: 1975, 1990, 2000 and 2015, is relevant for regional analysis. The data are available at: http://ghsl.jrc.ec.europa.eu/. The gridded data are the result of detecting the built-up land in satellite imagery and subsequently calculating the average population density per built-up pixel (at a 30 m resolution) by means of regional/national census data. However, when looking a specific location like our region, it shows some discrepancies. Adding a figure on the population is not relevant. You can see the figure below.*

[Figure]

1.2.3.  Lines 155- 160: add the website of different source data.

*The text specifies the references or the place to access the sources of the images. In order to make the text shorter, there is no need to specify for example the site to download Google Earth Pro. See lines* **143–152** */* **163-172.**

1.2.4.  Line 175: you can read the relevant references about landslide types, sush as Varnes, 1984; Cruden and Varnes, 1996; Hungr et al., 2014, and it may be better for your research.

*We are aware of these relevant references on the types of landslides according to the movement and the materials mobilized; in our research we make reference to Hungr et al. (2014). See line* **400** */* **444.**

1.2.5.  The section 2.2 may be put into section 1.1, you can check it.

*Since the reconstruction of the forest cover dynamics is a key element of this study; bringing new results, we believe that it should be included in a separate section of the materials and methods. We have therefore kept the original structured (as also suggested by Reviewer #2).*

1.2.6.  Lines 300-315: you can simplify.

*We made this part clearer. See Lines* **320-334** */* **357-374**.

**Lines 320-323 /357-361:** Although this is quite a straightforward approach that does not consider the possible interplay among predictor variables, this allows to have a first quantitative insight on the importance…

**L334 / 372:** …to indicate for each bin of the predictor variable the probability of occurrence of a landslide.

   1.2.7.   The format of Table 3 should be nice.

*We arranged this table better.*

**L381 / 423**

| Landslide type | Number of LS mapped in the images and checked in the field | TP | FP | FN | Precision (%) | Total number of LS viewed in the field |
|---|---|---|---|---|---|---|
| Deep-seated (old) | 248 | 239 | 9 | 60 | 96 | 308 |
| Deep-seated (recent) | 47 | 44 | 3 | 4 | 94 | 51 |
| Shallow | 426 | 420 | 6 | 55 | 99 | 481 |
| Mining | 15 | 9 | 6 | 2 | 60 | 17 |
| Road | 50 | 45 | 5 | 5 | 90 | 55 |
| Total | 786 | 757 | 29 | 126 | 96 | 912 |

   1.2.8.   Lines 530-545: authors have discussed the difference between Van Den Eeckhaut's achievements and this study, and this is well. If you can add others' achievements that is in similar area or nearby the study area, and it may be better.

*To our knowledge, there is no other study of this kind in an area similar or nearby to ours. Nevertheless, note that this paragraph has been improved (based also on the comment 1 from Reviewer #3). See lines **527-532** / **587 - 605**.*

   1.2.9.   I suggest that the previous achievements (similar results or research) should be added, and they can abundant your research in the section 4.1-4.4.

*As best as we can, we have documented our discussions with relevant studies conducted in the tropical and mountainous context. And the previous work of our research teams has been documented in sections 4.1-4.4 as well.*

*References*

Depicker, A., Jacobs, L., Mboga, N., Smets, B., Van Rompaey, A., Lennert, M., Wolff, E., Kervyn, F., Michellier, C., Dewitte, O. and Govers, G.: Historical dynamics of landslide risk from population and forest-cover changes in the Kivu Rift, Nat. Sustain., 4(11), 965–974, doi:10.1038/s41893-021-00757-9, 2021a.

Depicker, A., Govers, G., Jacobs, L., Campforts, B., Uwihirwe, J. and Dewitte, O.: Interactions between deforestation, landscape rejuvenation, and shallow landslides in the North Tanganyika–Kivu rift region, Africa, Earth Surf. Dyn., 9(3), 445–462, doi:10.5194/esurf-9-445-2021, 2021b.

Dewitte, O., Dille, A., Depicker, A., Kubwimana, D., Maki Mateso, J.-C., Mugaruka Bibentyo, T., Uwihirwe, J. and Monsieurs, E.: Constraining landslide timing in a data-scarce context: from recent to very old processes in the tropical environment of the North Tanganyika-Kivu Rift region, Landslides, 18(1), 161–177, doi:10.1007/s10346-020-01452-0, 2021.

Dille, A., Kervyn, F., Mugaruka Bibentyo, T., Delvaux, D., Ganza, G. B., Ilombe Mawe, G., Kalikone Buzera, C., Safari Nakito, E., Moeyersons, J., Monsieurs, E., Nzolang, C., Smets, B., Kervyn, M. and Dewitte, O.: Causes and triggers of deep-seated hillslope instability in the tropics – Insights from a 60-year record of Ikoma landslide (DR Congo), Geomorphology, 345, 106835, doi:10.1016/j.geomorph.2019.106835, 2019.

Hungr, O., Leroueil, S. and Picarelli, L.: The Varnes classification of landslide types, an update, Landslides, 11(2), 167–194, doi:10.1007/s10346-013-0436-y, 2014.

Monsieurs, Dewitte, Depicker and Demoulin: Towards a Transferable Antecedent Rainfall—Susceptibility Threshold Approach for Landsliding, Water, 11(11), 2202, doi:10.3390/w11112202, 2019a.

Monsieurs, E., Dewitte, O. and Demoulin, A.: A susceptibility-based rainfall threshold approach for landslide occurrence, Nat. Hazards Earth Syst. Sci., 19(4), 775–789, doi:10.5194/nhess-19-775-2019, 2019b.

**II.    Reply to Referee #2**

Dear Authors, I have read and carefully evaluatied your manuscript "Natural and human-induced landslides in a tropical mountainous region: the Rift flank west of Lake Kivu (DR Congo)". I am pleased to report that I found it a relevant, scientifically sound, and well drafted contribution to the journal. It surely deserves publication. However, I have some comments and I recommend to address them to further improve the paper.

Best regards.

*The authors sincerely thank the reviewer for his/her evaluation of our manuscript showing its relevance, scientific quality, and writing.*

**2.1.   General comments---**

2.1.1. My main concern is about the structure of the paper. Although it is excellently written, I found it too long and with many repetitions. These shortcomings can maybe be fixed with a reorganization of the paper structure. To be more precise, I found that some concepts are repeated at least twice. The first time in the material and methods section: there, they are outlined with a mid-level of detail, and many questions arise to the reader. Then, the results section repeats everything and add some more details answering most of the answers from the readers. This happens e.g. for landslide, forest, and parts of the analysis. Sometimes, things are repeated once more in the discussion. I think this structure does not help the reader and is not effective. You could try to either reorganize the structure (e.g. moving some preliminary results in the methods section) or shortening the information and comments in the material section to the minimum. In any case, please avoid repetitions and be concise and straight to the point.

*We understand that the structure of the document can be improved. We did our best to make the reading lighter and avoid repetition. However, we sticked to the "method/result/discussion" structure. Note also that Reviewer #3 specifically acknowledged the relevance of the structure.*

*Some lines are deleted, for example in:*

*Introduction section:* **L 55-57; 68-71; 142-144; 156-158.**

*Materials and methods section:* **L 205-206; 216-217; 260-262; 296-306; 360-361.**

*Results section:* **L 457-458; 472-474; 504-509; 521-522; 534-536.**

*Discussions section:* **L 563-567; 598-599; 603-605; 619-620.**

*We have also grouped the LULC analysis in Section 2.2, moving the now figure 2 from the results to this section. See Lines* **L208-221 / 231-244.**

2.1.2. The state of the art review could be improved. Basically, the core of your study is a landslide susceptibility mapping (LSM) activity. Therefore, it would be advisable to include a paragraph about LSM. My advice is not to provide a detailed literature review, but you could focus on works that: (i) pertain to the same/nearby areas or areas with similar characteristics; use the same susceptibility model; try deciphering the important role played by LULC dynamics or urbanization. In the literature, the last point is usually accounted for simply by using land cover maps and/or road network as input variables, but you may briefly acknowledge works that tried alternate approaches or specifically addressed this topic, such as:

Luti, T., Segoni, S., Catani, F., Munafò, M., & Casagli, N. (2020). Integration of remotely sensed soil sealing data in landslide susceptibility mapping. *Remote Sensing*, *12*(9), 1486.

Chen, L., Guo, Z., Yin, K., Shrestha, D. P., & Jin, S. (2019). The influence of land use and land cover change on landslide susceptibility: a case study in Zhushan Town, Xuan'en County (Hubei, China). *Natural hazards and earth system sciences*, *19*(10), 2207-2228.

Shu, H., Hürlimann, M., Molowny-Horas, R., González, M., Pinyol, J., Abancó, C., & Ma, J. (2019). Relation between land cover and landslide susceptibility in Val d'Aran, Pyrenees (Spain): Historical aspects, present situation and forward prediction. *Science of the total environment*, *693*, 133557.

Reichenbach, P., Mondini, A. C., & Rossi, M. (2014). The influence of land use change on landslide susceptibility zonation: the Briga catchment test site (Messina, Italy). *Environmental management*, *54*(6), 1372-1384.

*As also suggested by reviewer 1, a paragraph on LSM information has been added in the methodological section 2.3. Furthermore; some of the references suggested here have now been added to the "modified" section 2.2 dedicated to the forest dynamics. See Lines* **199** / **221 - 222**.

**L223-236** / **246-260**: Landslide susceptibility approaches are numerous and more or less complex in terms of modelling implementation and result interpretability (Reichenbach et al., 2018). In a regional analysis where our study area is included, Depicker et al.( 2020) used three susceptibility models, namely logistic regression, random forests, and support vector machines. These models gave relatively similar results in terms of quantitative performance and geomorphological plausibility. The same conclusion about marginal differences between susceptibility models can be drawn from many other studies. Since our study does not aim to develop a new methodology nor to show the ability to use complex methods; we relied on a logistic regression approach (Hosmer and Lemeshow, 2000) to determine the predictor variables related to the occurrence of the different types of landslides. Logistic regression isa straightforward method that has been widely used (Reichenbach et al., 2018) and that allows a rather easy interpretation of the results (e.g. Jacobs et al., 2018; Depicker et al., 2020).

Frequency ratio (Lee and Pradhan, 2007) models were used as a complementary approach to better understand the role of each variable in the contribution of the landslide occurrence in terms of process characterization. For example, when slope angle is highlighted by a logistic regression

model as a significant variable, we still remain unaware of the types of slopes that actually influence the occurrence of landslides.

2.1.3. To perform the LSM and to assess the variable importance you use logistic regression (LR) and frequency ratio (FR). These methods have a long tradition, but maybe they are a little outdated, as more effective and complex methods are continuously proposed (e.g. in the field of machine learning or deep learning). Don't you think this is a weakness of your work? I suggest defending the research strategy of using LR and FR on the introduction.

*We appreciate the relevance of your concern using LR and FR methodologies. In a regional study where our study area is included, Depicker et al (2020) used three susceptibility models (logistic regression, random forest, and support vector machine). These models give relatively similar results in terms of susceptibility assessment (both in terms of quantitative performance and geomorphological significance). The same conclusion about marginal differences between susceptibility models can be drawn from many other studies. Since the aim of our study was not to develop a new methodology or to show our ability to use complex methods; we relied on a LR approach, i.e. a method that has been widely used in different regions of the world (Reichenbach et al., 2018) and that allows a rather easy interpretation of the results.*

*Regarding the FR, the goal of its use is to better understand the role of each variable in the contribution of the landslide occurrence in terms of process characterization. For example, when slope angle is highlighted by the LR as a significant variable, we still remain unaware of the types of slope that actually influence the occurrence of landslides.*

*We added this information in the method section 2.3.*

**L224-230** / **247-253:** In a regional analysis where our study area is included, Depicker et al.( 2020) used three susceptibility models, namely logistic regression, random forests, and support vector machines. These models gave relatively similar results in terms of quantitative performance and geomorphological plausibility. The same conclusion about marginal differences between susceptibility models can be drawn from many other studies. Since our study does not aim to develop a new methodology nor to show the ability to use complex methods; we relied on a logistic regression approach (Hosmer and Lemeshow, 2000) to determine the predictor variables related to the occurrence of the different types of landslides.

2.1.4. To my understanding, the shape of the area-frequency curves is quite logical. It is normal to have a rollover: it can be interpreted that below that area, the inventory progressively becomes incomplete because smaller landslides are harder to identify (and map), for several reasons. So, I wouldn't spend so many energies to defend the presence of the rollover in your curves: it is a typical feature, useful to identify the size of the landslides that your model could probably miss.

*Thank you for the comment, we have judged to maintain these details as it is part of the originality of this study. Nevertheless, the corresponding paragraph has been modified to make this discussion clearer. See Lines* **527-532** */* **597 - 605**.

2.1.5. If I understood correctly, you assess the importance of a variable by running the susceptibility model with only that single variable. I am not very convinced about this approach. The possible interplay among variables is lost. Moreover, a single-variable susceptibility assessment seems of little use. At present, one of the reasons why more sophisticated LSM methods are used is that they also have internal modules that assess the variable importance.

*That is right; we have run the model for each predictor variable selected for shallow landslides and old deep-seated landslides. The goal was to evaluate the extent to which these predictors can be used to differentiate between landslide and no landslide locations. This step is to help us to better understand the multivariate LR models, although we agree that this is not to be considered without caveats. We added a line to make sure to stress this better in the manuscript.*

**L451-452 / 509-510:** All predictors considered for both types of landslides where thus considered in the multivariate logistic regression models (Depicker et al., 2020).

**2.2. Specific remarks**

2.2.1. L27 which dynamics? Please, be more specific.

*Thanks for the comment, it is about the forest loss. But we removed this line to make the abstract shorter.*

2.2.2. L35 which susceptibility models?

*As written, these are the susceptibility models calibrated for the shallow landslides and old deep-seated landslides. But we removed this line to make the abstract shorter.*

2.2.3. L56-59. It depends also how the human intervention was designed and executed. There is a big difference if you just cut a slope and build a house (or a road), or if the cut is accompanied by some additional works (drainages, concrete walls, …). This should also be highlighted elsewhere in the manuscript when you write about this issue.

*In this rural environment, concrete walls are almost non-existent. As for the drainage systems, when present, they are very basic (one or two usually-unmaintained ditches on (both) side(s) of the road. Note also that most of the roads are dirt roads, frequently impacted by rill and gully erosion due to a lack of maintenance. The roads are therefore contributing to an undesigned concentration and rerouting of the runoff. We provide these details in the section 1.1.*

**L109-110 / 122-124:** Most roads are dirt roads and are poorly maintained, and there are no built-up walls (concrete, gabions) to stabilize the cut slopes.

2.2.4. Section 1.1 Besides describing the lithology, a short overview of the geological setting could be a nice addendum to this section.

*We agree with the relevance of the brief description of the geological setting of the study area.*

**L80-83 / 94-97:** A significant portion of the study area is made of lithologies from the Archaen, the Mesoproterozoic and the Neoproterozoic, with various degrees of chemical weathering and fracturing. Lastly formed rocks are the old Neogene basalts, highly weathered, that were deposited between 11-4 Ma years.

2.2.5. Fig. 1 For the cities, I suggest using a color that better stands out from the colors used for elevation. E.g. black. And you could also add it in the legend. I initially confused cities outside the study area with parts of the study area.

*Thanks for the comment. We adjusted the color for the cities but not for the elevation.*

**L98 / 112:**

[Figure]

2.2.6. L155-160: From what dates are the images? (This is explained later, but at this point of the manuscript, it is a spontaneous question: see my first general comment).

*Thanks for the comment. We indicate here the dates, also in line with your general comment.*

**L144 / 164: …** from 2005 to 2019…

**L147-148** / **167-168**: …(see Albino et al., (2015) and Dewitte et al., (2021) for technical explanation on the production of the DEM).

2.2.7. L174-178. Usually, a landslide is also considered shallow when the ratio depth/width or depth/length is small. I guess this is also your case?

*Our criterion was based on the estimated depth of the surface of rupture. This was assessed through the analysis of the main scarp shape. As we are looking at the conditions that prevailed at the source of the landslides, we did not investigate further their morphometry. Note that in general deep-seated landslides can also have a small depth/length ratio and that, on the other hand, shallow landslides can have a rather long runout (as is frequently observed in our study area).*

2.2.8. L225. This is not clear to me.

*The SRTM digital terrain model used for the topographic analysis is posterior to the occurrence of the old deep-seated landslides. Therefore, the digital terrain model is affected by the deformation due to the landslides. Calculating the slope values at the level of the main scarp for this type of landslide would give values that are the consequences of landslides rather than the causes of their origin. Hence, we placed a second point outside the landslide on the nearby slope that seems unaffected by landsliding. The morphology of this slope would better reflect the topographic similarity before the triggering of this landslide. We made the text clearer.*

**L239-242 / 265-268**: …In doing so we also avoid the selection of the highest point of the landslide that rarely corresponds to its initiation point (Dille et al., 2019). The digital elevation model used for the analysis (see Table 1) is posterior to the occurrence of the old deep-seated landslides. Therefore, …

**L244-245 / 270-272**: Calculating the slope values at the level of the landslide head for this type of landslide would give values that are the consequences of landslides rather than the causes of their origin.

2.2.9. Table 1. the meaning of "reference" in the second column is explained only later. This is confusing.

*We made it clearer.*

**L263 / 292**:* Each dummy variable is compared with the reference group.

2.2.10. Table 1. The forest dynamics information is very interesting. In my opinion, it deserves also a figure. Unfortunately, the figure comes only after some pages. This is another example of specific issues comprehended in my first general comment.

*Thank you for the remark. As replies to the Reviewers #1; we have moved, from result section, the figure on reconstructed LULC changes. We have also grouped the texts discussing this figure in Section 2.2.*

In 1955-58, 42 % of the territory was already deforested (Fig. 2a). From 1955-58 to 2016, the loss of forest continued, the forest cover decreasing from 58 % to 24 % of the study area. The area affected by the forest loss over the last 60 years is larger than the remaining permanent forest (Fig. 2b). The comparison of forest areas between 1955-58 and 2016 allows to consider four classes for the forest dynamics. Permanent forest corresponds to forest areas that are present at both dates. The forest loss class corresponds to forests present in 1955-58 that have disappeared in 2016. Since it is impossible to identify for each portion of the landscape the exact cause of forest loss, this class contains a mix of various forest management practices and other causes of forest cut/removal. The forest gain class represents the new forest that has appeared since 1955-58. Similarly, the causes associated with the occurrence of new forest are not exactly known; afforestation and natural forest regeneration being certainly drivers at play. Permanent anthropogenic environment (e.g. cropland, grassland, built-up lands) means that the landscape was not forested in both dates and it is assumed that it remained so during that period.

[Figure]

Figure 5: Forest cover dynamics over the last 60 years. (a) Forest cover in 1955-58 and 2016; (b) Areas of forest cover change between 1955-58 and 2016. Details for the images used in this figure are in Table 1.

2.2.11. Figure 2. The forest cover color hides the information about elevation. Didn't you already display the elevation in Fig 1? Here, you could just use hillshade and forest cover.

*We tested several ways of presenting this figure, trying colors that contrast better with the elevation and the forest cover. In our opinion, both layers should appear on the figure to show a densification of landslides in the mountainous areas and sparseness in the forest area. Using hillshade alone will not allow readers to locate the topographic context where the different types of landslides are found. We present both types of information as best as possible without hiding the information in the underlying layer.*

**L 361** / **403:**

[Figure]

2.2.12. 377 "these sources"

*We corrected the mistake.*

**L 392** / **435: …**these sources…

2.2.13. Table 4. It seems to me that the bedrock lithology has little influence in determining if a landslide will be shallow or deep seated. Maybe because the lithologies produce similar soils and the actual depth of soils (driven by morphology) is the real control?

*We indeed show that the lithology is of lesser importance in our study area. For the deep-seated landslides this is in agreement with the findings of Depicker et al. (2021b), which show that the various lithologies in the region have similar rock strength properties. As we also show that the topography and the presence of faults play a role, it is therefore no surprise that the role of lithology is somehow attenuated.*

*Note also that despite the fact that we use an unprecedented lithological information (in terms of accuracy and resolution), there is a lack of data on the regolith, its depths and the soil types. Field work confirms that soils and regolith can be very different in terms of type and depth, the latter one being highly variable spatially (sometimes over a very short distance of a few meters along a hillslope, one have regolith thickness that varies from > 10 m to nearly zero). The only way to provide an assessment of the regolith thickness at the regional scale is to assume that it varies with slope gradient. It is based on this assumption, that is further supported by the analysis of Depicker et al. (2021b) carried on shallow landslides in the region, that we explain the distribution of the shallow landslides in section 4.3.*

2.2.14. 478. I like that recent landslides are reasonably well predicted by a model trained with the old ones. This is like a multitemporal validation. It could be worth mentioning it.

*Thank you for your appreciation! We now mention that.*

**L474-475 / 537-538**: …which validates to some extend the multi-temporal predicting performance of the assessment. The…

2.2.15. 493-In an earlier part of the manuscript you mentioned that elevation can be considered a proxy for meteo-climatic characteristics. Why you discard this interpretation here?

*You are right. Thanks for the comment. We add a line about this aspect.*

**L 490-491 / 555**: …and increased orographic rainfall associated to these elevations (Fig. 8c).

2.2.16. 593 - Actually, the explanation may be that with this approach you artificially create incompleteness in your inventory. (this interpretation is in accordance with my general comment about frequency-area curves).

*In this section our interpretations take into account all 1013 shallow landslides inventoried (see Figure 6), hence there is no creation of incompleteness of the inventory. For some analyses of the frequency density curves excluding event-related landslides from the inventory, we only investigate the potential bias that landslide events could introduce. See Lines **417-429**/**469-483**.*

2.2.17. L604 the influence of vegetation on slope stability is somehow a relevant part of the phenomena you are investigating, but this is never mentioned explicitly. Why didn't

you openly prepare this issue in advance, and you don't mention it explicitly? Forest loss means (I think) reduced root cohesion and reduced evapotranspiration. I would mention it clearly. You could also make reference to some works such as

Masi, E. B., Segoni, S., & Tofani, V. (2021). Root Reinforcement in Slope Stability Models: A Review. *Geosciences*, *11*(5), 212.

Schwarz, M., Preti, F., Giadrossich, F., Lehmann, P., & Or, D. (2010). Quantifying the role of vegetation in slope stability: A case study in Tuscany (Italy). *Ecological Engineering*, *36*(3), 285-291.

Arnone, E., Caracciolo, D., Noto, L. V., Preti, F., & Bras, R. L. (2016). Modeling the hydrological and mechanical effect of roots on shallow landslides. *Water Resources Research*, *52*(11), 8590-8612.

Glade, T. (2003). Landslide occurrence as a response to land use change: a review of evidence from New Zealand. *Catena*, *51*(3-4), 297-314.

*Thank you for the comment, we added two references.*

**L 598** / **673**: … by reduced root cohesion and evapotranspiration due to forest loss(Glade, 2003; Masi et al., 2021), …

> 2.2.18. 608-612. I think there is (also) another explanation: the slope value you are using is an averaged value, while the built environment may be characterized by a locally steeper value. As instance, in a slope cut you could have a small 90° slope, which may not be well captured by the DTM. Even outside artificial environment, a similar situation may be present.

*As mentioned in the methodology, we study the occurrence factors at the scale of a pixel, i.e. a point, taken manually, in the middle of the main scarp for shallow landslides. We have also discarded the shallow landslides found in direct proximity to the roads. Nevertheless, we can indeed not ignore that the SRTM, because of its resolution, does not capture all the slope characteristics such as the cuts. See Lines* **237-245** / **263-273.**

> 2.2.19. 670-675. The stylistic writing of this part is so different from the rest of the paper. Here the sentences are very short and telegraphic. I suggest to better link them.

*Thanks for the comment, we have rewritten this part of work.*

**L 668-673** / **743-748**: Our analysis shows that the importance of human activities must be considered when investigating landslide occurrence in regions under anthropogenic pressure. This is particularly needed when one sees that the changing spatio-temporal patterns of landslides associated with these activities tend to further exacerbate the risks that the population face. On a more technical/methodological note, our study also demonstrates the importance of considering the timing of landslides in susceptibility and distribution assessments.

References

Depicker, A., Jacobs, L., Delvaux, D., Havenith, H.-B., Maki Mateso, J.-C., Govers, G. and Dewitte, O.: The added value of a regional landslide susceptibility assessment: The western branch of the East African Rift, Geomorphology, 353, 106886, doi:10.1016/j.geomorph.2019.106886, 2020.

Depicker, A., Jacobs, L., Mboga, N., Smets, B., Van Rompaey, A., Lennert, M., Wolff, E., Kervyn, F., Michellier, C., Dewitte, O. and Govers, G.: Historical dynamics of landslide risk from population and forest-cover changes in the Kivu Rift, Nat. Sustain., 4(11), 965–974, doi:10.1038/s41893-021-00757-9, 2021a.

Depicker, A., Govers, G., Jacobs, L., Campforts, B., Uwihirwe, J. and Dewitte, O.: Interactions between deforestation, landscape rejuvenation, and shallow landslides in the North Tanganyika–Kivu rift region, Africa, Earth Surf. Dyn., 9(3), 445–462, doi:10.5194/esurf-9-445-2021, 2021b.

Reichenbach, P., Rossi, M., Malamud, B. D., Mihir, M. and Guzzetti, F.: A review of statistically-based landslide susceptibility models, Earth-Science Rev., 180(March), 60–91, doi:10.1016/j.earscirev.2018.03.001, 2018.

**III. Reply to Referee #3**

In this works, authors explore the impact of land use change in landslides activity. To this end, they consider the influence of forest cover dynamic (assessed as gains and losses), roads (looking at old and recent roads) and mining activity on landslides occurrence.

The paper is quite complex since, to achieve their main goal, authors had to: (i) compile an exhaustive and accurate landslides inventory; (ii) assess the susceptibility of the area to shallow landslides and to old deep-seated landslides; (iii) assess the influence of geo-topographic and anthropogenic variables (i.e. forest loss, distance to roads and permanent anthropogenic environment) on landslides occurrence.

The paper focuses on an interesting topic, which is undoubtedly highly relevant in the field of landslides assessment. The overall manuscript is well structured, methods are appropriate, results are complete, accurate and reproducible. For all these reasons, in my opinion, it deserves publication on NHESS, with minor revisions.

*The authors thank the reviewer for his/her evaluation of the manuscript and for highlighting the scientific relevance of the work.*

**3.1. General remarks**

3.1.1. As the paper is quite complex, it results too long. Therefore I suggest streamlining the content avoiding repetitions. Even if globally it is well written, sentences are quite long and need to be elaborated in a more succinct way.

*We thank you for these remarks, and we worked on improving the manuscript by making it shorter and, when possible improving the sentences (keeping in mind that reviewer HH specifically paised the style of the manuscript). In particular, we deleted some lines as answered to the Referee # 2:*

*For example in:*

*Introduction section:* **L 55-57; 68-71; 142-144; 156-158.**

*Materials and methods section:* **L 205-206; 216-217; 260-262; 296-306; 360-361.**

*Results section:* **L 457-458; 472-474; 504-509; 521-522; 534-536.**

*Discussions section:* **L 563-567; 598-599; 603-605; 619-620.**

*We have also grouped the LULC analysis in Section 2.2, moving the now figure 2 from the results to this section. See Lines* **L208-221 / 231-244.**

3.1.2. Although the elaboration of a susceptibility maps is not the main objective of this research (indeed the authors applied to this end a classical and intuitive model both for susceptibility – i.e. logistic regression – and for the ranking of the importance of the predictors – i.e. frequency ratio – ), other methods existing in literature to this end should be mentioned and cited and your choice for the selected method justified.

*Our choices are better explained in the methodological section 2.3 as requested also by Referee #1 and #2.*

**L223-236** / **246-260**: Landslide susceptibility approaches are numerous and more or less complex in terms of modelling implementation and result interpretability (Reichenbach et al., 2018). In a regional analysis where our study area is included, Depicker et al.( 2020) used three susceptibility models, namely logistic regression, random forests, and support vector machines. These models gave relatively similar results in terms of quantitative performance and geomorphological plausibility. The same conclusion about marginal differences between susceptibility models can be drawn from many other studies. Since our study does not aim to develop a new methodology nor to show the ability to use complex methods; we relied on a logistic regression approach (Hosmer and Lemeshow, 2000) to determine the predictor variables related to the occurrence of the different types of landslides. Logistic regression isa straightforward method that has been widely used (Reichenbach et al., 2018) and that allows a rather easy interpretation of the results (e.g. Jacobs et al., 2018; Depicker et al., 2020).

Frequency ratio (Lee and Pradhan, 2007) models were used as a complementary approach to better understand the role of each variable in the contribution of the landslide occurrence in terms of process characterization. For example, when slope angle is highlighted by a logistic regression model as a significant variable, we still remain unaware of the types of slopes that actually influence the occurrence of landslides.

**3.2. Specific remarks**

3.2.1. Line 225 – The analysis was performed at the scale of one point per landslides, namely the centroid. Other authors use to extract randomly a certain percentage of points per events, or they consider the slope unit, or the highest pixel of each landslides (where the scarp is generally located). Please elaborate more this to justify your choice and its limits.

*We do not use the centroid of the entire landslide but a point that is manually placed at the center of the source/trigger area (line 239: we specify "trigger area"). We elaborate more on this point (see answer to Reviewer #2).*

**L239-242 / 265-286:** …In doing so we also avoid the selection of the highest point of the landslide that rarely corresponds to its initiation point (Dille et al., 2019). The digital elevation model used for the analysis (see Table 1) is posterior to the occurrence of the old deep-seated landslides. Therefore, …

**L244-245 / 270-272:** Calculating the slope values at the level of the landslide head for this type of landslide would give values that are the consequences of landslides rather than the causes of their origin.

    3.2.2. Line 252 - OpenStreetMap (OSM) is a digital map database of the world built through crowdsourced volunteered geographic information (VGI). Therefore, there is no systematic quality check performed on the data, and the detail, precision and accuracy varies across space. Can you be sure that no major changes in the network have occurred over the last 60 years or maybe they could have not been detected?

*We can confirm that there have been no major changes in the road network. Good knowledge of the study area and the analysis of very high-resolution Google Earth images allowed us to verify the road network proposed by OpenStreetMap.*

**L 267-268 / 295-296:** …Good knowledge of the study area and the analysis of the very high-resolution Google Earth images allowed us to verify the high accuracy of the road network proposed by OpenStreetMap.

    3.2.3. Legend of Fig.2: I propose to change "Landslide events" with "Landslides clustered events" or "Shallow landslides clusters".

*Thanks for the comment. We changed it with 'Landslides clustered events' as it constituted to both shallow and recent deep-seated landslides.*

**L 361 / 403:**

---

## Referee Report (RR1)

Dear Authors,

The papers present an interesting study to map and assess landslide susceptibility in an area where it is difficult to have detailed data both from the field and remote sensing.

The last version improved from the previous versions, and the question of the previous reviewer where addressed.
However, Some critical points, sometimes already evidenced by other reviewers, need to be clarified. In particular, the description and the discussion of landslide classification and the effect of anthropic activity.

**General comments**

Q1: Do you have only hillshade from Tandem-X DTM or the full data? In the second case, why did not use Tandem-X DTM instead of SRTM, especially for the shallow landslides? In addition, a figure showing the Tandem-X hillside should be added, for instance, in some panels of figure 4

Q2: The classification of landside should be made in different ways, in fact, there are at least three types of classification:
1. One based on landslide depth: shallow/deep-seated
2 The second one is related to natural/anthropic (mining or road)  causes
3 The third is based on landslide occurrence time  recent/historical (pre-1955)
4 The landslides type (e.g., Cruden and Varnes), as shown in figure 4, should be another classification parameter at least for deep-seated landslide

Thus, the five categories of landslides presented in this way, in my opinion, it is not the best solution for analysing the results and also for the discussion sections (5.4 should be removed and some parts included in the previous section).

A flow chart, table or multiple figures with of landslides classifications will also help the work on inventory.

Q3: The papers should focus more on distinguishing 1) shallow landslides related to rainfall events from 2) deep landslides,  because the two categories have pretty different suscetiblity models.
Then, you can consider the road, the land-use change (deforestation) and mining activity as predisposing factors (when necessary) for landslides and their weight in susceptibility models, as shown in Tables 4 and 5.
The impact of human activity should be alos shown with some more specific figures or plot, showing for instance, the comparison of shallow landslides trigger points density for different land-use conditions.

Q4: The deep-seated landslides should be classified in pre- 1955 and post- 1955 rather than recent/ancient.

Q5: Lines 615-625 consider that several recent deep-seated landslides are, probably, in several cases, close to a shallow landside depth and were easily detected on hillsahde and Google Earth images. While the pre- 1955 landslides detection is based on low-resolution aerial photos, which could have introduced a bias in their different characteristics  (the area). ?

Q8: As most of the shallow landslides were triggered by 2014 event, is it possible to have a more detailed description of the event? Such as rainfall data and distribution (from satellite data such as GPM)?

Q7: Conclusion should be rewritten based on previous points with the most significant numerical results

**Specific comments**

Figure 2 and Figure 3: These figures should be mixed: the forest cover is not necessary in figure 3. At the same time, it should be better to show a map that compares forest cover change overlapped with landslide distribution.

Figure 8: In the caption, better details the contents of each sub-figures

---

## Author Response (AR2)

Response to Referees comments:

**Characteristics and causes of natural and human-induced landslides in a tropical mountainous region: the Rift flank west of Lake Kivu (DR Congo)**

October 14, 2022

The authors would like to thank the editor and the two reviewers for their interest in our work and the relevance of their comments. Once again, we thank them for their appreciation of the quality of the work. We have taken into account these comments which revealed certain weaknesses in the presentation of our work. We note that reviewers #4 and #5 were not the reviewers involved in the first round of proofreading. We also note that Reviewer #5 is the one asking for a major revision. We also note that this reviewer does not want to proofread the new version of the revised manuscript. We hope that by showing here that the reviewer#5's comments could be easily addressed by small rephrasing/bringing some extra details, a third round of review will not be called. Indeed, with these two rounds of reviews (3 minor revisions/2 major) we have demonstrated that our work is sound both in terms of method/approaches and in terms of results/discussion. Most comments where associated with asking extra information.

On the specific concern that your also raised about the self-citation, we have removed whenever possible referencing to our work. Nevertheless, we want to recall that research on landslides in the tropics in general is very limited and it is especially so in Africa where the state of the art in the topic is unfortunately often associated with the work of the co-authors.

Our answers to the reviewers' comments are presented as follows: the reviewers' comments are shown in **black**; the answers are text in *blue italics*. And the revised texts are in **green**. The lines of the final manuscript are shown in **purple**, while the lines of the manuscript with the tracked changes are in **orange**.

**Contents**

**I.    Reply to Referee #4**

The authors presented a two-folded paper dealing with, on one hand, the redaction of landslide inventory through visual interpretation of aerial photos and google Earth imagery, and on the other hand, the landslide susceptibility mapping through Logistic Regression.Both the procedures are carried out with standard practices, and in general the paper do not present any particular novel content, however the analyses are clearly conducted and rigourously and critically discussed. Moreover, the environmental setting and the social implications in the territory make this paper a fair contribution in the landslide studies field.

*We appreciate that the reviewer understands that our work is not method-oriented. In our goal to study landslide processes in the context of human-induced environmental changes, we use several classical key methods; logistic regression being one of them and used together with frequency-size statistics and frequency ratio.*

**1.1.    General comments**

Before the final publication I have some concerns and some comments:
- First of all, I do not find the title very reflective of the aim of the paper. I would try to be more specific on that.

*We agree that a more specific title can be proposed. We now have: "Characteristics and causes of natural and human-induced landslides in a tropical mountainous region of Africa"*

- The introduction section is pretty synthetic and I find that a more solid comparison with the state of the art should be done, with more recent papers (dealing with use of remote sensing or modern technologies for the inventory part, and with machine/deep learning for the susceptibility, for instance). Moreover, I would move section 1.1 in another chapter, since it is the descritition of the study area, not an introduction. Moreover, this section needs a better geomorphological description of the area.

*We have modified the introduction in order to better support our visual-based approach for the building of the multi-temporal regional inventory of landslides; which is the key aspect of this research. For the susceptibility focus, however, we have not included such a section as this is only*

*one of the three key (and classical) methods (together with frequency size and frequency ratio) that is used for the landslide process analysis. In addition, as stressed earlier, the originality of our work is not framed around methodological achievements; we therefore do not want to insist on those aspects in the introduction.*

*The main changes in the introduction are in lines* **L58-64** / **L58-64**.

*We have moved section 1.1 to a new numbered section 2 and adapted the subsequent numbering accordingly. That was an oversight.*

**L58-64** / **L58-64**: New methodologies have been proposed in the past years to automatically map landslides with the use of, for example, Earth Observation data and machine learning (e.g. Prakash et al. 2021). However, such automotic approaches only perform well with recent landslides with a clear spectral signature. Futhermore, they are not always well adapted to an accurate understanding of the processes (Jones et al., 2021), especially when the landscapes are complex and highly influenced by human activities (Jacobs et al., 2018). The need for a visual identification of landslides is even more important when the movemets that are studied are older and have occurred at an unknow period of time, much before the availability of sattelite images (Pánek et al., 2021).

- In section 2.1 authors are claiming to estimate landslide depth through visual analysis. Could you please explain how this procedure is based? through which geomorphological features? How a 5 m depth can be estimated through visual interpretation?

*The paragraph has been rephrased in order to better explain this depth estimation. We stress that this is based on a robust field-based expertise and validation.*

**L177-190** / **L180-194**: The estimation of the depth of a landslide is important when the role of LULC is to be considered; shallow landslides being much more sensitive to the vegetation characteristics than deep-seated landslides (Sidle and Bogaard, 2016). In the literature, a landslide is usually defined as shallow when the depth of its surface of rupture ranges between 2 to 5 m (Keefer, 1984; Bennett et al., 2016; Sidle and Bogaard, 2016). Here, landslides with a depth < 5 m were considered as shallow. This criteria is based on the numerous field observations in the region that show that regolith can easily develop over a depth of several meters and that trees often show deep rooting systems. Following the approach of Depicker et al. (2020) and Dewitte et al., (2021), The distinction between deep-seated and shallow landslides was made by visually estimating the relative landslide depth from © Google Earth and the 5 m resolution TanDEM-X hillshade images. Extensive in situ-field observations of several hundreds of recent landslides where then carried out to valid the assessment. The landslides occurring in mining and quarrying sites were all classified as mining landslides. A specific attention was also given to the landslides occurring along roads. Mining and road landslides are assumed to be related to important anthropogenic changes in the

topography. Once they have occurred, field observations show that these landslides are commonly reworked and often further excavated. Therefore, for these two types of landslides, their depth was not assessed.

- Section 2.3, first paragraph: I think this section could be rewritten to make it more sound and coincise.

*The paragraph has been rephrased; we mainly left out the reference to our study area.*

**L258-260** / **L-265-267**: Landslide susceptibility approaches are commonly used to determine the factors that control the occurrence of landslides. There are numerous approaches which are more or less complex in terms of modelling implementation, data needs, and result interpretability (Reichenbach et al., 2018).

- Logistic regression and frequency ratio have been applied on a splitted dataset of landslides, selecting on one hand the shallow ones, and on the other hand the deep-seated ones. This was made including both fast- and slow-moving landslides? Or without any distinction between the type of landslide and the material involved (i.e., rockfalls and debris flow)?

*We have brought extra information about this, mostly in Section 3.1 (see lines **L204-206** / **L208-211**). We also better insist that we carry out our analysis with landslide categories instead of landslide types. We have made sure to adjust the text to this differentiation.*

**L204-206** / **L208-211**: We performed this analysis separately for five categories of the inventory considered together or in isolation: all landslides, old and recent deep-seated landslides, shallow landslides, mining landslides and road landslides (see Section 3.1)..

- Section 3.1, first and second line. The sentence does not sound correct or I'm not getting the message?
*We have simplified the sentence and move some of its information (that was more associated with a discussion) to line **L378-380** / **L397-400** at the start of Section 4.1. (Results)*

**L378-380** / **L397-400**:… Overall, we mapped 2730 landslides (Fig. 3a; Table 2). The landslides are diverse in terms of size, age and type (Fig. 4). The inventoried landslides cover ~3 % of the study area. The largest landslide is an old and deep-seated complex movement (426 ha), while the smallest detected landslide is a shallow debris avalanche (16 m$^2$).

Figure 4, if one of the pictures is showing any cluster displayed in the inventory, please mention it in the caption.

*We have indicated that Figure 4a shows part of the clustered event 2 landslides in Figure 3a.*

**L441** / **L463-464**: … the image illustrate a part of the landslides clustered event 2 shown on Fig. 3a) …

Section 4.4, lines 632-633. Landslide can be favoured by road cuts, not only in the region, but worldwide. Please mention some case outside from your study area. In general the paper is full of self-citations and I'm not pretty sure they are all necessary.

*We have added some references that were already used in the introduction.*

*With respect to the self-citation, we would like to recall that we are among the very few to study landslides in data scarce tropical environments, especially in Africa. Discussing our results with respect to the state of the art automatically implies some self-citation. We however would like to make sure that this is not perceived as such and whenever possible we have reduced reference to our work.*

**L674-676** / **L704-706**: … as observed not only in the region (e.g. Kubwimana et al., 2021), but worldwide (Froude and Petley, 2018; Sidle et al., 2006; Brenning et al., 2015; Arca et al., 2018; McAdoo et al., 2018; Vuillez et al., 2018; Muñoz-Torrero Manchado et al., 2021; Tanyaş et al., 2022); …

**II.     Reply to Referee #5**

The theme addressed in the manuscript "Natural and human-induced landslides in a tropical mountainous region: The Rift flank west of Lake Kivu (DR Congo)" by Mateso et al., is an interesting and relevant study that explores how landslides occurrence since the late 50's of the last century is impacted by anthropogenic changes related to forest cover, roads, and mining activities in a rural tropical mountainous region under high anthropogenic pressure. Nevertheless, the manuscript, presents some aspects that should be better addressed. These are detailed described in the Specific Comments and Technical Comments sections.

*The authors thank the reviewer for taking the time to read the manuscript in detail and to evaluate its content. We also thank him/her for the relevance of the comments that we were able to address in our revived version.*

**2.1.     Specific comments**

1)    Are some of the inventories used by authors covering landslides trigger by rainfall and by earthquakes. That should be turned clear in the landslide inventory section. In addition, the

classification of landslides as mining and road landslides despite their depth and type is not the most adequate and constrain possible explanations for their occurrence.

*For the recent landsides, i.e. those that have occurred in the last 60 years (the historical aerial photographs being used for this discrimination), rainfall is the only triggering factor that we have identified.  This is clearly mentioned in lines **119-148 / 121-151** in section 2*

*For the old landslides, as we are in a rift context where seismicity is present, we cannot exclude that some of the landslides in the inventory are associated with a seismic trigger. This is discussed in lines **119-148 / 121-151** in section 2.*

*In section 3.1 about the inventory of landslides, we made it sure that it is repeated briefly (see lines **L151-155** /**L154-158**). However, overall, we would like to recall that the goal of our research is not to look at the triggers, but at the causes of the landslides. Following the advice of reviewer 1 we have made it clearer from, for example, the title.*

**L151-155** /**L154-158:** The landslide inventory is a significant update of the inventory compiled by Depicker et al. (2020) who used only © Google Earth imagery for mapping the features whatever their type, age and rainfall, seismic or non-triggered origin as explained in Section 2. Since the focus of Depicker et al., (2020) was to study landslides over a much larger region than the one of the present research, their inventory was built on a limited search-time on our study area and without any field survey.

*With respect to the adequacy of the classification of the mining and road landslides, we would have appreciated some orientation. The comment does not really help us understand how we could improve our inventory. However, it is important to keep in mind that for both categories of landslides, we observed that their topography is very altered by the road and mining contexts. In addition, very often, road and mining landslides are used by locals as material sources, which can lead to altered morphology and behavior. Lastly, making sub-categories among road and mining landslides would reduce (if not hinder) the statistical analysis as we would fall short of having enough information. We therefore believe that our analysis with this classification scheme is the most appropriate one to serve our research objective. We have brought extra information in lines **L188-190** / **L192-194** to explain this.*

**L188-190** / **L192-194**: … Mining and road landslides are assumed to be related to important anthropogenic changes in the topography. Once they have occurred, field observations show that these landslides are commonly reworked and often further excavated. Therefore, for these two types of landslides, their depth was not assessed.

2) What this work brings differently from the works of Depicker and co-authors for example?

*This work is different from that of Depicker et al. 2020 in many ways:*

- *Depicker et al. (2020) used only Google Earth imagery with a limited time search as their goal was to look at landslide regional trends for a much larger region than our study area. Here, we have spent much more time on this task to identify as much landslides as possibe.*
- *We have achieved intense field investigation, visiting the region six times and validating more than 700 landslides over an area that represent 20% of the study area. Considering the remoteness of the region as well as its safety concerns, this is a achievement that we considered exceptional.*
- *In addition to Google Earth imagery, we have used 5 m resolution TanDEM-X DEM derived hillshade maps as well as hundreds of historical aerial photographs. It allow us to better discriminate between the depth of the processes as well as their timing.*
- *We have made differentiation between the landslide types, and grouped them in several natura and human-induced categories. .*

*Overall, our inventory contains 3 times more landslides than the inventory of Depicker et al. (2020). That information on the inventory can be found lines* **L538-540 / L567-569***, section 4. We have also in some parts improved the text to make sure that it is well understood that our research brought a significant and unprecedented new dataset. See lines* **L151-155** */***L154-158.**

**L538-540 / L567-569:** Despite its high precision, and the fact that with more than 2700 mapped landslides we have identified more than three times as many features as in the inventory of Depicker et al. (2020), we are aware that the dataset is still incomplete..

**L151-155** /**L154-158:** The landslide inventory is a significant update of the inventory compiled by Depicker et al. (2020) who used only © Google Earth imagery for mapping the features whatever their type, age and rainfall, seismic or non-triggered origin as explained in Section 2. Since the focus of Depicker et al., (2020) was to study landslides over a much larger region than the one of the present research, their inventory was built on a limited search-time on our study area and without any field survey.

*The work of other co-authors (Depicker et al., 2021a, 2021b; Dewitte et al., 2021) was either based on Depicker et al (2020) or on information that helped us to explain and discuss the regional context (Monsieurs et al., 2018a, 2018b, Dewitte et al., 2021); this latter information being not used as dataset in our research*

3) If I understood well only exists aerial photographs from late 50´s of the last century and from 2016 land cover ESA model. How authors stablished the correlation between landslides that occurred outside the time frame of these two land cover images (part of the landslides are

dated from 2005-2019 images) and the predisposing terrain conditions related with forest cover?

This is, in each part of this 60-year period occurred the forest loss, forest gain? For example (L. 208-218) how can someone interpret based on figure 2a that in 1955-58 48 % was already deforested. It means that the entire area was covered with forest before? The comparison only allows to compare land use cover changes between the two-time frames. I strongly believe that these limitations should be carefully considered and the dynamic component revised by authors.

*First of all, as explained in lines* **L96-97** / **L97-98***, the natural vegetation of the region is forest (Nzabandora and Roche, 2015). In addition, it is known that the region was already deforested for quite some time along the major road axes present in the 1950's (Depicker et al., 2021b). We have rephrased and added information in lines* **97-98** / **98-99** *to make sure that it is easily understood.*

**L97-98 / L98-99 :** The roads built during the late 19th and first half of the 20th centuries played a key role on further expanding this (Aleman et al., 2018).

*Indeed, for the pre-satellite era, historical photographs from the 1950's are the only existing source of information for this region. We have rephrased the text to make sure that it is clearly understood. We have also provided extra information on our justification for the use of the ESA dataset. See lines* **L234-237** / **L238-241**.

**L234-237** / **L238-241**: This satellite-based product has an accuracy of roughly 86 % in the region and has demonstrated its relevance in another study on landslides (Depicker et al., 2021). Note also that between 2016 and 2019, i.e. the date that corresponds to the most recent images in Google Earth used for the inventory, very little forest cover changes were observed.

*In section 3.1 (lines* **L151-155** /**L154-158***) we explain that the goal of our research is to complement the analysis conducted by Depicker et al. (2021b; see section 2) that focused on the impact of deforestation on shallow landslides over the last 20 years. Here, we therefore reconstructed the forest dynamics over the last ~60 years (Section 3.2). Depicker et al. (2021b) used satellite-derived products and could work on a yearly base. Here, we cannot do that has we only have two temporal frames. We have brought extra information in lines* **L232** / **L236-237** *to make sure that it is better understood.*

**L151-155** /**L154-158:** The landslide inventory is a significant update of the inventory compiled by Depicker et al. (2020) who used only © Google Earth imagery for mapping the features whatever their type, age, and rainfall, seismic or non-triggered origin as explained in Section 2. Since the focus of Depicker et al., (2020) was to study landslides over a much larger region than the one of

the present research, their inventory was not only built on a limited search-time on our study area but, also, without any field survey.

**L232** ⁄ **L236-237:** …these photographs being the only existing pre-satellite era source of information.

4) A composed figure with the predisposing factors should be considered by authors

*In the main text of the manuscript, we want to give priority to figures that show significant results. In addition, we provide already in Sections 2 and 3.2 as well as in figures 1 and 2 quite a significant amount of information on the environment that we study. Therefore, we added this suggested figure as supplementary material. Reference to this figure is made lines* **L283-285** / **L295-297**.

**L283-285** / **L295-297**: The purpose of this research is to examine the predictor variables (See supplementary Figure 1 for the predictor variables not displayed in the main manuscript) that contribute to the susceptibility of the different landslide categories.

[Figure]

5)  In the methods section authors should clearly describe the susceptibility modelling strategy and with which landslide inventory partitions intend to validate the susceptibility maps. For example, validate the susceptibility maps with road and mining landslides and why.

*We have rephrased sentences and added extra information in Section 3.3 to improve the method section. For example, in Lines* **270** */***281** *we insist on the fact that we look at the categories of landslides defined in section 3.1.*

**L270** */***L281:** The analysis was carried out according to the five categories of landslides defined in Section 3.1.

*For the validation of the susceptibility maps with road and mining landslides, this is explained Lines* **L474-479** */* **L497-503***:"Depicker et al. (2020) assessed the impacts of the size of the landslide training dataset to calibrate a landslide susceptibility model. They showed that the quality of a susceptibility assessment is questionable if the number of landslides is too small. In view of the low number of recent deep-seated, mining, and road landslides in the present study (Table 3), we did not calibrate susceptibility models from these three types of landslides. Instead, we tested these inventories against the two susceptibility models computed from the shallow and/or old deep-seated landslide datasets, from which we could derive prediction rates (Fig. 7)".*

6)   In the discussion section, please address better the possible bias that could overcome from the resampling of predictor variables for the SRTM resolution

*Our choice of using this resolution was clearly justified in the former version. Now we have rephased this part to improve its understanding. In lines* **L316-318** */* **330-332** *we say the following:*

**L316-318** */* **330-332:** Prior to analysis, the non topographically-derived predictor variables were resampled at the resolution of the SRTM DEM data; a resolution that is commonly used is many susceptibility analyses (Reichenbach et al., 2018) and that provided the best results in similar regions (e.g. Jacobs et al., 2018).

*Since we are analyzing thousands of landslides, we are looking at patterns and trends, not at characteristics of individual landslides. We therefore believe that using a resampled lithological information that provides actual information at a scale of 1/500,000 does not have an impact on our outputs. Similarly, lowering the resolution of the LULC datasets is expected to have no impacts.*

*Since uniformizing dataset at the resolution of SRTM data is a very common practice, we believe that such an issue does not need to be discussed in the specific case of our regional analysis.*

**2.2. Detailed comments**

L. 54: I agree with authors regarding deep-seated and shallow landslides differentiation but a discussion regarding the type of landslide is also acknowledge in this introduction section. The same should be better addressed in

In addition to rephrasing in the introduction (lines 48-49; line 53), we have improved the text at several locations to make sure that shallow and deep-seated landsides are here considered as 2 categories of processes, where several types of movements are present. In section 4.1 (lines 439-453) we explain this in detail.

L. 179-185: The classification of the landslide type in mining and road landslides does not seem the most interesting. Even so, what type of landslides are mapped? In addition, it's important to earlier in the manuscript to clarify which type of landslides authors are used to assess susceptibility. Only slides? Adjust terminology along the manuscript accordingly.

*We agree that we needed to make a clarification on the terminology. Section 4.1 (lines **L380-400 /L399-420**) details the types of landslides we have identified and explains well how they have been grouped into five categories for the frequency size, susceptibility and frequency ratio analyses. This grouping into categories is now also explained in the abstract (line **21/ 21**) as well as in other places in the text (lines **204 / 208-209; 270 /281**).*

**L380-400 /L399-420:** The landslides are grouped into five categories (Fig. 3a; Table 2):

- Old deep-seated landslides represent 45,5 % of the inventoried landslides and cover 93 % of the total landslide affected area. Most of these landslides are of the rock slide type. Rock avalanches, although much less frequent, are also present. Rockfalls can be associated with the presence of the main scarps of these old landslides. However, they have not been considered in the inventory and the subsequent analysis;
- Shallow landslides represent 40.4 % of inventoried landslides, but represent only 2.7 % of the total affected area. Most of these landslides are of the debris avalanche type. These landslides are all recent and clearly associated with rainfall. The landslides clustered events all fall in this category;
- Recent deep-seated landslides represent a small percentage of landslides (5.8 %) but cover an area (2.9 %) similar to shallow landslides. Most of the landslides are of the slide type. Their trigger, when identified, is associated with rainfall;
- Mining landslides (that also include quarrying landslides) represent 5.6 % of the inventoried landslides and cover 1.2 % of the total landslide affected area;
- Road landslides: the inventory shows that 115 landslides are located within 50 meters of roads. 60 of these landslides are shallow, 13 recent and deep-seated, 35 old and deep-seated, and 7 are mining landslides. Only the shallow and recent deep-seated landslides were

classified as road landslides; i.e. a total of 73 landslides The old deep-seated landslides located close to roads were retained in the old deep-seated landslide category because their timing is likely to precede road construction. The mining landslides were also retained in their respective category.

**L21/ L21:** …that we group into five categories…

For the use of the categories road and mining landslides, we refer to our reply to specific comment 1 where this issue was already raised and where we explain its relevance in the context of this research.

In section 2.1 a description of the base inventory constructed by Depicker et al (2020) is welcome. For example, what landslide inventory period is covered by the work of Depicker et al (2020)?

*We have added information to better clarify the added value of our work as compared to that of Depicker (see lines **151-155** / **154-158**). We also bring a more detailed answer with respect to specific question 2.*

**L151-155** / **L154-158:** The landslide inventory is a significant update of the inventory compiled by Depicker et al. (2020) who used only © Google Earth imagery for mapping the features whatever their type, age and rainfall, seismic or non-triggered origin as explained in Section 2. Since the focus of Depicker et al., (2020) was to study landslides over a much larger region than the one of the present research, their inventory was built on a limited search-time on our study area and without any field survey.

L. 142: Please address better what the mean of "differentiated between the processes"

*We have rephased this sentence (lines **L155-157** / **L158-160**) as the following: "Moreover, in our research we differentiated between the types (according to the updated Varnes' classification proposed by Hungr et al. (2014)) and timing of landsliding"*

**L155-157** / **L158-160**: Moreover, in our research we differentiated between the types (according to the updated Varnes' classification proposed by Hungr et al. (2014)) and timing of landsliding. We strongly relied on three image products:

L. 154-156: For authors what is consider a recent landslide? It is a landslide with less than 60 years? How distant in time is possible to relate landslides with a date of occurrence. This point regarding the definition of old and recent landslides, concerning landslide age needs to be clarified in my opinion.

*This definition of landslide "age" is clear to us and is provided here (lines **L169-171** / **L172-174**): "The historical aerial photographs allowed to differentiate between old deep-seated landslides (i.e. landslides with an unknown time of origin and already present on the photographs) and recent deep-seated landslides that have occurred during the last 60 years (i.e. after the acquisition of the photographs)."*

L. 162-165: the visual estimation of landslide depth is discussable. Please turn clear.

*This was also commented by reviewer 4. Here is our reply: "The paragraph has been rephrased in order to better explain this depth estimation. We stress that this is based on a robust field-based expertise and validation. "*

**L177-180** / **L180-194**: The estimation of the depth of a landslide is important when the role of LULC is to be considered; shallow landslides being much more sensitive to the vegetation characteristics than deep-seated landslides (Sidle and Bogaard, 2016). In the literature, a landslide is usually defined as shallow when the depth of its surface of rupture ranges between 2 to 5 m (Keefer, 1984; Bennett et al., 2016; Sidle and Bogaard, 2016). Here, landslides with a depth < 5 m were considered as shallow. This criteria is based on the numerous field observations in the region that show that regolith can easily develop over a depth of several meters and that trees often show deep rooting systems. Following the approach of Depicker et al. (2020) and Dewitte et al., (2021), The distinction between deep-seated and shallow landslides was made by visually estimating the relative landslide depth from © Google Earth and the 5 m resolution TanDEM-X hillshade images. Extensive in situ-field observations of several hundreds of recent landslides where then carried out to valid the assessment. The landslides occurring in mining and quarrying sites were all classified as mining landslides. A specific attention was also given to the landslides occurring along roads. Mining and road landslides are assumed to be related to important anthropogenic changes in the topography. Once they have occurred, field observations show that these landslides are commonly reworked and often further excavated. Therefore, for these two types of landslides, their depth was not assessed.

L. 170-171 "by selecting representative areas with various landslide and landscape characteristics". How the representative areas are selected and how are these areas are defined? Which percentage of study area was covered by field survey? The use of the criteria "various landslides" should also be clarified.

*We have clarified this part of the text explaining that 20% of the area was surveyed. See lines* **L194-197** / **L198-201**.

**L194-197** / **L198-201**: The work was carried out by selecting representative areas with various types of landslides and areas with less or no landslides. These areas, that cover a total of ~20% of the region, were selected based on different landscape characteristics (lithology, slope, LULC),

while taking into account accessibility and safety issues that prevent to access many places (Jaillon, 2020).

L. 188: Include in the manuscript the criteria that define a landslide event.

*We now provide a clearer definition. See lines* **L129-131** / **L131-134**.

**L129-131** / **L131-134**: A catalogue of > 150 accurately dated landslide events, i.e. landslides that can be clearly associated with a common well-defined triggering rainfall event over the same area, was compiled for the NTK Rift for the last two decades. .

L.191-192: why a maximum of 30 landslides per cluster? The concept of minus event, containing other isolated landslides (L. 193) was not perfectly clear to me.

*This methodological choice is now better explained with respect to the statistical soundness of the analysis. See lines* **L216-220** / **L221-224**

**L216-220** / **L221-224**: Thus, for the shallow landslides susceptibility analysis (see Section 3.2), we retained a maximum of 30 landslides per cluster, randomly sampled in order to strengthen the statistical analysis and avoid overfitting. The choice of this selection is also guided by the concern to have at least the minimum of data required for training and validating the susceptibility models (Depicker et al., 2020).

L. 250: Is missing from the list of predictor variables the forest dynamics between 1955-58 and 2016

**L288-290** */* **L302-304***: These are natural factors that influence landslide occurrence. Forest cover dynamics between 1955-58 and 2016 are presented among the anthropogenic predictors (see* **L302-303** */* **L317-318** *and table 1).*

L. 237-239: The difference between shallow landslides and deep-seated landslides is relevant if landslides are of the same landslide type, e.g., slides. Authors made this explanation latter on the manuscript, but that should be turned clear earlier, in section 2.1. Moreover, how is selected one point (pixel) per landslide. The landslide susceptibility predictive power provides significant differences depending on the use of the entire landslide area for training the susceptibility model considering larger usually deep-seated slides or shallow slides (usually smaller in size).

*Regarding your previous comment on this subject (2.2. Detailed comments), we have added information in section 3.1 to better explain this differentiation between processes and categories. We also clarify this better in section 4.1 (lines* **L380-400** */* **L399-420:** *).*
**L380-400** */* **L399-420:** The landslides are grouped into five categories (Fig. 3a; Table 2):

- Old deep-seated landslides represent 45,5 % of the inventoried landslides and cover 93 % of the total landslide affected area. Most of these landslides are of the rock slide type. Rock avalanches, although much less frequent, are also present. Rockfalls can be associated with the presence of the main scarps of these old landslides. However, they have not been considered in the inventory and the subsequent analysis;
- Shallow landslides represent 40.4 % of inventoried landslides, but represent only 2.7 % of the total affected area. Most of these landslides are of the debris avalanche type. These landslides are all recent and clearly associated with rainfall. The landslides clustered events all fall in this category;
- Recent deep-seated landslides represent a small percentage of landslides (5.8 %) but cover an area (2.9 %) similar to shallow landslides. Most of the landslides are of the slide type. Their trigger, when identified, is associated with rainfall;
- Mining landslides (that also include quarrying landslides) represent 5.6 % of the inventoried landslides and cover 1.2 % of the total landslide affected area;

Road landslides: the inventory shows that 115 landslides are located within 50 meters of roads. 60 of these landslides are shallow, 13 recent and deep-seated, 35 old and deep-seated, and 7 are mining landslides. Only the shallow and recent deep-seated landslides were classified as road landslides; i.e. a total of 73 landslides The old deep-seated landslides located close to roads were retained in the old deep-seated landslide category because their timing is likely to precede road construction. The mining landslides were also retained in their respective category.

*We have rephrased this part where we explain how the point (pixel) per landslide is selected (Lines* **L272-276** / **L284-288**). *There we also justify this approach to avoid spatial autocorrelation as well as a temporal bias for the large and old landslides. .*

**L272-276** / **L284-288**: The point is manually positioned in the central region of the visually delineated landslide's source area to represent the conditions as close to reality as possible that cause its occurrence. In doing so we also avoid the selection of the highest point of the landslide that rarely corresponds to its initiation point (Dille et al., 2019). As stressed by Tanyaş et al., (2018), landsides growth with time. Therefore, considering one pixel per landslide instead of its whole source area allows to avoid a temporal-induced bias.

L.239: Please turn clear what is the centre of the landslide trigger area, this concept is not clear to me. Do you mean rupture zone, depletion zone, initiation area? Please, see my previous comment on this topic.

*We now explain this in a clearer way. See lines* **L271-272** / **L284-286**. *Note that here we prefer the use of source area instead of depletion area.*

**L272-273 / L284-285:** The point is manually positioned in the central region of the visually delineated landslide's source area to represent the conditions as close to reality as possible that cause its occurrence.

L. 247-249: How spatial predisposing predictors allow to discuss triggering conditions? I apologize, but I think the description is too general and maybe it should be preferable to direct the explanation for preparatory conditions for landsliding.

*This part makes reference to the key principles in tectonic geomorphology that are associated with interactions of tectonic, landscape and climate. We have rephrased the paragraph and added relevant references to support this statement. See Lines* **L283-287** / **L295-300**.

**L283-287** / **L295-300**: The purpose of this research is to examine the predictor variables (See supplementary Figure 1 for the predictor variables not displayed in the main manuscript) that contribute to the susceptibility of the different landslide categories. As such we mainly investigate the causes of the landslides. Nevertheless, the predictors highlighted by the susceptibility analysis may also help to discuss triggering conditions since the tectonic, landscape and climate of a region are commonly interlinked (Whipple, 2009; Whittaker, 2012).

L. 270 – 272m. Authors mentioning that the few recent landslides observed along these roads confirm the assumption that the direct impact of the main roads on the occurrence of landsides is limited. This are not results? Move for the results section.

*This is an observation that can indeed be interpreted as results. However, that is an observation that is necessary to define our methodological choices. Since here we do not present yet specific results on the inventory, we prefer to leave this sentence here (***L307-309** / **L321-323***).*

L. 287-292: I understand authors idea, but how much of these shallow landslides occurred along or near the fault zone? If authors are using the distance to faults to correlate with weathering, are not these weathering materials more prone for landslding? This should be better addressed.

*We understand the relevance of your question. However, for the shallow slides, we have to keep in mind that the inventory is temporally biased. We explain our methodological choice in lines* **L324-329** / **L339-344**: ➔ *For the analysis of deep-seated landslides, the predictor variables associated with anthropogenic activities were excluded. For the shallow landslides, the 'distance to faults' variable was also excluded. As explained earlier, the shallow landslide inventory represents a narrow time window of observation. As such, the spatial distribution of the shallow landslides could be biased by the stochastic pattern of the recent heavy rainfall events and anthropogenic disturbances rather than being the reflect of the longer-term impact of weathering conditions associated with seismicity."*

L. 355- Seven landslides are mining landslides? But this category is not related to road landslides? Are landslides classified in more than one category? L. 357-358: please avoid repetitions, the idea that road landslides include only the landslides located within 50 m of roads was already written above in the paragraph.

*7 of the 115 landslides located with a 50 m buffer from the roads are mining landslides; these mining landslides being retained in their respective category. We have rephased this part to make sure that it is better understood. See lines:* **L393-398** */* **L414-420**.

**L393-398** */* **L414-420**: Road landslides: the inventory shows that 115 landslides are located within 50 meters of roads. 60 of these landslides are shallow, 13 recent and deep-seated, 35 old and deep-seated, and 7 are mining landslides. Only the shallow and recent deep-seated landslides were classified as road landslides; i.e. a total of 73 landslides The old deep-seated landslides located close to roads were retained in the old deep-seated landslide category because their timing is likely to precede road construction. The mining landslides were also retained in their respective category.

L. 367-372: how much of the 634 shallow landslides of the October 2014 rainstorm event occurred in the study area? Only 14? This was not clear to me. Where they occur specifically on the shores of Lake Kivu? The blue dots on figure 3 seems few for these 634 landslides mentioned.

*The scale of the map does somehow bias our interpretation of the number of the identified landslides. Figure 4.a presents a rather small partial view of the landslide event 2 (see location in Figure 3) were more than 50 sources of shallow debris avalanches are identified. In the image below, this figure 4a view is localized within the whole area impacted by the landsides of this clustered event.*

[Figure]

*As you can see from the figure above, there are not just 14 landslides, but 634. The text is quite clear (**L405-407** / **L426-430**), the 14 landslides are those that were triggered before the major rainfall event of October 2014 and were subsequently reactivated during this event.*

**L405-407** / **L426-430**: We identified several shallow landslides clustered events. One of the events is related to the Kalehe rainstorm of October 2014 (Fig. 3a: event 2; Fig. 4a) reported by Maki Mateso and Dewitte (2014). This rainfall triggered 634 shallow landslides, 346 of them being connected to talwegs and providing materials to 17 debris flows.

L. 388-390: 25% more deep-seated landslides were identified in the field but not used in the analysis to avoid biases due to overrepresentation. Are the predisposing conditions associated to these landslides like the ones used to assess landslides susceptibility?

If not, how good can be the predictive susceptibility map. Validating the produced landslide susceptibility map with those landslides could help answering this question.

*We visited 786 (i.e. 25%) of the landslides identified in the images. In addition, we identified only in the field alone an extra 126 landslides; these field landslides were not considered in the analysis to avoid bias and overrepresentation (**L421-429** / **L443-451**). We have brought a little change to the original text that we believed clearly explained the issue. See lines **L428** / **L450**.*

**L428** / **L450:**… extra 24% of observations (Table 3: see column FN).

L. 421: the fact that 72 % of the landslides are found in areas of forest loss really means that they occurred with that predisposing condition? These landslides occurred before or after the forest loss? The dates of the landslides not always match the date of the forest cover image.

*Indeed, 72 % of the landslides have occurred in a deforested context. In lines* **L236-237** */ **L240-241*** *we have added an explanation to stress that between 2016 (the ESA most forest coverage used in this analysis) and 2019 (the most recent images from Google Earth sued for the analysis), no substantial change is observed.*

**L236-237** */ **L240-241***: Note also that between 2016 and 2019, i.e. the date that corresponds to the most recent images in Google Earth used for the inventory, very little forest cover changes were observed.

*Regarding the last part of your question, we have already answered that in your question 3 in specific remarks. You can refer to the following text in the manuscript (**L239-241** / **L243-245**).*

**L239-241** / **L243-245:** Knowing that the natural vegetation of the study area is forest (Section 2.1), in 1955-58, 42 % of the territory was already deforested (Fig. 2a). From 1955-58 to 2016, the loss of forest continued, the forest cover decreasing from 58 % to 24 % of the study area.

L. 462-463: The authors analysis conclude that shallow landslides are 2.5 times more likely to occur as deforestation increases? Please see my previous comment on this topic.

*The 2.5 times comes from the logistic regression analysis, more specifically to the odds ratio (Table 5). We think that with the clarifications that we brought earlier, this is something that does not need to be further commented.*

L. 473: This steep was not clear from the methods section. If the susceptibility model was produced for shallow slides, why validate it with mining landslides. It seems to me from figure 5b that mining landslides are closer to the characteristics of recent deep-seated landslides than to shallow slides. This validation with landslides datasets not used to construct the susceptibility maps is not clear from the methods section.

*Your question has been answered above in question 5 (specific comments: **L474-479** / **L497-503**). In the method section, the paragraph in lines **L283-287** / **L295-300** has been added for clarification. Overall we believe the changes made throughout the text clarify many methodological aspects.*

**L283-287** / **L295-300**: The purpose of this research is to examine the predictor variables (See supplementary Figure 1 for the predictor variables not displayed in the main manuscript) that contribute to the susceptibility of the different landslide categories. As such we mainly investigate

the causes of the landslides. Nevertheless, the predictors highlighted by the susceptibility analysis may also help to discuss triggering conditions since the tectonic, landscape and climate of a region are commonly interlinked (Whipple, 2009; Whittaker, 2012).

L. 525-536: landslides in the permanent anthropogenic environment – what is the relationship between the short time for which the shallow landslides were inventoried in the study area and the fact that the study area was not altered by mechanized farming? When mechanized farming begins in the study area? Which terrain anthropogenic conditions lead to slope instability in these anthropogenic landslides. This should be better addressed.

*Shallow landslides are known to disappear rather quickly from these tropical landscapes, sometimes in a matter of a few years (Dewitte et al., 2022). In addition, mechanized farming is known to accelerate such a process of landslide scar alteration. In our study area, as we have several images cover over the 2005-2019 period, we are confident that we have not a shallow landslide inventory biased by time-related disappearance. Furthermore, as said in section 2 and repeated here, there is no mechanized farming in our study area. A lot of background information to understand this paragraph is provided in the first paragraph of this discussion section 5.1, of which we have provided extra details (lines* **L543** / **L571-572**). *We therefore believe that in this revised version of the manuscript the text that the reviewer highlighted in his/her comment does not need to be modified (lines* **L567-578** / **L595-606**).

*With respect to the terrain anthropogenic conditions that lead to slope instability, at this regional level of analysis, only assumption can be made in a more or less robust way. That is the goal of our discussion.*

**L542** / **L571-572**: …although, here, since we have used several image covers from Google Earth, this issue should be nuanced.
L. 545: the model show that seismic activity play a dominant role in deep-seated landslide distribution in the study area – ok. Were all the old deep-seated landslides earthquake triggered, if not, are the terrain conditions of old deep-seated rainfall-triggered landslides like the previous ones?

*In lines* **L587-588** / **L615-616**, *we clearly say that the model suggests (we remain moderated with the use of "seem" in the sentence) a role of seismic activity on the occurrence of deep-seated landsides. Then we discuss this in lines* **L614-625** / **L641-654**. *We believe that the modifications that were brought at several places in the text to explain the potential role of triggering factors make now this paragraph easier to be understood. We therefore decided not to modify it.*

L. 592-593: the regional susceptibility model shows that deforestation is the most important factor. Again, and I apologize for my repetition, this is difficult to understand since landslide dates do not have a landcover photography that allows to stablish a direct relationships with landcover. This

section of drivers for shallow slides should be better addressed concerning the relationship between forest cover type and landslides.

*We believe that the modifications that were brought at several places in the text to explain the date of the landslides and the temporal aspects of LULC make now this paragraph easier to be understood. We therefore decided not to modify it*

L. 615: why authors say that colluvium deposits result in a concentration of susceptible places. This is not clear to me. Please address better.

*The availability of a certain minimum depth of colluvium is necessary for the occurrence of a shallow landslide. We have added information and a reference to support this. See lines* **L655-657** / **L684-686**.

**L655-657** / **L684-686**: This could be extra material available for the formation of landslides; the colluvium supply and a minimum depth of material being recognized as playing a key role in the occurrence of shallow landslides (Parker et al., 2016).

Section 4.4. Why should the produced susceptibility models validate the mining and road landslides: Are these two categories related to shallow or deep-seated landslides?

*We have already explained better in the revised manuscript, based on earlier comments, that we do not aim to validate the susceptibility models with mining and road landslides. In addition, we have rephrased lines* **L664-665** / **L694-695** *to better explain this.*

*We have also better explained the reason why we cannot provide a depth criteria with the definition the mining and road landslides (lines* **L188-190** / **L192-194**). *We also explain that there is not enough mining and road landslides to calibrate statistically-robust susceptibility models (lines* **L474-478** /**L497-502**: *"Depicker et al. (2020) assessed the impacts of the size of the landslide training dataset to calibrate a landslide susceptibility model. They showed that the quality of a susceptibility assessment is questionable if the number of landslides is too small. In view of the low number of recent deep-seated, mining, and road landslides in the present study (Table 3), we did not calibrate susceptibility models from these three types of landslides. Instead, we tested these inventories against the two susceptibility models computed from the shallow and/or old deep-seated landslide datasets, from which we could derive prediction rates (Fig. 7).")*
).

*Therefore, looking at how mining and road landslides are distributed on the two susceptibility models is an original way that we have found to analyze their distribution.*

**L664-665** / **L694-695**: The poor prediction rates of mining and road landslides when compared to the two shallow and deep-seated susceptibility models (Fig.7) shows that they respond to different environmental factors.

**L188-190** / **L192-194**: Mining and road landslides are assumed to be related to important anthropogenic changes in the topography. Once they have occurred, field observations show that these landslides are commonly reworked and often further excavated. Therefore, for these two types of landslides, their depth was not assessed.

**2.3.    Technical comments**

L. 58, L. 123: Monsieurs et al., 2018 and references therein along the manuscript: please check and indicate if the references to the work of Monsieurs et al., 2018 is 2018a or 2018b.

*Well thanks for the remark, we have corrected* **L132** / **L135** and **L135** / **L137**

**L133** / **L135** and **L136** / **L139**: …Monsieurs et al., 2018b; …

L. 87: please check "altitude" or "elevation", If related with relief it should be elevation.

*Thanks for the remark, we have corrected in lines L96 and L534*.

**L94** / **L95** and **L526** / **L555**: …elevation…

L. 203-205: Create an appropriate reference in the reference list for the work of Smets et al., to be submitted) and cite accordingly in the manuscript.

*We have removed the reference to these authors*.

L. 794 and L. 797 please indicate which reference is 2021a and 2021b

*L849-853: Thank you for the remark, we have specified the two references*

**L838-842** / **L878-882**:
 Heri-Kazi, A. B. and Bielders, C. L.: "Cropland characteristics and extent of soil loss by rill and gully erosion in smallholder farms in the KIVU highlands, D.R. Congo," Geoderma Reg., 26(May), e00404, doi:10.1016/j.geodrs.2021.e00404, 2021a.

Heri-Kazi, A. B. and Bielders, C. L.: Erosion and soil and water conservation in South-Kivu (eastern DR Congo): The farmers' view, L. Degrad. Dev., 32(2), 699–713, doi:10.1002/ldr.3755, 2021b.

Items not cited in the manuscript but present in the References list: L. 760: Trumbore et al 2019….; L. 773: Fisher et al 2012….; L. 779: Geenen 2012; L. 810: Kampunzu et al 1998….; L. 928: Vanacker et al 2003….; Hungr et al., 2014. Please check references/reference list and respective citation on the manuscript.

*Trumbore is a co-author in the article by Drake et al. (2019) see* **L101** */-***103**

**L102** / **L103**: … construction (Musumba Teso et al., 2019; Drake et al., 2019).

*We have added the citation from Fisher et al. (2012) see* **L67** / **L68**

**L68** / **L68**: …available on © Google Earth (Fisher et al., 2012), which are widely…

*We have added the Geenen (2012) citation in the text, see* **L90** / **L93**

**L91** / **L92-93**: … artisanal and small-scale mining and quarrying (Van Acker, 2005; Geenen, 2012;…

*The reference Kampunzu et al. (1998) has been added from the list of references.*
**L89** / **L90:** …fracturing (Kampunzu et al., 1998).

*We have updated the Vanacker et al. (2003) citation in* **L38** / **L39** *and* **L45** / **L46**.
**L39** / **L39** and **L46** / **L46:** Vanacker et al., 2003;…

*Hungr et al. (2014) have been well cited on the legend of Figure 4 see L447 and the reference has been put on the texts added on the article L166.*

**L156** / **L159**: … (according to the updated Varnes' classification proposed by Hungr et al. (2014) and timing …

**L439** / **L461**: … Examples of landslide types (according to Varnes' new classification – Hungr et al., 2014).

**2.4.   Figures and Tables**

Page 3 – Figure 1: Check graphic representation of scale bars of maps 1a and 1b, uniformize and add bottom line in scale bar of map 1b. The same should be checked in all figures. Include in legend of figure 1a the meaning of the yellow star in the upper-right part of figure. It represents the location of study area?

*The graphical representations of the scale bars in Figure 1 are corrected. We have included the location of the study area in the legend. See **L106** / **L108***

[Figure]

Page 6 – Figure 2: In figure 2a and 2b consider having the same legend for classes "no forest" in figure 2a and "permanent anthropogenic environments" on figure 2b, if representing the same class variable. The same for "Forest cover" in figure 2a and "permanent forest" in figure 2b.

*We have corrected the figure.* **L253** / **L259**

[Figure]

Page 10 – Figure 3: Somewhere along the manuscript text clarify what criteria define a heavy rainfall event. Furthermore, the shallow slides associated to each cluster should be identified. Consider adding polygon lines (limits) grouping all the landslides for each event. Figure 3b include scale bar. The additional landslides identified only in the field could be associated with some of the rainfall events identified on figure 3a? Please turn clear along text.

*Lines 132-138 provide a good explanation of what intense rainfall events consist of in our study area. We have added the polygons including all shallow landslides from a rainfall event (see Line* **L401** */* **L421***).*

*We have added the scale bar for Figure 3b.*
*The © Google Earth images are well clear to map landslides related to these major rainfall events.*

[Figure]

Page 17 – Figure 8: What the meaning of values above the vertical bars since they do not match with both right and left side bars. Is the frequency ratio score. Please adjust caption.

*These are frequency ration scores. We have added some words to specify the meaning of these values L527*

[revised manuscript text omitted]

---

## Author Response (AR3)

Response to Referees comments:

**Characteristics and causes of natural and human-induced landslides in a tropical mountainous region: the Rift flank west of Lake Kivu (DR Congo)**

December 29, 2022

The authors thank the editor and the two reviewers for their proofreading of the manuscript. Reviewer #4, after the second reading, accepted the manuscript as is for publication. We have taken note of the comments of reviewer #6; not all of them being relevant. Indeed, key comments/ are on issues that were already well addressed in the manuscript. We however have tried to improve the manuscript wherever possible based on these. We have provided explanations to all the questions raised. In addition, we have spotted a few typos and made a few small changes here and there to improve the readability.

Our answers to the reviewers' comments are presented as follows: the reviewers' comments are shown in **black**; the answers are text in *blue italics*. And the revised texts are in **green**. The lines of the final manuscript are shown in **purple**, while the lines of the manuscript with the tracked changes are in **orange**.

**Contents**

**Reply to Referee #6**

Dear Authors,

The papers present an interesting study to map and assess landslide susceptibility in an area where it is difficult to have detailed data both from the field and remote sensing.

The last version improved from the previous versions, and the question of the previous reviewer where addressed.

However, Some critical points, sometimes already evidenced by other reviewers, need to be clarified.

In particular, the description and the discussion of landslide classification and the effect of anthropic activity.

*The authors thank reviewer's #6 for reading the manuscript and for reviewing the former replies. We have paid attention to all the comments and made a specific effort to address and clarify the classification issues. This has led to changes in the abstract, in the text (lines 21-24 / 21-24) and in the conclusion (729-737/ 741-749) .*

*We would like to recall that the goal of our research goes beyond the sole aspect of a landslide susceptibility assessment and that, in that sense, we do not fully agree with the first sentence of the review.*

**L21-24 / 21-24**: … into five categories that are adapted to study the impact of human activities on slope stability: old (pre 1950's) and recent (post 1950's) deep-seated landslides, shallow landslides, mining landslides and road landslides. We analyze the landslides according to this classification protocol via frequency-area statistics, frequency ratio distribution and logistic regression susceptibility assessment.

**729-737/ 741-749:** On a more technical/methodological note, our study also demonstrates the importance of considering the timing and the depth of landslides as well as the differentiation between mining and road landslides. While several well-known landslide classification systems are used at the international level (Hungr et al., 2014; Sidle and Bogaard, 2016), these systems are not framed around the combination of the differentiation criteria that are used in this research. Our study does propose a unique effort at classifying landslide types in order to investigate them in the context of the Anthropocene. We believe that our mapping effort and classification protocol is the most adapted (based on strong field observation and comprehensive understanding of the landscape) in this case to address the problem of natural and human-induced landslides in the region. However, it certainly needs improvement to be used in a more universal way.

**1.1 General comments**

Q1: Do you have only hillshade from Tandem-X DTM or the full data? In the second case, why did not use Tandem-X DTM instead of SRTM, especially for the shallow landslides? In addition, a figure showing the Tandem-X hillside should be added, for instance, in some panels of figure 4.

*- In regard to the first part of your question; indeed, the TanDEM-X digital elevation model at 5 m resolution is unique for our study area. As pointed out on line 168 / 169, the TanDEM-X model does not cover our entire study area. In addition, the 5 m resolution DEM is associated with noise and artifact. TanDEM-X data cannot indeed be used at this resolution without such a caveat (Albino et al., 2015; Jacobs et al., 2018). As described by Dewitte et al. (2021), the production of this DEM was aimed at analyzing visually topographic features at an unprecedented resolution. The choice of the SRTM DEM at 30 m was further guided by the fact that it is commonly used in susceptibility analyses at regional scales (Reichenbach et al., 2018). In addition, in a specific landslide susceptibility analysis carried out in the Rwenzori Mountains, a neighbor region of Uganda, Jacobs et al. (2018) evidenced that the 30 m resolution DEM clearly outperforms higher resolution products (here at 10 m) arguing that there is always a trade-off between model complexity and data needs. This was already explained in lines on lines* **326-328** / **335-337**. *We have improved this section to make the point clearer.*

**L326-328** / **335-337:** Furthermore, in a region of Uganda located in a relative proximity, Jacobs et al. (2018) evidenced that the 1 arc second SRTM DEM clearly outperforms higher resolution products derived, in that specific case, from TanDEM-X.

*- For the second part of your question on visualizing hillshade information, we have made some attempts, but it does not show anything relevant at this scale of the photos. Therefore we have brought extra information directly in the text (lines* **165-167** / **173-175**).

**L165-167** / **173-175:** Despite some artefacts present in the DEM (Albino et al., 2015), this resolution allows to visually identify geomorphological features relevant for characterizing landslide processes (Dewitte et al., 2021).

Q2: The classification of landside should be made in different ways, in fact, there are at least three types of classification:

1. One based on landslide depth: shallow/deep-seated

2 The second one is related to natural/anthropic (mining or road) causes

3 The third is based on landslide occurrence time recent/historical (pre-1955)

4 The landslides type (e.g., Cruden and Varnes), as shown in figure 4, should be another classification parameter at least for deep-seated landslide

Thus, the five categories of landslides presented in this way, in my opinion, it is not the best solution for analysing the results and also for the discussion sections (5.4 should be removed and some parts included in the previous section).

A flow chart, table or multiple figures with of landslides classifications will also help the work on inventory.

*Our study does propose a unique effort at classifying landslide types in order to make an analysis between the natural and human induced processes. We have better formulated this in lines **156-159** / **164-167** and lines **729-737** / **741-749** in order to better highlight this specificity. We believe that our mapping effort and classification protocol is the most adapted (based on strong field observation and comprehensive understanding of the landscape) in this case to address the problem of natural and human-induced landslides in the region. We therefore do not change this part of our research which is also strongly supported by the international literature (see section 3.1) and has not been questioned by the other reviewers.*

*To our knowledge, we are the first ones to really address the problem of mining landslides at such a scale. We therefore believe that section 5.4 is a key text that must be kept in the manuscript.*

*We paid attention to better highlight the originality of our work in order to clarify our methodological choices.*

Q3: The papers should focus more on distinguishing 1) shallow landslides related to rainfall events from 2) deep landslides, because the two categories have pretty different suscetiblity models.

Then, you can consider the road, the land-use change (deforestation) and mining activity as predisposing factors (when necessary) for landslides and their weight in susceptibility models, as shown in Tables 4 and 5.

The impact of human activity should be also shown with some more specific figures or plot, showing for instance, the comparison of shallow landslides trigger points density for different land-use conditions.

*We are not sure to really understand what the reviewer is asking. With respect to the first point, we clearly say that shallow landslides are rainfall-triggered. This is clearly expressed for instance in lines **398-400** / **407-409**. In addition, we clearly make a distinction between shallow landslides and deep-seated landslides. That point is the key of the whole susceptibility analysis (see Figure 7 for example).*

*With respect to the use of predisposing factors, we explicitly use distance to roads and forest cover dynamics in the models. This is clearly justified in the text (see lines **310-323** / **319-333**) and presented in Table 4. Here also what the reviewer is suggesting is already clearly done. With respect to the mining activities, one key predisposing factor that can be identified is clearly the altered topography. However, we clearly explain in lines **310-323** / **319-333** that there is no topographic dataset that exists at a spatial resolution that can capture these tiny changes in the landscapes. This is one of the reasons why we do not perform a susceptibility analysis of the mining landslides.*

Q4: The deep-seated landslides should be classified in pre- 1955 and post- 1955 rather than recent/ancient.

*We have explained in lines* **172-176 / 181-184** *that old deep-seated landslides are pre-1955 and recent deep-seated landslides are post-1955. We think that to simplify the understanding of the terminology the terms old and recent deep-seated landslides should be maintained.*

*Lines* **172-176 / 181-184***: The historical aerial photographs allowed to differentiate between old deep-seated landslides (i.e. landslides with an unknown time of origin and already present on the photographs) and recent deep-seated landslides that have occurred during the last 60 years (i.e. after the acquisition of the photographs).*

Q5: Lines 615-625 consider that several recent deep-seated landslides are, probably, in several cases, close to a shallow landside depth and were easily detected on hillsahde and Google Earth images. While the pre- 1955 landslides detection is based on low-resolution aerial photos, which could have introduced a bias in their different characteristics (the area)?

*We are aware that there is always an uncertainty associated with the depth characterization of the landsides. However, our extensive field investigation shows that our inventory is reliable (Table 3), especially with respect to the hundreds of deep-seated landslides that are considered in this analysis. We can therefore assume that gteh outputs of our regional analysis are not biased by a few outliers. At a more general level, that justifies our methodological strategy to work with data-driven models; these models being perfectly adapted to datasets that will always contain uncertainties.*

*In lines* **160-171 / 168-179** *and* **549-561 / 558-571***, we discuss this issue of resolution of the photographs (1 m² resolution vs 0.5 m² for the satellite images). However, as quantified in Table 3, such a small issue of detection cannot explain the fact that on average old deep-seated landslides are 4 times larger than recent deep-seated landslides (Table 2).*

Q8: As most of the shallow landslides were triggered by 2014 event, is it possible to have a more detailed description of the event? Such as rainfall data and distribution (from satellite data such as GPM)?

*In the region, detailed description of the rainfall characteristics of specific landslide events is difficult and associated with lots of uncertainties (Monsieurs et al., 2018a; Monsieurs et al., 2018b; Monsieurs et al., 2019a; Monsieurs et al., 2019b). Indeed, the only products we can rely on are satellite measurements (there is no rain gauge in the site or in the very close vicinity of the event that was operational at the time of the event). For example, IMERG (10 km resolution, available every 3 hours), is certainly one of the most adapted satellite-based sources of information (Nakulopa et al., 2022). However, km-scale resolution products are not valid for a spatial characterization of the rainfall over such a small area. In addition, such products may miss the extremes locally* (Monsieurs et al., 2018a). *In addition, in the region, studies on precipitation thresholds triggering landslides show that the triggering event does not necessarily depend on the daily precipitation (of the day of the event) but may be associated with antecedent conditions of several  days (Monsieurs et al., 2019a; Monsieurs et al.,*

*2019b). According to the daily GPM satellite data, the event did not occur on the day of maximum precipitation (see circled in red). We have presented the October 25, 2014 event extensively in the manuscript (lines* **416-422 / 425-431***); we do not believe further details are necessary, as our study is not interested in investigating precipitation thresholds triggering landslides.*

[Figure]

Q7: Conclusion should be rewritten based on previous points with the most significant numerical results

*We believe that the conclusion should not be rewritten as most of the comments raised by the reviewer were not very appropriate. In addition, we do not really understand what the reviewer measn by "most significant numerical results; some insights would have been welcome. Nevertheless, based on the comments raised on the landslide classification system used in this research, we have added a last paragraph which we believe increases the international dimension of this work (lines* **729-737/ 741-749***) .*

**729-737/ 741-749:** On a more technical/methodological note, our study also demonstrates the importance of considering the timing and the depth of landslides as well as the differentiation between mining and road landslides. While several well-known landslide classification systems are used at the international level (Hungr et al., 2014; Sidle and Bogaard, 2016), these systems are not framed around the combination of the differentiation criteria that are used in this research. Our study does propose a unique effort at classifying landslide types in order to investigate them in the context of the Anthropocene. We believe that our mapping effort and classification protocol is the most adapted (based on strong field observation and comprehensive understanding of the landscape) in this case to address the problem of natural and human-induced landslides in the region. However, it certainly needs improvement to be used in a more universal way.

**1.2   Specific comments**

Figure 2 and Figure 3: These figures should be mixed: the forest cover is not necessary in figure 3. At the same time, it should be better to show a map that compares forest cover change overlapped with landslide distribution.

*Figure 2 was put in the methodology section on request of previous reviewers. By mixing the two figures, we will have a map overloaded with information. In addition, some of the information in two figures may disappear (such as Figure 2a).*

*Note that we here provide an improved version of figure 2, the layout of the coordinate system of figure 2b being made more readable (and visually lighter). For Figure 3, we have corrected a typo in the legend, "Shallow landslides clustered event" is replace by "Shallow landslide clustered events".*

Figure 8: In the caption, better details the contents of each sub-figures

*We have added extra details in the caption **530-531** / **539-540**.*

**L530-531** / **539-540**: Figure 8: Frequency distribution for shallow and deep-seated landslides in function of key predictor variables. Figures c, d, and e allow a multivariate comparison of the predictors.

**1.3   References**

Albino, F., Smets, B., D'Oreye, N., and Kervyn, F.: High-resolution TanDEM-X DEM: An accurate method to estimate lava flow volumes at Nyamulagira Volcano (D. R. Congo), J. Geophys. Res. Solid Earth, 120, 4189–4207, https://doi.org/10.1002/2015JB011988, 2015.

Dewitte, O., Dille, A., Depicker, A., Kubwimana, D., Maki Mateso, J.-C., Mugaruka Bibentyo, T., Uwihirwe, J., and Monsieurs, E.: Constraining landslide timing in a data-scarce context: from recent to very old processes in the tropical environment of the North Tanganyika-Kivu Rift region, Landslides, 18, 161–177, https://doi.org/10.1007/s10346-020-01452-0, 2021.

Hungr, O., Leroueil, S., and Picarelli, L.: The Varnes classification of landslide types, an update, Landslides, 11, 167–194, https://doi.org/10.1007/s10346-013-0436-y, 2014.

Jacobs, L., Dewitte, O., Poesen, J., Sekajugo, J., Nobile, A., Rossi, M., Thiery, W., and Kervyn, M.: Field-based landslide susceptibility assessment in a data-scarce environment: the populated areas of the Rwenzori Mountains, Nat. Hazards Earth Syst. Sci., 18, 105–124, https://doi.org/10.5194/nhess-18-105-2018, 2018.

Monsieurs, E., Kirschbaum, D. B., Tan, J., Maki Mateso, J.-C., Jacobs, L., Plisnier, P.-D., Thiery, W., Umutoni, A., Musoni, D., Bibentyo, T. M., Ganza, G. B., Mawe, G. I., Bagalwa, L., Kankurize, C., Michellier, C., Stanley,

T., Kervyn, F., Kervyn, M., Demoulin, A., and Dewitte, O.: Evaluating TMPA Rainfall over the Sparsely Gauged East African Rift, J. Hydrometeorol., 19, 1507–1528, https://doi.org/10.1175/JHM-D-18-0103.1, 2018a.

Monsieurs, E., Jacobs, L., Michellier, C., Basimike Tchangaboba, J., Ganza, G. B., Kervyn, F., Maki Mateso, J.-C., Mugaruka Bibentyo, T., Kalikone Buzera, C., Nahimana, L., Ndayisenga, A., Nkurunziza, P., Thiery, W., Demoulin, A., Kervyn, M., and Dewitte, O.: Landslide inventory for hazard assessment in a data-poor context: a regional-scale approach in a tropical African environment, Landslides, 15, 2195–2209, https://doi.org/10.1007/s10346-018-1008-y, 2018b.

Monsieurs, E., Dewitte, O., and Demoulin, A.: A susceptibility-based rainfall threshold approach for landslide occurrence, Nat. Hazards Earth Syst. Sci., 19, 775–789, https://doi.org/10.5194/nhess-19-775-2019, 2019a.

Monsieurs, E., Dewitte, O., Depicker, A., and Demoulin, A.: Towards a Transferable Antecedent Rainfall—Susceptibility Threshold Approach for Landsliding, Water, 11, 2202, https://doi.org/10.3390/w11112202, 2019b.

Nakulopa, F., Vanderkelen, I., Van de Walle, J., van Lipzig, N. P. M., Tabari, H., Jacobs, L., Tweheyo, C., Dewitte, O., and Thiery, W.: Evaluation of High-Resolution Precipitation Products over the Rwenzori Mountains (Uganda), J. Hydrometeorol., 23, 747–768, https://doi.org/10.1175/JHM-D-21-0106.1, 2022.

Reichenbach, P., Rossi, M., Malamud, B. D., Mihir, M., and Guzzetti, F.: A review of statistically-based landslide susceptibility models, Earth-Science Rev., 180, 60–91, https://doi.org/10.1016/j.earscirev.2018.03.001, 2018.

Sidle, R. C. and Bogaard, T. A.: Dynamic earth system and ecological controls of rainfall-initiated landslides, Earth-Science Rev., 159, 275–291, https://doi.org/10.1016/j.earscirev.2016.05.013, 2016.

---

## Author Response (AR4)

Author's Response:

**Characteristics and causes of natural and human-induced landslides in a tropical mountainous region: the Rift flank west of Lake Kivu (DR Congo)**

January 19, 2023

Dear editor,

Thank you for having accepted the manuscript.

With respect to the statement in line 128, we refer to a work (Depicker et al., 2021b) that is targeting erosion associated with the occurrence of new shallow landslides. The erosion is among other things defined from a combination of landslide frequency and landslide size assessments.

We have clarified the sentence. Here is what we propose: "A more detailed investigation of the annual evolution of the forest cover over the last 20 years showed that deforestation increases the erosion due to the occurrence of new shallow landslide erosion 2-8 times during a period of approximately 15 years before it eventually falls back to a level similar to forest conditions."

We have corrected the typo in line 549. The two sentences are now merged: "Despite its high precision, and the fact that with more than 2700 mapped landslides we have identified more than three times as many features as in the inventory of Depicker et al. (2020), we are aware that the dataset is still incomplete."

Thank you for your comments.

Best regards

Jean-Claude